# Exploratory Preference Optimization: Harnessing Implicit Q*-Approximation for Sample-Efficient RLHF

**Tengyang Xie**[*]
UW-Madison
tx@cs.wisc.edu

**Dylan J. Foster**[*]
Microsoft Research
dylanfoster@microsoft.com

**Akshay Krishnamurthy**
Microsoft Research
akshaykr@microsoft.com

**Corby Rosset**
Microsoft Research
corbyrosset@microsoft.com

**Ahmed Awadallah**
Microsoft Research
ahmed.awadallah@microsoft.com

**Alexander Rakhlin**
MIT
rakhlin@mit.edu

## Abstract

This paper investigates a basic question in reinforcement learning from human feedback (RLHF) from a theoretical perspective: how to efficiently explore in an online manner under preference feedback and general function approximation. We take the initial step towards a theoretical understanding of this problem by proposing a novel algorithm, *Exploratory Preference Optimization* (XPO). This algorithm is elegantly simple—requiring only a one-line modification to (online) Direct Preference Optimization (DPO; Rafailov et al., 2023)—yet provides the strongest known provable guarantees. XPO augments the DPO objective with a novel and principled *exploration bonus*, enabling the algorithm to strategically explore beyond the support of the initial model and preference feedback data. We prove that XPO is provably sample-efficient and converges to a near-optimal policy under natural exploration conditions, regardless of the initial model's coverage. Our analysis builds on the observation that DPO implicitly performs a form of *Bellman error minimization*. It synthesizes previously disparate techniques from language modeling and theoretical reinforcement learning in a serendipitous fashion through the lens of *KL-regularized Markov decision processes*.

## 1 Introduction

Reinforcement learning from human feedback (RLHF) is a central tool to align language models to human values and elicit useful behavior (Christiano et al., 2017; Bai et al., 2022; Ouyang et al., 2022). Using human-labeled preference data, RLHF achieves enhanced capabilities using a modest amount of data compared to unsupervised pre-training (on the order of tens of millions versus trillions of tokens) by treating the language model as a "policy" and optimizing it with reinforcement learning techniques.

Even though RLHF is typically only applied with preference data from humans or other language models, one might hope that it has potential to produce super-human capabilities because recognizing novel behavior and insights is typically easier than *generating* novel behavior. Indeed, it is often much easier to verify correctness of a given proof or program than it is to produce one from scratch. By repeatedly generating new proposals and labeling them with human feedback, a language model could gradually push beyond the boundary of human capabilities. Unfortunately, even with the great disparity in difficulty between generation and verification, a major barrier to achieving enhanced capabilities via RLHF is the volume of human feedback, i.e., *sample complexity*, required by existing methods. Thus, a promising research direction is to develop sample-efficient methods for RLHF.

A natural way to address the sample efficiency problem for RLHF is to augment algorithms with *online exploration*. Online exploration exploits interactive access to human or AI feedback by deliberately encouraging the model to produce diverse, novel responses. RLHF algorithms that exploit online feedback have received limited investigation, and in spite of encouraging initial results, existing approaches either do not update the language model (Dwaracherla et al., 2024), or engage

---

[*]Equal contribution

in purely passive exploration (Guo et al., 2024; Gao et al., 2024), with no mechanism to encourage novelty or diversity. Passive exploration is intuitively insufficient, as we are unlikely to generate novel and correct proofs by chance; we make this precise in Proposition 2.1. Thus, the full potential of online exploration as a new paradigm for language model training has yet to be realized.

In this paper, we take a first step toward developing a theoretical understanding of efficient exploration in language models. The central challenge in equipping language models with deliberate exploration is to efficiently navigate the vast, combinatorially large space of token sequences to find responses for which feedback will be maximally informative. The contemporary theory of reinforcement learning offers—at a conceptual level—solutions to this problem, providing algorithm design principles for exploration that can optimally take advantage of problem structure and achieve sample efficiency to the best extent one can hope for (Jiang et al., 2017; Agarwal et al., 2019; Foster and Rakhlin, 2023). However, the most powerful approaches in this space are computationally intractable in the general RL setting (Jiang et al., 2017; Jin et al., 2021; Foster et al., 2021), and prior attempts to adapt them to RLHF either make unrealistic modeling assumptions (i.e., do not allow for general function approximation) (Xu et al., 2020; Novoseller et al., 2020; Pacchiano et al., 2021; Wu and Sun, 2023; Zhan et al., 2023b; Du et al., 2024; Das et al., 2024), or are computationally inefficient and not feasible to faithfully implement (Chen et al., 2022; Wang et al., 2023; Ye et al., 2024). Can we, perhaps by specializing to language modeling, develop simple, yet provably sample-efficient online exploration methods for RLHF?

## 1.1 CONTRIBUTIONS

We propose a new algorithm for online exploration in RLHF, *Exploratory Preference Optimization* (XPO), which is simple—a one-line change to (online) Direct Preference Optimization (DPO; Rafailov et al. (2023); Guo et al. (2024))—yet enjoys the strongest known provable guarantees. XPO augments the DPO objective with a novel and principled *exploration bonus*, empowering the algorithm to explore outside the support of the initial model. We show that XPO is provably sample-efficient, and converges to a near-optimal language model policy under natural exploration conditions (Jin et al., 2021; Xie et al., 2023; Zhong et al., 2022). Critically, and in contrast to prior work, our theory holds irrespective of whether the initial model is sufficiently exploratory on its own. To summarize:

> XPO *offers the first simple, yet provably sample-efficient online exploration algorithm for RLHF with general function approximation.*

**Technical highlights.** Our design and analysis of XPO uses previously disparate techniques from language modeling and theoretical reinforcement learning, combining them in a serendipitous fashion through the perspective of *KL-regularized Markov decision processes* (Neu et al., 2017).

1. First, generalizing Rafailov et al. (2024), we observe that DPO can be viewed as implicitly performing *Bellman error minimization* (Xie and Jiang, 2020) to approximate the optimal value function $Q^\star$ in a *KL-regularized MDP*. We use this to provide a novel KL-regularized regret decomposition.

2. Then, we show that *global optimism* (Jiang et al., 2017; Jin et al., 2021; Xie et al., 2023), a powerful RL exploration technique that has classically been viewed as computationally intractable (Dann et al., 2018; Kane et al., 2022; Golowich et al., 2024), can be implemented in any KL-regularized MDP with deterministic transitions (generalizing language modeling) by adding a surprisingly simple exploration bonus to the DPO objective. This yields the XPO objective.

We expect our analysis techniques and perspective to be useful more broadly. In particular, the guarantees for XPO hold not just for language models, but for any RL problem with a stochastic starting state and (potentially unknown) deterministic transition dynamics ("Deterministic Contextual MDP").

**Concurrent work.** Two concurrent and independent works posted to arXiv in the same week as this paper, Cen et al. (2024); Zhang et al. (2024), propose algorithms that equip DPO with exploration bonuses similar to XPO. On the theoretical side, both works are restricted to the contextual bandit formulation of RLHF, and do not consider the general reinforcement learning framework in this work or make the connection to $Q^\star$-approximation and KL-regularized MDPs. Compared to our results, which give provable sample complexity guarantees with general function approximation, Zhang et al. (2024) do not provide sample complexity guarantees, while Cen et al. (2024) provide guarantees only for linear contextual bandits. In addition, and importantly, the sample complexity guarantees in Cen et al. (2024) have exponential dependence on the KL regularization parameter, which our results avoid.

We mention in passing that another concurrent work of Liu et al. (2024b) applies a similar bonus—with a flipped sign—to implement pessimism in *offline* RLHF; this is complementary to the online setting we focus on, and the analysis techniques and assumptions are quite different.

## 2 BACKGROUND

This section contains necessary background to present our main results. We begin by recalling the standard formulation of reinforcement learning from human feedback from offline data (Section 2.1), then introduce the *online feedback* model and highlight the need for systematic exploration (Section 2.2).

**Notation.** For an integer $n \in \mathbb{N}$, we let $[n]$ denote the set $\{1, \ldots, n\}$. For a set $\mathcal{X}$, we let $\Delta(\mathcal{X})$ denote the set of all probability distributions over $\mathcal{X}$. We adopt standard big-oh notation, and write $f = \widetilde{O}(g)$ to denote that $f = O(g \cdot \max\{1, \text{polylog}(g)\})$ and $a \lesssim b$ as shorthand for $a = O(b)$.

### 2.1 REINFORCEMENT LEARNING FROM HUMAN FEEDBACK

We study RLHF in a general reinforcement learning formulation which subsumes the *token-level MDP* formulation considered in prior work (Rafailov et al., 2024), but is somewhat broader.

**Markov decision processes.** We consider an episodic finite-horizon Markov decision process framework. Formally, a horizon-$H$ MDP $M = (H, \mathcal{S}, \mathcal{A}, P, r, \rho)$ consists of a (potentially very large) state space $\mathcal{S}$, action space $\mathcal{A}$, probability transition function $P : \mathcal{S} \times \mathcal{A} \to \Delta(\mathcal{S})$, reward function $r : \mathcal{S} \times \mathcal{A} \to \mathbb{R}$, and initial state distribution $\rho \in \Delta(\mathcal{S})$. We assume without loss of generality that the state space is *layered* such that $\mathcal{S} = \mathcal{S}_1 \cup \mathcal{S}_2 \cup \cdots \cup \mathcal{S}_H$, where $\mathcal{S}_h$ is the set of states reachable at step $h$, and $\mathcal{S}_h \cap \mathcal{S}_{h'} = \varnothing$ for $h \neq h'$. A (randomized) policy is a mapping $\pi : \mathcal{S} \to \Delta(\mathcal{A})$, and induces a distribution over trajectories $\tau = (s_1, a_1), \ldots, (s_H, a_H)$ and rewards $r_1, \ldots, r_H$ via the following process. The initial state is drawn via $s_1 \sim \rho$, then for $h = 1, \ldots, H$: $a_h \sim \pi(s_h)$, $r_h = r(s_h, a_h)$, and $s_{h+1} \sim P(s_h, a_h)$. We let $\mathbb{E}_\pi[\cdot]$ and $\mathbb{P}_\pi[\cdot]$ denote expectation and probability under this process, respectively, and define $J(\pi) = \mathbb{E}_\pi\left[\sum_{h=1}^H r_h\right]$. We assume that $\sum_{h=1}^H r_h \in [0, R_{\max}]$ almost surely for a parameter $R_{\max} > 0$. For a trajectory $\tau$ and policy $\pi$ we define $r(\tau) \coloneqq \sum_{h=1}^H r(s_h, a_h)$ and $\pi(\tau) \coloneqq \prod_{h=1}^H \pi(a_h \mid s_h)$.

In the context of language modeling, the main object of interest is the *token-level MDP* (Rafailov et al., 2024). Here, $s_1 \sim \rho$ represents a prompt, each action $a_h$ represents a token (with $\mathcal{A}$ representing the vocabulary), and the state $s_h = (s_1, a_1, \ldots, a_{h-1})$ is the prompt and sequence of tokens so far. The language model is represented by a policy $\pi$, which maps the current context $s_h = (s_1, a_1, \ldots, a_{h-1})$ to a distribution over the next token $a_h$. The trajectory $\tau = (s_1, a_1), \ldots, (s_H, a_H)$ produced by this process can be interpreted as the language model's response to the prompt $s_1$; we will occasionally use the terms "trajectory" and "response" synonymously in this context.

Our main results apply to any *Deterministic Contextual MDP* (DCMDP) for which the initial state is stochastic, but subsequent transition dynamics are deterministic and potentially unknown. This formulation encompasses but strictly generalizes the token-level MDP.

**RLHF with offline data.** In the classical RLHF formulation (Christiano et al., 2017; Bai et al., 2022; Ouyang et al., 2022), we assume access to a dataset $\mathcal{D}_{\text{pref}} = \{(\tau_+, \tau_-)\}$ of labeled preference data. Each pair of trajectories (responses) $(\tau_+, \tau_-)$ represents a positive and negative example; both trajectories begin from the same initial state (prompt) $s_1$, and are generated by first sampling a pair $(\tau, \widetilde{\tau})$ via $\tau \sim \pi_{\text{ref}} \mid s_1$ and $\widetilde{\tau} \sim \pi_{\text{ref}} \mid s_1$ in the underlying DCMDP $M$ (e.g., token-level MDP), and then ordering them as $(\tau_+, \tau_-)$ based on a binary preference $y \sim \mathbb{P}(\tau \succ \widetilde{\tau} \mid s_1)$. Here, $\pi_{\text{ref}}$ is a *reference policy* (language model), which is typically obtained via supervised fine-tuning, and the *preference* $y \sim \mathbb{P}(\tau \succ \widetilde{\tau} \mid s_1)$ is obtained from a human or AI annotator. Following a standard assumption (Christiano et al., 2017; Ouyang et al., 2022; Rafailov et al., 2023), we assume that preferences follow the *Bradley-Terry* model (Bradley and Terry, 1952): For trajectories $\tau$ and $\widetilde{\tau}$ both beginning with $s_1$,

$$\mathbb{P}(\tau \succ \widetilde{\tau} \mid s_1) = \frac{\exp(r(\tau))}{\exp(r(\tau)) + \exp(r(\widetilde{\tau}))}. \tag{1}$$

Based on the preference dataset $\mathcal{D}_{\text{pref}}$, the goal is to learn a policy $\widehat{\pi}$ with high reward. Following prior theoretical works on RLHF, we consider a *KL-regularized* reward objective (Xiong et al., 2023;

Ye et al., 2024), defined for a regularization parameter $\beta > 0$, via

$$J_\beta(\pi) := J(\pi) - \beta \cdot \sum_{h=1}^{H} \mathbb{E}_\pi[D_{\mathrm{KL}}(\pi(\cdot \mid s_h) \parallel \pi_{\mathrm{ref}}(\cdot \mid s_h))] = \mathbb{E}_\pi\left[r(\tau) - \beta \log \frac{\pi(\tau)}{\pi_{\mathrm{ref}}(\tau)}\right]. \quad (2)$$

We aim to compute a policy $\widehat{\pi}$ such that $\max_\pi J_\beta(\pi) - J_\beta(\widehat{\pi}) \leq \varepsilon$ for some small $\varepsilon > 0$. Such a guarantee means that $\widehat{\pi}$ near-optimally maximizes reward, yet stays relatively close to $\pi_{\mathrm{ref}}$ (as a function of $\beta$). The choice of $\beta > 0$, which is important for safety and reliability, is typically viewed as a domain specific hyperparameter (Tang et al., 2024a). Our main focus in this paper is the *small-$\beta$* regime, which allows $\widehat{\pi}$ to meaningfully deviate from $\pi_{\mathrm{ref}}$ and generate potentially novel responses. Notably, by taking $\beta$ sufficiently small, it is possible to translate suboptimality bounds for the regularized reward into bounds for the unregularized reward (e.g., Zhu et al., 2023; Zhan et al., 2023a).

We refer to this setting as *offline RLHF* because the algorithm relies only on the offline dataset $\mathcal{D}_{\mathrm{pref}}$ for training, and does not perform any active data collection.

**Direct preference optimization (DPO).** Initial approaches to offline RLHF (Christiano et al., 2017; Ouyang et al., 2022) proceed by first estimating a reward function $\widehat{r}$ from $\mathcal{D}_{\mathrm{pref}}$ using the Bradley-Terry model, then optimizing an estimated version of the KL-regularized objective in Eq. (2) using policy optimization methods like PPO, i.e., $\widehat{\pi} \approx \mathrm{argmax}_{\pi \in \Pi} \mathbb{E}_\pi\left[r(\tau) - \beta \log \frac{\pi(\tau)}{\pi_{\mathrm{ref}}(\tau)}\right]$. The starting point for our work is an alternative approach introduced by Rafailov et al. (2023), Direct Preference Optimization (DPO). DPO is motivated by a closed-form solution for the policy that optimizes the KL-regularized objective in Eq. (2), and condenses the two-step process above into a single policy optimization objective, removing the need for reward function estimation. Concretely, DPO solves[1]

$$\widehat{\pi} = \mathrm{argmin}_{\pi \in \Pi} \sum_{(\tau_+, \tau_-) \in \mathcal{D}_{\mathrm{pref}}} - \log\left[\sigma\left(\beta \log \frac{\pi(\tau_+)}{\pi_{\mathrm{ref}}(\tau_+)} - \beta \log \frac{\pi(\tau_-)}{\pi_{\mathrm{ref}}(\tau_-)}\right)\right] \quad (3)$$

for a user-specified policy class $\Pi$, where $\sigma(x) := \frac{\exp(x)}{1+\exp(x)}$ is the sigmoid function.

## 2.2 Online Feedback and Exploration in RLHF

DPO and other offline RLHF methods have achieved great success in language model alignment, but are fundamentally limited to behaviors that are well-supported by the initial model $\pi_{\mathrm{ref}}$ and data $\mathcal{D}_{\mathrm{pref}}$. RLHF with *online feedback* offers a promising approach to move beyond this limitation by collecting feedback from responses sampled from the model *during training* (Guo et al., 2024).

Formally, the protocol proceeds in $T$ rounds. At each round $t$, we receive an initial state $s_1^{(t)}$ and sample two responses $\tau \sim \pi^{(t)} \mid s_1$ and $\widetilde{\tau} \sim \pi^{(t)} \mid s_1$ from the current policy $\pi^{(t)}$. The prompts are then labeled as $(\tau_+^{(t)}, \tau_-^{(t)})$ and added to the preference dataset via $\mathcal{D}_{\mathrm{pref}}^{(t+1)} \leftarrow \mathcal{D}_{\mathrm{pref}}^{(t)} \cup \{(\tau_+^{(t)}, \tau_-^{(t)})\}$, which is then used to compute an updated policy $\pi^{(t+1)}$. In practice, the prompts are typically labeled via human feedback or AI feedback (e.g., a larger, more powerful language model (Guo et al., 2024; Rosset et al., 2024)); we assume the preferences $\mathbb{P}(\tau^{(t)} \succ \widetilde{\tau}^{(t)} \mid s_1^{(t)})$ follow the Bradley-Terry model in Eq. (1).

## 2.3 The Necessity of Deliberate Exploration

Existing approaches to online RLHF adapt offline techniques by applying them iteratively. As an example, *Online* DPO (Guo et al., 2024) proceeds as follows:[2]

1. Compute $\pi^{(t)}$ by solving the DPO objective in Eq. (3) with the current preference dataset $\mathcal{D}_{\mathrm{pref}}^{(t)}$.

2. Sample $\tau^{(t)}, \widetilde{\tau}^{(t)} \sim \pi^{(t)} \mid s_1^{(t)}$, then label as $(\tau_+^{(t)}, \tau_-^{(t)})$ and update $\mathcal{D}_{\mathrm{pref}}^{(t+1)} \leftarrow \mathcal{D}_{\mathrm{pref}}^{(t)} \cup \{(\tau_+^{(t)}, \tau_-^{(t)})\}$.

We refer to such an approach as *passive exploration*, as the responses are sampled directly from the policy $\pi^{(t)}$ without an explicit mechanism to encourage diversity. The following proposition shows that passive exploration is insufficient to discover novel behavior: Unless the initial policy $\pi_{\mathrm{ref}}$ has good coverage, Online DPO can fail to learn a near-optimal policy.

---

[1]We adopt the convention that the value of the DPO objective is $+\infty$ if $\pi$ does not satisfy $\pi \ll \pi_{\mathrm{ref}}$.

[2]The closely related *Iterative* DPO approach (Xu et al., 2023; Tran et al., 2024) proceeds in the same fashion, but samples a large batch of preference pairs from each policy $\pi^{(t)}$, and performs fewer updates.

**Proposition 2.1** (Necessity of deliberate exploration). *Fix* $\beta \in (0, \frac{1}{8}\log(2))$, *and consider the* bandit *setting* ($H = 1$, $\mathcal{S} = \varnothing$, *and* $|\mathcal{A}| = 2$). *There exists* $\pi_{\text{ref}}$ *such that for all* $T \leq \frac{1}{2}\exp(\frac{1}{8\beta})$, *with constant probability, all of the policies* $\pi^{(1)}, \ldots, \pi^{(T+1)}$ *produced by Online* DPO *satisfy*

$$\max_{\pi} J_{\beta}(\pi) - J_{\beta}(\pi^{(t)}) \geq \frac{1}{8} \quad \forall t \in [T+1].$$

That is, the sample complexity required by Online DPO is *exponential* in $\frac{1}{\beta}$, which is unacceptable in the small-$\beta$ regime; inspecting the proof, it is straightforward to see that the same conclusion holds for Iterative DPO and purely offline DPO. The idea behind Proposition 2.1 is simple: If $\pi_{\text{ref}}$ places small probability mass on the optimal action, Online DPO may fail to ever explore this action until the number of iterations is exponentially large. This reflects the intuition that in the small-$\beta$ regime, more deliberate exploration is required to discover behaviors or capabilities not already covered by $\pi_{\text{ref}}$.

**Remark 2.1.** *Various empirical works have suggested that offline* DPO *can under-perform relative to vanilla RLHF with PPO due to a lack of on-policy sampling (Xiong et al., 2023; Guo et al., 2024; Dong et al., 2024; Tang et al., 2024a). Proposition 2.1 highlights a conceptually distinct phenomenon, where both of the aforementioned algorithms (as well as online variants of* DPO*) fail due to poor coverage from* $\pi_{\text{ref}}$, *in spite of on-policy sampling.*

## 3   ONLINE EXPLORATION FOR LANGUAGE MODELS: EXPLORATORY PREFERENCE OPTIMIZATION

We now present our main algorithm XPO, which addresses the limitations of existing alignment methods by augmenting DPO with active exploration. We first describe the algorithm and motivation (Section 3.1), then present theoretical guarantees (Section 3.2), and sketch the analysis (Section 3.3).

### 3.1   THE XPO ALGORITHM

---

**Algorithm 1** Exploratory Preference Optimization (XPO)

**input:** Number of iterations $T$, KL-regularization coefficient $\beta > 0$, optimism coefficient $\alpha > 0$.
1: Initialize $\pi^{(1)} \leftarrow \pi_{\text{ref}}, \mathcal{D}_{\text{pref}}^{(0)} \leftarrow \varnothing$.
2: **for** iteration $t = 1, 2, \ldots, T$ **do**
3:    **Generate response pair** $(\tau^{(t)}, \widetilde{\tau}^{(t)})$ **via:** $s_1^{(t)} \sim \rho$, $\tau^{(t)} \sim \pi^{(t)} \mid s_1^{(t)}$, and $\widetilde{\tau}^{(t)} \sim \pi_{\text{ref}} \mid s_1^{(t)}$.
4:    **Label with preference:** Label $(\tau^{(t)}, \widetilde{\tau}^{(t)})$ as $(\tau_+^{(t)}, \tau_-^{(t)})$ with preference $y^{(t)} \sim \mathbb{P}(\tau^{(t)} \succ \widetilde{\tau}^{(t)})$.
5:    **Update preference data:** $\mathcal{D}_{\text{pref}}^{(t)} \leftarrow \mathcal{D}_{\text{pref}}^{(t-1)} \bigcup \{(\tau_+^{(t)}, \tau_-^{(t)})\}$.
6:    **Direct preference optimization with global optimism:** Calculate $\pi^{(t+1)}$ via

$$\pi^{(t+1)} \leftarrow \operatorname*{argmin}_{\pi \in \Pi} \left\{ \alpha \sum_{i=1}^{t} \log \pi(\widetilde{\tau}^{(i)}) - \sum_{(\tau_+, \tau_-) \in \mathcal{D}_{\text{pref}}^{(t)}} \log \left[ \sigma \left( \beta \log \frac{\pi(\tau_+)}{\pi_{\text{ref}}(\tau_+)} - \beta \log \frac{\pi(\tau_-)}{\pi_{\text{ref}}(\tau_-)} \right) \right] \right\}.$$

7: **return:** $\widehat{\pi} = \operatorname*{argmax}_{\pi \in \{\pi^{(1)}, \ldots, \pi^{(T+1)}\}} J_{\beta}(\pi^{(t)})$.        `// Can compute using validation data.`

---

XPO (Exploratory Preference Optimization) is displayed in Algorithm 1. The algorithm takes as input a user-specified policy class $\Pi$ and proceeds in almost the same fashion as Online DPO. For each step $t \in [T]$, given the current policy $\pi^{(t)}$ and an initial state $s_1^{(t)}$, the algorithm begins by sampling a pair of trajectories $\tau^{(t)} \sim \pi^{(t)} \mid s_1^{(t)}$ and $\widetilde{\tau}^{(t)} \sim \pi_{\text{ref}} \mid s_1^{(t)}$, which are labeled as $(\tau_+^{(t)}, \tau_-^{(t)})$ based on the preference feedback and used to update the preference dataset via $\mathcal{D}_{\text{pref}}^{(t+1)} \leftarrow \mathcal{D}_{\text{pref}}^{(t)} \cup \{(\tau_+^{(t)}, \tau_-^{(t)})\}$. The most important step is Line 6, which updates the policy to $\pi^{(t+1)}$ via the following *optimistic* variant of the DPO objective:

$$\pi^{(t+1)} \leftarrow \operatorname*{argmin}_{\pi \in \Pi} \left\{ \alpha \sum_{i=1}^{t} \log \pi(\widetilde{\tau}^{(i)}) - \sum_{(\tau_+, \tau_-) \in \mathcal{D}_{\text{pref}}^{(t)}} \log \left[ \sigma \left( \beta \log \frac{\pi(\tau_+)}{\pi_{\text{ref}}(\tau_+)} - \beta \log \frac{\pi(\tau_-)}{\pi_{\text{ref}}(\tau_-)} \right) \right] \right\}. \quad (4)$$

Here, $\alpha \geq 0$ is an *optimism parameter*; for $\alpha = 0$, the algorithm nearly equivalent to Online DPO, except that we sample $\tau^{(t)} \sim \pi^{(t)} \mid s_1^{(t)}$ and $\widetilde{\tau}^{(t)} \sim \pi_{\text{ref}} \mid s_1^{(t)}$ instead of sampling $(\tau^{(t)}, \widetilde{\tau}^{(t)}) \sim \pi^{(t)} \mid s_1^{(t)}$

at each iteration. As we will see now, for $\alpha > 0$, the term

$$\alpha \sum_{i=1}^{t} \log \pi(\widetilde{\tau}^{(i)}) \tag{5}$$

in Eq. (4) encourages the policy to behave *optimistically*, and produce diverse responses $\tau$.

**Motivation.** *Optimism in the face of uncertainty* is a widely used technique in reinforcement learning theory (Agarwal et al., 2019; Lattimore and Szepesvári, 2020; Foster and Rakhlin, 2023). In its most standard form, the optimism principle is usually stated as follows: *One should explore by choosing their actions according to the most optimistic view of the world, given all of the data that has already been observed.* The idea is that if we choose a decision according to this principle, one of two good things can happen: (i) the optimistic view is correct, and we receive large reward; or (ii) the optimistic view is incorrect, but we receive useful information that will help to better estimate the state of the world in subsequent iterations.

Optimism is typically implemented by directly estimating rewards, and it is not obvious at first glance why Eq. (5) can even be interpreted as a form of optimism. To understand, this consider a log-linear policy $\pi_f(a_h \mid s_h) = \pi_{\mathsf{ref}}(a_h \mid s_h) \exp\left(\frac{f(s_h, a_h) - V_f(s_h)}{\beta}\right)$, where $V_f(s_h) := \beta \log \sum_{a_h \in \mathcal{A}} \pi_{\mathsf{ref}}(a_h \mid s_h) e^{f(s_h, a_h)/\beta}$. Define $[\mathcal{T}_\beta f](s_h, a_h) := r(s_h, a_h) + \mathbb{E}\left[V_f(s_{h+1}) \mid s_h, a_h\right]$ as the KL-regularized Bellman operator (Ziebart et al., 2008; Ziebart, 2010). We observe, generalizing Watson et al. (2023); Rafailov et al. (2024), that for any DCMDP, for all trajectories $\tau = (s_1, a_1), \ldots, (s_H, a_H)$,

$$\beta \log \frac{\pi_f(\tau)}{\pi_{\mathsf{ref}}(\tau)} = r(\tau) - V_f(s_1) + \sum_{h=1}^{H} \left(f(s_h, a_h) - [\mathcal{T}_\beta f](s_h, a_h)\right). \tag{6}$$

That is, the policy can be viewed as maintaining an internal model for the trajectory reward, up to (i) a constant offset $V_f(s_1)$ that depends only on $s_1$; and (ii) the sum of *Bellman errors* $(f(s_h, a_h) - [\mathcal{T}_\beta f](s_h, a_h))$. The optimal KL-regularized policy $\pi_\beta^\star = \operatorname{argmax}_\pi J_\beta(\pi)$ satisfies $\pi_\beta^\star = \pi_{Q_\beta^\star}$, where $Q_\beta^\star / V_\beta^\star$ denote KL-regularized value functions (see Appendix C.4 for formal definitions and details), and has zero Bellman error ($Q_\beta^\star = [\mathcal{T}_\beta Q_\beta^\star]$), so that

$$\beta \log \frac{\pi_\beta^\star(\tau)}{\pi_{\mathsf{ref}}(\tau)} = r(\tau) - V_\beta^\star(s_1) \quad \forall \tau. \tag{7}$$

In other words, $\pi_\beta^\star$ implements an accurate internal reward model. From this viewpoint:

1. The standard DPO term in Eq. (4) encourages the policy $\pi$ to build an accurate internal model for rewards under the Bradley-Terry model; this can be viewed as a form of *implicit $Q^\star$-approximation*, since we are implicitly minimizing the Bellman errors in Eq. (6).

2. In light of Eq. (7) it is natural to approximate $V_\beta^\pi(s_1)$, the regularized value function for $\pi$, by $r(\tau) - \beta \log \frac{\pi(\tau)}{\pi_{\mathsf{ref}}(\tau)}$. Using this approximation, the first term in Eq. (4) biases the policy toward a large value function such that $V_\beta^\star \lesssim V_\beta^\pi$, implementing *implicit (global) optimism* in the face of uncertainty (up to an inconsequential difference in on-policy rewards). The fact that this suffices to drive exploration is quite subtle, and leverages non-trivial properties of the KL-regularized MDP, including the fact that Eq. (6) holds on a *per-trajectory* basis.

**On the sampling policy.** As remarked above, another difference between XPO and online/iterative DPO is that instead of sampling the preference pairs via $(\tau^{(t)}, \widetilde{\tau}^{(t)}) \sim \pi^{(t)}$, we sample $\tau^{(t)} \sim \pi^{(t)} \mid s_1^{(t)}$ and $\widetilde{\tau}^{(t)} \sim \pi_{\mathsf{ref}} \mid s_1^{(t)}$. This small change is important: it is possible to show that in general, sampling $(\tau^{(t)}, \widetilde{\tau}^{(t)}) \sim \pi^{(t)}$ can lead to degenerate behavior in which the algorithm fails to adequately explore in the small-$\beta$ regime, even when $\pi_{\mathsf{ref}}$ itself has good coverage.

While we use $\widetilde{\tau}^{(t)} \sim \pi_{\mathsf{ref}} \mid s_1^{(t)}$ in Algorithm 1, XPO is significantly more general, and leads to provable guarantees for any fixed sampling policy $\widetilde{\tau}^{(t)} \sim \widetilde{\pi} \mid s_1^{(t)}$, as well as certain data-dependent sampling schemes (e.g., sampling $\widetilde{\tau}^{(t)} \sim \mathsf{unif}(\pi^{(1)}, \ldots, \pi^{(t)}) \mid s_1^{(t)}$); different choices may have different tradeoffs and benefits in practice. A general version of XPO which leaves the sampling distribution for $\widetilde{\tau}^{(t)}$ as a free parameter is given in Appendix C.1 (Algorithm 2).

**Simplicity.** While the focus of this paper is purely theoretical, we emphasize that XPO is highly practical, and can easily be incorporated into existing language modeling and RLHF pipelines as a drop-in replacement for Online DPO (a one-line change to existing code). The theoretical guarantees for the algorithm continue to hold under standard modifications such as (i) incorporating additional preference data from $\pi_{\text{ref}}$ or another reference policy; and (ii) performing a smaller number of iterations, but collecting a larger batch of preference data from $\pi^{(t)}$ (as in Iterative DPO).

## 3.2 Theoretical Guarantees

To provide sample complexity guarantees for XPO, we make some standard statistical assumptions. The first asserts that the policy class $\Pi$ is powerful enough to represent the optimal KL-regularized policy.

**Assumption 3.1** (Policy realizability). *The policy class $\Pi$ satisfies $\pi_\beta^\star \in \Pi$.*

Policy realizability is a minimal assumption for sample-efficient reinforcement learning (Agarwal et al., 2019; Lattimore and Szepesvári, 2020; Foster and Rakhlin, 2023); through Eq. (7), it is equivalent to a form of reward/value realizability. For language modeling, $\Pi$ will typically correspond to a class of language models with fixed architecture but variable weights. Next, we make a regularity assumption on the policies in $\Pi$ (Rosset et al., 2024).

**Assumption 3.2** (Bounded density ratios). *For all $\pi \in \Pi$ and trajectories $\tau$,*

$$\left| \log\left( \frac{\pi(\tau)}{\pi_{\text{ref}}(\tau)} \right) \right| \leq \frac{V_{\text{max}}}{\beta}. \tag{8}$$

$V_{\text{max}}$ is measurable and controllable in practice; our guarantees scale polynomially with this parameter. For log-linear policies where $\pi(a \mid s) \propto \exp(f(s,a)/\beta)$, we expect $V_{\text{max}} \lesssim R_{\text{max}}$.

To quantify the rate at which the algorithm converges to an optimal policy, we require an *exploration condition*, which limits the amount of times the algorithm can be surprised by substantially new state distributions; such assumptions are necessary for reinforcement learning with general function approximation (Jiang et al., 2017; Jin et al., 2021; Xie et al., 2023). Our main result is stated in terms of a condition known as *coverability* (Xie et al., 2023), but more general guarantees are given in Appendix C. Define $d^\pi(\tau) := \mathbb{P}_\pi((s_1, a_1), \ldots, (s_H, a_H) = \tau)$.

**Definition 3.1** (Coverability). *The trajectory-level coverability coefficient is given by*

$$C_{\text{cov}}(\Pi) := \inf_{\mu \in \Delta((\mathcal{S} \times \mathcal{A})^H)} \sup_{\tau \in (\mathcal{S} \times \mathcal{A})^H} \sup_{\pi \in \Pi} \frac{d^\pi(\tau)}{\mu(\tau)}. \tag{9}$$

Assumption 3.2 implies a trivial bound of $C_{\text{cov}}(\Pi) \lesssim \exp\left(\frac{V_{\text{max}}}{\beta}\right)$. Indeed, $C_{\text{cov}}(\Pi)$ measures coverage with respect to the best possible distribution $\mu$, while the bound implied by Assumption 3.2 takes $\mu = \pi_{\text{ref}}$, so we expect $C_{\text{cov}}(\Pi) \ll \exp(V_{\text{max}}/\beta)$ when $\pi_{\text{ref}}$ does not provide adequate coverage on its own (e.g., the example in Proposition 2.1). This is precisely the setting where we expect deliberate exploration to be helpful. We also note that there is a trivial bound $C_{\text{cov}}(\Pi) \leq |\mathcal{A}|^H$, but because coverability depends on the structure of the (restricted) class $\Pi$, the value can be significantly smaller in general (e.g., if policies $\pi \in \Pi$ are highly correlated or stochastic).

The main sample complexity guarantee for XPO is as follows.

**Theorem 3.1** (Sample complexity bound for XPO). *Suppose that Assumptions 3.1 and 3.2 hold. For any $\beta > 0$ and $T \in \mathbb{N}$, if we set $\alpha = c \cdot \frac{\beta}{(V_{\text{max}} + R_{\text{max}})e^{2R_{\text{max}}}} \cdot \sqrt{\frac{\log(|\Pi|T\delta^{-1})}{T \cdot C_{\text{cov}}(\Pi)}}$ for an absolute constant $c > 0$, then Algorithm 1 ensures that with probability at least $1 - \delta$,*[3]

$$J_\beta(\pi_\beta^\star) - J_\beta(\widehat{\pi}) \lesssim (V_{\text{max}} + R_{\text{max}})e^{2R_{\text{max}}} \cdot \sqrt{\frac{C_{\text{cov}}(\Pi) \log(|\Pi|T\delta^{-1}) \log^2(T)}{T}}.$$

---

[3]Exponential dependence on the reward range $R_{\text{max}}$ is an intrinsic feature of the Bradley-Terry model, and can be found in all prior sample complexity guarantees for this framework, offline and online (Das et al., 2024; Rosset et al., 2024); this exponential dependence is also a focal point of the closely related literature on logistic bandits (Faury et al., 2020; Abeille et al., 2021).

Let us discuss some key features of this result.

**Statistical efficiency.** Theorem 3.1 shows that XPO converges to a near-optimal policy with sample complexity polynomial in the coverability coefficient $C_{\mathsf{cov}}(\Pi)$; in particular, to learn an $\varepsilon$-optimal policy $T = \widetilde{O}\left(\frac{C_{\mathsf{cov}}(\Pi)\log|\Pi|}{\varepsilon^2}\right)$ episodes are required.[4] By scaling with $C_{\mathsf{cov}}(\Pi)$, Theorem 3.1 can be viewed as a strict improvement over offline RLHF (Zhu et al., 2023; Zhan et al., 2023a), as well as prior works on online RLHF that rely on passive exploration (Xiong et al., 2023; Gao et al., 2024; Chang et al., 2024). In particular, these works scale with *coverage parameters* for $\pi_{\mathsf{ref}}$, the simplest of which take the form $C_{\mathsf{conc}}(\Pi) := \sup_{\tau \in (\mathcal{S} \times \mathcal{A})^H} \sup_{\pi \in \Pi} \frac{\pi(\tau)}{\pi_{\mathsf{ref}}(\tau)}$. Under Assumption 3.2, we have that $C_{\mathsf{conc}}(\Pi) = \exp(V_{\mathsf{max}}/\beta)$ which, as discussed above, upper bounds $C_{\mathsf{cov}}(\Pi)$ but can be much larger when $\pi_{\mathsf{ref}}$ has poor coverage. The dependence on $C_{\mathsf{cov}}(\Pi)$ in Theorem 3.1 reflects the fact that XPO can explore responses not covered by $\pi_{\mathsf{ref}}$.[5]

In Appendix C, we give a generalization of Theorem 3.1 (Theorem 3.1′) which scales with a more comprehensive exploration parameter, the *Sequential Extrapolation Coefficient* (SEC), matching (for DCMDPs) the most general results in prior work on exploration in RLHF, but with a significantly simpler algorithm (Chen et al., 2022; Wang et al., 2023; Ye et al., 2024). The SEC also leads to polynomial sample complexity for tabular and linear MDPs, a common setting considered in prior work (Xu et al., 2020; Novoseller et al., 2020; Pacchiano et al., 2021; Wu and Sun, 2023; Zhan et al., 2023b; Das et al., 2024). See Appendix A for a detailed comparison. We emphasize that Theorem 3.1 applies to any DCMDP (including but not limited to the token-level MDP), even if the dynamics are unknown; as such, the result meaningfully extends beyond the *contextual bandit* formulation of RLHF found in many prior works (Zhu et al., 2023; Xiong et al., 2023; Das et al., 2024; Ye et al., 2024).

**Remark 3.1** (Nontriviality and role of $\beta$)**.** *By avoiding explicit dependence on $\exp(\frac{1}{\beta})$,* XPO *provably improves upon Online* DPO *when $\beta$ is small; per Proposition 2.1, the latter must pay $\exp(\frac{1}{\beta})$ even when $C_{\mathsf{cov}}(\Pi) \leq 2$. This improvement stems from the fact that KL-regularization does not automatically lead to exploration or grant meaningful control of coverability in the small-$\beta$ regime.*

*To highlight the importance of the small-$\beta$ regime, we note that by taking $\beta = \mathrm{poly}(1/T)$, Theorem 3.1 immediately leads to bounds on the* unregularized *reward $J(\pi)$. This would not be possible if the sample complexity guarantee explicitly scaled with $\exp(\frac{1}{\beta})$.*

**Computational efficiency.** Most prior approaches to RL with general function approximation that incorporate global forms of optimism similar to Eq. (5) (Jiang et al., 2017; Sun et al., 2019; Du et al., 2021; Jin et al., 2021; Xie et al., 2023; Liu et al., 2024a) are known to be computationally intractable to implement in general (Dann et al., 2018), and involve solving non-convex, non-differentiable constrained optimization problems. Thus, it is natural to ask why our result is not too good to be true. The answer is that even though the objective in Eq. (4) is simple, it is still non-convex in general, even if one employs log-linear policies of the form $\pi_\theta(a \mid s) \propto \exp\left(\frac{1}{\beta}\langle \phi(s,a), \theta \rangle\right)$ for $\theta \in \mathbb{R}^d$. This non-convexity is precisely caused by the presence of the optimistic term Eq. (5); Theorem 3.1 is valid for all choices of $\beta > 0$, but we expect that the optimization problem in Eq. (4) will become more difficult to solve as $\beta \to 0$.[6] In light of this, our work can be viewed as using the unique structure of the KL-regularized MDP formulation and deterministic contextual MDP (DCMDP) to derive an optimistic exploration objective which—while still non-convex—is differentiable and directly amenable to implementation with language models. This technique is novel even in the context of reward-driven (as opposed to preference-based) RL, and we expect it to find broader use.

**Additional remarks.** Separately, we mention in passing that we believe it should be possible to derive tighter sample complexity bounds for large $\beta > 0$, in the vein of Tiapkin et al. (2023a).

---

[4]We state the result for finite classes to simplify presentation, following the standard in RL theory (Agarwal et al., 2019; Foster and Rakhlin, 2023)

[5]Many works consider more general notions of coverage that account for reward function structure, in the same vein as SEC, as well as single-policy variants; both can be problematic for similar reasons.

[6]Interestingly, one can show that for an appropriate $\alpha$, our objective converges to the standard global optimism objective (Jin et al., 2021) under this parameterization as $\beta \to 0$. Conversely for very large $\beta$ ($\beta \gtrsim R_{\mathsf{max}}$), the objective becomes convex. We leave a dedicated analysis of the optimization landscape for future work.

**Remark 3.2** (Limitations of the DPO objective). *Our results are limited to MDPs with deterministic dynamics and stochastic start state (DCMDPs). We believe that without further modifications, the DPO objective is not suitable for stochastic dynamics, as Eq. (7) no longer holds on a per-trajectory basis.*

**Remark 3.3** (Trajectory coverability). *A related point concerns trajectory coverability. In the standard (as opposed to preference-based) RL setting, it is possible to achieve guarantees that scale with* state-action coverability *(Xie et al., 2023), defined via* $C_{\mathsf{st}}(\Pi) := \inf_{\mu \in \Delta(\mathcal{S} \times \mathcal{A})} \sup_{s \in \mathcal{S}, a \in \mathcal{A}} \sup_{\pi \in \Pi} \frac{d^\pi(s,a)}{\mu(s,a)}$, *where* $d^\pi(s,a) := \mathbb{P}_\pi(s_h = s, a_h = a)$. *In general, we can have* $C_{\mathsf{st}}(\Pi) \ll C_{\mathsf{cov}}(\Pi)$. *We expect that trajectory-level coverability is necessary for algorithms based on the DPO objective. Nonetheless, the difference is immaterial for language modeling in the token-level MDP, which has* $C_{\mathsf{st}}(\Pi) = C_{\mathsf{cov}}(\Pi)$.

### 3.3 PROOF SKETCH FOR THEOREM 3.1

Our starting point for the proof of Theorem 3.1 is the following regret decomposition, which is proven as a consequence of the implicit $Q^\star$-approximation result in Eq. (7).

**Lemma 3.1** (Central regret decomposition). *For any pair of policies $\pi$ and $\nu$, it holds that*

$$J_\beta(\pi_\beta^\star) - J_\beta(\pi) = \mathbb{E}_{\tau \sim \nu}\left[\beta \log \pi(\tau)\right] - \mathbb{E}_{\tau \sim \nu}\left[\beta \log \pi_\beta^\star(\tau)\right] \tag{10}$$

$$+ \mathbb{E}_{\tau \sim \pi}\left[\beta \log \frac{\pi(\tau)}{\pi_{\mathsf{ref}}(\tau)} - r(\tau)\right] - \mathbb{E}_{\tau \sim \nu}\left[\beta \log \frac{\pi(\tau)}{\pi_{\mathsf{ref}}(\tau)} - r(\tau)\right]. \tag{11}$$

This result decomposes the error of any policy into two pairs of terms: The first pair in Eq. (10) measures the extent to which the policy's internal reward model overestimates the optimal value, and directly informs the notion of optimism in XPO, while the second pair in Eq. (11) measures the reward model's predictive accuracy. Critically, as a consequence of the fact that Eq. (7) holds uniformly for all trajectories, the regret decomposition measures error under (i) the policy $\pi$ itself (on-policy error), and (ii) an *arbitrary* reference policy $\nu$, which we will instantiate as the historical data distribution.

Let $\boldsymbol{\mu}^{(t)} := \frac{1}{t-1}\sum_{i<t} \pi^{(i)} \otimes \pi_{\mathsf{ref}}$ denote the policy that, given $s_1$, samples $\tau \sim \pi^{(i)}$ for $i \sim \mathsf{unif}([t-1])$ and samples $\widetilde{\tau} \sim \pi_{\mathsf{ref}}$, with the convention that $\boldsymbol{\mu}^{(1)}$ is arbitrary. Observe that $\min_{t \in [T+1]} J_\beta(\pi_\beta^\star) - J_\beta(\pi^{(t)}) \le \frac{1}{T}\sum_{t=1}^T J_\beta(\pi_\beta^\star) - J_\beta(\pi^{(t)})$. For each step $t$, applying Lemma 3.1 with $\pi = \pi^{(t)}$ and $\nu = \pi_{\mathsf{ref}}$ gives

$$\frac{1}{T}\sum_{t=1}^T J_\beta(\pi_\beta^\star) - J_\beta(\pi^{(t)}) \le \frac{1}{T}\sum_{t=1}^T \mathbb{E}_{\tau \sim \pi_{\mathsf{ref}}}\left[\beta \log \pi^{(t)}(\tau) - \beta \log \pi_\beta^\star(\tau)\right]$$

$$+ \frac{1}{T}\sum_{t=1}^T \mathbb{E}_{s_1 \sim \rho, \tau \sim \pi^{(t)}|s_1, \widetilde{\tau} \sim \pi_{\mathsf{ref}}|s_1}\left[\beta \log \frac{\pi^{(t)}(\tau)}{\pi_{\mathsf{ref}}(\tau)} - r(\tau) - \beta \log \frac{\pi^{(t)}(\widetilde{\tau})}{\pi_{\mathsf{ref}}(\widetilde{\tau})} + r(\widetilde{\tau})\right]. \tag{12}$$

The reward estimation error term in Eq. (12) samples $\tau \sim \pi^{(t)} \mid s_1$ and $\widetilde{\tau} \sim \pi_{\mathsf{ref}} \sim s_1$ (on-policy). To relate this to the purely off-policy objective in Line 6 of XPO, we use a potential argument based on coverability (Xie et al., 2023) which, for any $\alpha > 0$, allows us to bound the above expression by

$$\lesssim \frac{\alpha}{\beta} \cdot C_{\mathsf{cov}}(\Pi) + \frac{1}{T}\sum_{t=1}^T \mathbb{E}_{\tau \sim \pi_{\mathsf{ref}}}\left[\beta \log \pi^{(t)}(\tau) - \beta \log \pi_\beta^\star(\tau)\right]$$

$$+ \frac{\alpha^{-1}\beta}{T}\sum_{t=1}^T \mathbb{E}_{s_1 \sim \rho, (\tau, \widetilde{\tau}) \sim \boldsymbol{\mu}^{(t)}|s_1}\left[\left(\beta \log \frac{\pi^{(t)}(\tau)}{\pi_{\mathsf{ref}}(\tau)} - r(\tau) - \beta \log \frac{\pi^{(t)}(\widetilde{\tau})}{\pi_{\mathsf{ref}}(\widetilde{\tau})} + r(\widetilde{\tau})\right)^2\right]. \tag{13}$$

Let $\Psi_{\mathsf{XPO}}^{(t)}(\pi) := \mathbb{E}_{\tau \sim \pi_{\mathsf{ref}}}\left[\beta \log \pi(\tau) - \beta \log \pi_\beta^\star(\tau)\right] + \alpha^{-1}\beta \mathbb{E}_{s_1 \sim \rho, (\tau, \widetilde{\tau}) \sim \boldsymbol{\mu}^{(t)}|s_1}\left[\left(\beta \log \frac{\pi(\tau)}{\pi_{\mathsf{ref}}(\tau)} - r(\tau) - \beta \log \frac{\pi(\widetilde{\tau})}{\pi_{\mathsf{ref}}(\widetilde{\tau})} + r(\widetilde{\tau})\right)^2\right]$. If we could choose $\pi^{(t)} = \operatorname{argmin}_{\pi \in \Pi} \Psi_{\mathsf{XPO}}^{(t)}(\pi)$, we would be done, since by Eq. (7) this would yield

$$\Psi_{\mathsf{XPO}}^{(t)}(\pi^{(t)}) \le \Psi_{\mathsf{XPO}}^{(t)}(\pi_\beta^\star) = \mathbb{E}_{s_1 \sim \rho, (\tau, \widetilde{\tau}) \sim \boldsymbol{\mu}^{(t)}|s_1}\left[\left(\beta \log \frac{\pi_\beta^\star(\tau)}{\pi_{\mathsf{ref}}(\tau)} - r(\tau) - \beta \log \frac{\pi_\beta^\star(\widetilde{\tau})}{\pi_{\mathsf{ref}}(\widetilde{\tau})} + r(\widetilde{\tau})\right)^2\right] = 0.$$

The XPO objective in Line 6 minimizes an empirical analogue of this quantity (up to a standard translation between log-loss and square loss under the Bradley-Terry model), so a concentration argument (Lemma C.5) allows us to conclude that the iterates of XPO satisfy $\Psi_{\mathsf{XPO}}^{(t)}(\pi^{(t)}) \lesssim \alpha^{-1}\frac{\log|\Pi|}{t} + \sqrt{\frac{\log|\Pi|}{t}}$.

Plugging this bound into Eq. (13) yields $\frac{1}{T}\sum_{t=1}^T J_\beta(\pi_\beta^\star) - J_\beta(\pi^{(t)}) \lesssim \sqrt{\frac{C_{\mathsf{cov}}(\Pi) \log|\Pi|}{T}}$ after tuning $\alpha$.

## 4    DISCUSSION

Our work provides the first simple, yet provably sample-efficient online exploration algorithm for RLHF with general function approximation, a step toward fully realizing the potential of online exploration for aligning language models. Our results also show that viewing DPO as a form of implicit $Q^\star$-approximation can directly inform new algorithmic interventions (e.g., implicit optimism), and offer an example of fruitful interplay between language modeling and theoretical reinforcement learning. Building on this viewpoint, an exciting direction for future work is to import the broader set of tools from the literature on reinforcement learning theory (e.g., more powerful exploration principles (Foster et al., 2021)) and harness them for language modeling and alignment; in this context, we expect our analysis techniques based on the KL-regularized MDP to find broader use.

From a reinforcement learning perspective, interesting technical directions for future work include (i) providing instance-dependent sample complexity bounds for XPO; and (ii) supporting RL settings beyond deterministic contextual MDPs. On the practical side, immediate followup directions include extending XPO to support general preference models (Munos et al., 2023; Swamy et al., 2024) or more general feedback modalities (Ethayarajh et al., 2024).

## ACKNOWLEDGEMENTS

AR acknowledges support from the ARO through award W911NF-21-1-0328, as well as from the Simons Foundation and the NSF through awards DMS-2031883 and PHY-2019786.

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

# Contents of Appendix

## A  RELATED WORK

**Theoretical algorithms for RLHF.**  Theoretical analysis of algorithms for RLHF is becoming an active area of research. Much of this research focuses on purely offline RLHF (Zhu et al., 2023; Zhan et al., 2023a), which is complementary to our work. Many works also consider a so-called *hybrid* RLHF setting, where the algorithm has access to online feedback, but requires the initial policy $\pi_{\mathrm{ref}}$ to have good coverage (e.g., bounded concentrability or related quantities) (Xiong et al., 2023; Gao et al., 2024; Chang et al., 2024).[7] These hybrid algorithms do not engage in systematic exploration (i.e., they explore passively), and hence cannot provide meaningful guarantees if $\pi_{\mathrm{ref}}$ does not adequately cover the optimal policy (e.g., for the setting in Proposition 2.1).

For online RLHF, the most relevant related work can be summarized as follows:

- Most prior work (Xu et al., 2020; Novoseller et al., 2020; Pacchiano et al., 2021; Wu and Sun, 2023; Zhan et al., 2023b; Du et al., 2024; Das et al., 2024) gives algorithms and sample complexity guarantees for the special case of tabular or linear MDPs; these algorithms use exploration bonuses that are tailored to linear models, and are not suitable for the general function approximation setting we consider (e.g., for LLMs). Nonetheless, we obtain polynomial sample complexity guarantees for tabular and linear MDPs (Examples C.1 and C.2), though our results are restricted to deterministic dynamics (we believe that moving beyond the DPO objective is likely required to handle stochastic dynamics).

- More relevant to our work is Ye et al. (2024), who give algorithms and sample complexity guarantees for online RLHF with general function approximation for the special case of contextual bandits ($H = 1$). For contextual bandits, their sample complexity guarantees scale with a complexity measure, the *eluder coefficient*, which is equivalent to the Sequential Extrapolation Coefficient in our most general result, Theorem 3.1′. However, their exploration algorithm requires solving a rather complicated optimization problem, and it is unclear whether it is possible to implement it faithfully for language models (in particular, their experiments use an alternative, heuristic approach to exploration which is only loosely inspired by the theory).

- Lastly, Chen et al. (2022); Wang et al. (2023) give guarantees for RLHF with general function approximation based on eluder dimension-like complexity measures which are incomparable to, but

---

[7]To our knowledge, all prior works in this space require uniform notions of concentrability as opposed to single-policy concentrability. Gao et al. (2024) state guarantees in terms of single-policy concentrability under the assumption that certain regression errors can be bounded, but this cannot be achieved in general without further coverage or exploration-like conditions.

in some cases more general than Theorem 3.1′. However, these works require model-based function approximation (as opposed to the model-free setup we consider), and do not lead to efficient or practical algorithms when specialized to language modeling.

A difference worth highlighting between our work and some (but not all) of the works above (Zhu et al., 2023; Xiong et al., 2023; Das et al., 2024; Ye et al., 2024) is that we model RLHF as a general reinforcement learning problem as opposed to a contextual bandit problem. The problem of autoregressive sequence prediction can equivalently be formulated as RL in the token-level MDP, or as a contextual bandit problem (RL with horizon $H = 1$) in which the "action space" consists of all possible token sequences. However, because our work supports general deterministic contextual MDPs (DCMDPs) with unknown dynamics and not just the token-level MDP, it is strictly more general than the contextual bandit formulation.

Recent work of Rafailov et al. (2024) (see also Nachum et al. (2017); Garg et al. (2021); Watson et al. (2023); Zhong et al. (2024)) shows that DPO, when applied to the token-level MDP can be viewed as estimating the KL-regularized value function $Q_\beta^\star$; their work does not consider sample complexity or online exploration. Our results extend their observation to any deterministic contextual MDP and—more importantly—show that it is possible to harness this perspective to provide provable end-to-end sample complexity guarantees.

**Empirical algorithms for RLHF.** Our work uses DPO (Rafailov et al., 2023) as a starting point. Many algorithms prior works have built upon DPO with the aim of addressing specific shortcomings Liu et al. (2023); Tang et al. (2024b); Azar et al. (2024); Rosset et al. (2024); Chen et al. (2024); Wu et al. (2024); Tajwar et al. (2024), but which are largely orthogonal to exploration.

Online exploration in RLHF has received limited exploration so far, with notable examples including Online DPO (Guo et al., 2024) and Iterative DPO (Xu et al., 2023; Tran et al., 2024; Pang et al., 2024; Mitra et al., 2024; Dong et al., 2024). As discussed in Section 2, these methods engage in purely *passive* exploration, meaning that sample from the current model $\pi^{(t)}$ without an explicit mechanism to encourage diverse, exploratory responses.

Dwaracherla et al. (2024) perform a dedicated empirical evaluation of active exploration for language models. However, this work does not actually train the language model, and thus cannot be viewed as a form of RLHF; instead the authors train a reward model iteratively, and use this in tandem with various active sampling schemes to accept or reject responses proposed by $\pi_{\text{ref}}$. Nevertheless, the positive results achieved by Dwaracherla et al. (2024) in this limited setting are suggestive of the potential power of online exploration in RLHF. Similarly, Ye et al. (2024) perform a limited evaluation of empirical exploration schemes inspired by theoretical RL, but only report results for reward modeling benchmarks, not language modeling.

Most closely related, Xiong et al. (2023); Dong et al. (2024) perform an extensive empirical evaluation of Iterative DPO variants, and find that Iterative DPO with passive exploration can already have significant benefits over offline DPO. These works also incorporate a "best/worst-over-$n$" trick for preference pair construction, which can be viewed as a heuristic to promote exploration, but does not have provable guarantees.

**Theoretical reinforcement learning.** Outside the context of language models, an active line of research provides structural complexity measures and algorithms that enable sample-efficient exploration in reinforcement learning in general settings (Russo and Van Roy, 2013; Jiang et al., 2017; Sun et al., 2019; Wang et al., 2020; Du et al., 2021; Jin et al., 2021; Foster et al., 2021; Xie et al., 2023; Foster et al., 2023; Liu et al., 2024a). The techniques from this line of research that support general function approximation, while sample-efficient, are computationally intractable to implement in general (Dann et al., 2018), involving non-convex and non-differentiable constrained optimization problems. We use the unique structure of the KL-regularized MDP formulation and deterministic contextual MDP (DCMDP) to derive the exploration objective in XPO which—while still non-convex—is differentiable and directly amenable to implementation with language models.

**Entropy- and KL-regularized reinforcement learning.** First introduced in Ziebart et al. (2008); Ziebart (2010), a number of recent works provide sample complexity guarantees for reinforcement learning in KL-regularized or entropy-regularized MDPs (Kozuno et al., 2022; Tiapkin et al., 2023b;a), mainly focusing on the special case of tabular (finite-state/action) MDPs. To the best of our knowledge, the optimistic objective in XPO is novel in this context.

# B    TECHNICAL TOOLS

**Lemma B.1** (Azuma-Hoeffding). *Let $(X_t)_{t \leq T}$ be a sequence of real-valued random variables adapted to a filtration $(\mathscr{F}_t)_{t \leq T}$. If $|X_t| \leq R$ almost surely, then with probability at least $1 - \delta$,*

$$\left| \sum_{t=1}^{T} X_t - \mathbb{E}_{t-1}[X_t] \right| \leq R \cdot \sqrt{8T \log(2\delta^{-1})}.$$

**Lemma B.2** (Martingale Chernoff (e.g., Foster et al., 2021)). *For any sequence of real-valued random variables $(X_t)_{t \leq T}$ adapted to a filtration $(\mathscr{F}_t)_{t \leq T}$, it holds that with probability at least $1 - \delta$, for all $T' \leq T$,*

$$\sum_{t=1}^{T'} -\log\big(\mathbb{E}_{t-1}\big[e^{-X_t}\big]\big) \leq \sum_{t=1}^{T'} X_t + \log(\delta^{-1}). \tag{14}$$

# C    PROOF OF THEOREM 3.1

This section is organized as follows. First, in Appendix C.2, we present a more general version of XPO, which makes use of an arbitrary, user-specified sampling policy for the second response $\widetilde{\tau}$. Then, in Appendix C.2, we state a more general version of Theorem 3.1 (Theorem 3.1′), and show how it implies Theorem 3.1. Examples are then given in Appendix C.3.

In the remainder of the section, we prove Theorem 3.1′. We first prove a number of intermediate results:

- In Appendix C.4, we state preliminaries regarding the KL-regularized MDP, and use them to prove the implicit $Q^\star$-approximation lemma (Lemma C.3).

- In Appendix C.5, we prove the central regret decomposition lemma (Lemma 3.1).

- In Appendix C.6, we prove a key concentration result used within Theorem 3.1′.

Finally, in Appendix C.7, we prove Theorem 3.1′, with proofs for supporting lemmas deferred to Appendix C.8.

## C.1    GENERAL VERSION OF XPO

---

**Algorithm 2** Exploratory Preference Optimization (XPO) with general sampling policy.

---

**input:** Number of iterations $T$, KL-regularization coefficient $\beta > 0$, optimism coefficient $\alpha > 0$, sampling strategy $\boldsymbol{\pi}_{\mathsf{samp}}$.

1: Initialize $\pi^{(1)}, \widetilde{\pi}^{(1)} \leftarrow \pi_{\mathsf{ref}}, \mathcal{D}_{\mathsf{pref}}^{(0)} \leftarrow \varnothing$.

2: **for** iteration $t = 1, 2, \ldots, T$ **do**

3:    **Generate response pair** $(\tau^{(t)}, \widetilde{\tau}^{(t)})$ **via:** $s_1^{(t)} \sim \rho$, $\tau^{(t)} \sim \pi^{(t)} \mid s_1^{(t)}$, and $\widetilde{\tau}^{(t)} \sim \widetilde{\pi}^{(t)} \mid s_1^{(t)}$.

4:    **Label with preference:** Label $(\tau^{(t)}, \widetilde{\tau}^{(t)})$ as $(\tau_+^{(t)}, \tau_-^{(t)})$ with preference $y^{(t)} \sim \mathbb{P}(\tau^{(t)} \succ \widetilde{\tau}^{(t)})$.

5:    **Update preference data:** $\mathcal{D}_{\mathsf{pref}}^{(t)} \leftarrow \mathcal{D}_{\mathsf{pref}}^{(t-1)} \bigcup \{(\tau_+^{(t)}, \tau_-^{(t)})\}$.

6:    **Update optimism data:** Compute dataset $\mathcal{D}_{\mathsf{opt}}^{(t)}$ of $t$ samples from $\widetilde{\pi}^{(t)}$.

   // When $\widetilde{\pi}^{(t)} = \pi_{\mathsf{ref}}$, can re-use previous samples as in Algorithm 1.

7:    **Direct preference optimization with global optimism:** Calculate $\pi^{(t+1)}$ via

$$\pi^{(t+1)} \leftarrow \operatorname*{argmin}_{\pi \in \Pi} \left\{ \alpha \sum_{\tau \in \mathcal{D}_{\mathsf{opt}}^{(t)}} \log \pi(\tau) - \sum_{(\tau_+, \tau_-) \in \mathcal{D}_{\mathsf{pref}}^{(t)}} \log \left[ \sigma \left( \beta \log \frac{\pi(\tau_+)}{\pi_{\mathsf{ref}}(\tau_+)} - \beta \log \frac{\pi(\tau_-)}{\pi_{\mathsf{ref}}(\tau_-)} \right) \right] \right\}.$$

8:    **Update sampling policy:** $\widetilde{\pi}^{(t+1)} \leftarrow \boldsymbol{\pi}_{\mathsf{samp}}(\pi^{(1)}, \ldots, \pi^{(t+1)})$.

9: **return:** $\widehat{\pi} = \operatorname*{argmax}_{\pi \in \{\pi^{(1)}, \ldots, \pi^{(T+1)}\}} J_\beta(\pi^{(t)})$.    // Can compute using validation data.

---

Algorithm 2 presents a general version of XPO. The algorithm is identical to Algorithm 1, except that it makes use of an arbitrary, user-specified user-specified sampling policy for the second response $\widetilde{\tau}$.

In more detail, the algorithm takes as input a *sampling strategy* $\boldsymbol{\pi}_{\mathsf{samp}}$ which, at step $t$, computes a *sampling policy* $\widetilde{\pi}^{(t)}$ via $\widetilde{\pi}^{(t)} \leftarrow \boldsymbol{\pi}_{\mathsf{samp}}(\pi^{(1)}, \ldots, \pi^{(T)})$. The algorithm then samples the response pair

$(\tau^{(t)}, \widetilde{\tau}^{(t)})$ via $\tau^{(t)} \sim \pi^{(t)} \mid s_1^{(t)}$ and $\widetilde{\tau}^{(t)} \sim \widetilde{\pi}^{(t)} \mid s_1^{(t)}$. Algorithm 1 is a special case of this scheme in which $\widetilde{\pi}^{(t)} = \pi_{\mathsf{ref}}$ for all $t$.

A secondary difference from Algorithm 1 is that Algorithm 2 assumes access to a dataset $\mathcal{D}_{\mathsf{opt}}^{(t)}$ consisting of $t$ responses sampled from $\widetilde{\pi}^{(t)}$, which are used to compute the optimistic term in Line 7. In Algorithm 1, because $\widetilde{\pi} = \pi_{\mathsf{ref}}$ is static, we can simply re-use the responses $\widetilde{\tau}^{(1)}, \ldots, \widetilde{\tau}^{(t)}$ for this task, setting $\mathcal{D}_{\mathsf{opt}}^{(t)} = \{\widetilde{\tau}^{(1)}, \ldots, \widetilde{\tau}^{(t)}\}$. However, for general time-varying sampling scheme, it may be necessary to draw a fresh dataset of responses from $\widetilde{\pi}^{(t)}$ to compute $\mathcal{D}_{\mathsf{opt}}^{(t)}$.

As a practical example, Algorithm 3—displayed below—instantiates the general scheme in Algorithm 2 by setting $\widetilde{\pi}^{(t)} = \mathsf{unif}(\pi^{(1)}, \ldots, \pi^{(t)})$ to sample from the historical data distribution at step $t$. For this scheme, it suffices to set $\mathcal{D}_{\mathsf{opt}}^{(t)} = \{\tau^{(1)}, \ldots, \tau^{(t)}\}$, re-using the responses sampled from $\pi^{(1)}, \ldots, \pi^{(t)}$.

---

**Algorithm 3** Exploratory Preference Optimization (XPO) with historical sampling.

    **input:** Number of iterations $T$, KL-regularization coefficient $\beta > 0$, optimism coefficient $\alpha > 0$, sampling strategy $\boldsymbol{\pi}_{\mathsf{samp}}$.

1: Initialize $\pi^{(1)}, \widetilde{\pi}^{(1)} \leftarrow \pi_{\mathsf{ref}}, \mathcal{D}_{\mathsf{pref}}^{(0)} \leftarrow \varnothing$.

2: **for** iteration $t = 1, 2, \ldots, T$ **do**

3:     **Generate response pair** $(\tau^{(t)}, \widetilde{\tau}^{(t)})$ **via:** $s_1^{(t)} \sim \rho$, $\tau^{(t)} \sim \pi^{(t)} \mid s_1^{(t)}$, and $\widetilde{\tau}^{(t)} \sim \mathsf{unif}(\pi^{(1)}, \ldots, \pi^{(t)}) \mid s_1^{(t)}$.

4:     **Label with preference:** Label $(\tau^{(t)}, \widetilde{\tau}^{(t)})$ as $(\tau_+^{(t)}, \tau_-^{(t)})$ with preference $y^{(t)} \sim \mathbb{P}(\tau^{(t)} \succ \widetilde{\tau}^{(t)})$.

5:     **Update preference data:** $\mathcal{D}_{\mathsf{pref}}^{(t)} \leftarrow \mathcal{D}_{\mathsf{pref}}^{(t-1)} \bigcup \{(\tau_+^{(t)}, \tau_-^{(t)})\}$.

6:     **Update optimism data:** Compute dataset $\mathcal{D}_{\mathsf{opt}}^{(t)}$ of $t$ samples from $\widetilde{\pi}^{(t)}$.
                                      `// When `$\widetilde{\pi}^{(t)} = \pi_{\mathsf{ref}}$`, can re-use previous samples as in Algorithm 1.`

7:     **Direct preference optimization with global optimism:** Calculate $\pi^{(t+1)}$ via

$$\pi^{(t+1)} \leftarrow \underset{\pi \in \Pi}{\mathrm{argmin}} \left\{ \alpha \sum_{i=1}^{t} \log \pi(\tau^{(i)}) - \sum_{(\tau_+, \tau_-) \in \mathcal{D}_{\mathsf{pref}}^{(t)}} \log \left[ \sigma \left( \beta \log \frac{\pi(\tau_+)}{\pi_{\mathsf{ref}}(\tau_+)} - \beta \log \frac{\pi(\tau_-)}{\pi_{\mathsf{ref}}(\tau_-)} \right) \right] \right\}.$$

8: **return:** $\widehat{\pi} = \mathrm{argmax}_{\pi \in \{\pi^{(1)}, \ldots, \pi^{(T+1)}\}} J_\beta(\pi^{(t)})$.         `// Can compute using validation data.`

---

### C.2   General Version of Theorem 3.1

Our most general sample complexity guarantee for XPO (Algorithm 1 and Algorithm 2), Theorem 3.1′, is stated in terms of the following preference-based analogue of the *Sequential Extrapolation Coefficient* (SEC) from Xie et al. (2023) (also known as an eluder coefficient or decoupling coefficient (Zhong et al., 2022; Ye et al., 2024)). Recall that for a trajectory $\tau = (s_1, a_1), \ldots, (s_H, a_H)$, we define

$$\pi(\tau) = \prod_{h=1}^{H} \pi(a_h \mid s_h), \quad \text{and} \quad r(\tau) = \sum_{h=1}^{H} r(s_h, a_h). \tag{15}$$

For a pair of policies $\pi$ and $\widetilde{\pi}$, we define $\pi \otimes \widetilde{\pi}$ as the joint policy that, given $s_1$, samples $\tau \sim \pi \mid s_1$ and $\widetilde{\tau} \sim \widetilde{\pi} \mid s_1$. We write $(\tau, \widetilde{\tau}) \sim \pi \otimes \widetilde{\pi} \mid s_1$ as shorthand for this process.

**Definition C.1** (Sequential Extrapolation Coefficient). *For a policy class $\Pi$, sampling strategy $\boldsymbol{\pi}_{\mathsf{samp}}$, and entropy regularization parameter $\beta > 0$, we define the Sequential Extrapolation Coefficient via*

$$\mathsf{SEC}_{\mathsf{RLHF}}(\Pi, T, \beta; \boldsymbol{\pi}_{\mathsf{samp}}) \tag{16}$$

$$= \sup_{\pi^{(1)}, \ldots, \pi^{(T)} \in \Pi} \left\{ \sum_{t=1}^{T} \frac{\left( \mathbb{E}_{s_1 \sim \rho, \tau \sim \pi^{(t)} \mid s_1, \widetilde{\tau} \sim \widetilde{\pi}^{(t-1)} \mid s_1} \left[ \beta \log \frac{\pi^{(t)}(\tau)}{\pi_{\mathsf{ref}}(\tau)} - r(\tau) - \beta \log \frac{\pi^{(t)}(\widetilde{\tau})}{\pi_{\mathsf{ref}}(\widetilde{\tau})} + r(\widetilde{\tau}) \right] \right)^2}{V_{\mathsf{max}}^2 \vee (t-1) \cdot \mathbb{E}_{s_1 \sim \rho, (\tau, \widetilde{\tau}) \sim \boldsymbol{\mu}^{(t)} \mid s_1} \left[ \left( \beta \log \frac{\pi^{(t)}(\tau)}{\pi_{\mathsf{ref}}(\tau)} - r(\tau) - \beta \log \frac{\pi^{(t)}(\widetilde{\tau})}{\pi_{\mathsf{ref}}(\widetilde{\tau})} + r(\widetilde{\tau}) \right)^2 \right]} \right\},$$

*where $\widetilde{\pi}^{(t)} = \boldsymbol{\pi}_{\mathsf{samp}}(\pi^{(1)}, \ldots, \pi^{(t)})$, and where we define $\boldsymbol{\mu}^{(t)} := \frac{1}{t-1} \sum_{i<t} \pi^{(i)} \otimes \widetilde{\pi}^{(i)}$, with the convention that $\mu^{(1)}$ is arbitrary.*

Note that for Algorithm 1, which sets $\widetilde{\pi}^{(t)} = \pi_{\mathsf{ref}}$ for all $t$, we can simplify the definition above to

$$\mathsf{SEC}_{\mathsf{RLHF}}(\Pi, T, \beta; \pi_{\mathsf{ref}}) \tag{17}$$

$$:= \sup_{\pi^{(1)},\ldots,\pi^{(T)} \in \Pi} \left\{ \sum_{t=1}^{T} \frac{\left( \mathbb{E}_{s_1 \sim \rho, \tau \sim \pi^{(t)} | s_1, \widetilde{\tau} \sim \pi_{\mathsf{ref}} | s_1} \left[ \beta \log \frac{\pi^{(t)}(\tau)}{\pi_{\mathsf{ref}}(\tau)} - r(\tau) - \beta \log \frac{\pi^{(t)}(\widetilde{\tau})}{\pi_{\mathsf{ref}}(\widetilde{\tau})} + r(\widetilde{\tau}) \right] \right)^2}{V_{\mathsf{max}}^2 \vee (t-1) \cdot \mathbb{E}_{s_1 \sim \rho, (\tau, \widetilde{\tau}) \sim \boldsymbol{\mu}^{(t)} | s_1} \left[ \left( \beta \log \frac{\pi^{(t)}(\tau)}{\pi_{\mathsf{ref}}(\tau)} - r(\tau) - \beta \log \frac{\pi^{(t)}(\widetilde{\tau})}{\pi_{\mathsf{ref}}(\widetilde{\tau})} + r(\widetilde{\tau}) \right)^2 \right]} \right\},$$

where $\boldsymbol{\mu}^{(t)} := \frac{1}{t-1} \sum_{i < t} \pi^{(i)} \otimes \pi_{\mathsf{ref}}$.

**Main sample complexity guarantee.** Our general sample complexity guarantee is as follows.

**Theorem 3.1′** (General version of Theorem 3.1)**.** *Suppose Assumptions 3.1 and 3.2 hold. For any $\beta > 0$ and $T \in \mathbb{N}$, if we set $\alpha = c \cdot \frac{\beta}{(V_{\mathsf{max}} + R_{\mathsf{max}})e^{2R_{\mathsf{max}}}} \cdot \sqrt{\frac{\log(|\Pi|T\delta^{-1})\log(T)}{T \cdot \mathsf{SEC}_{\mathsf{RLHF}}(\Pi, T, \beta; \boldsymbol{\pi}_{\mathsf{samp}})}}$ for an absolute constant $c > 0$, then Algorithm 2 ensures that with probability at least $1 - \delta$,*

$$J_\beta(\pi_\beta^\star) - J_\beta(\widehat{\pi}) \lesssim (V_{\mathsf{max}} + R_{\mathsf{max}})e^{2R_{\mathsf{max}}} \cdot \sqrt{\frac{\mathsf{SEC}_{\mathsf{RLHF}}(\Pi, T, \beta; \boldsymbol{\pi}_{\mathsf{samp}})\log(|\Pi|T\delta^{-1})\log(T)}{T}}.$$

*As a special case, if we set $\alpha = c \cdot \frac{\beta}{(V_{\mathsf{max}} + R_{\mathsf{max}})e^{2R_{\mathsf{max}}}} \cdot \sqrt{\frac{\log(|\Pi|T\delta^{-1})\log(T)}{T \cdot \mathsf{SEC}_{\mathsf{RLHF}}(\Pi, T, \beta; \pi_{\mathsf{ref}})}}$ for an absolute constant $c > 0$, then Algorithm 1 ensures that with probability at least $1 - \delta$,*

$$J_\beta(\pi_\beta^\star) - J_\beta(\widehat{\pi}) \lesssim (V_{\mathsf{max}} + R_{\mathsf{max}})e^{2R_{\mathsf{max}}} \cdot \sqrt{\frac{\mathsf{SEC}_{\mathsf{RLHF}}(\Pi, T, \beta; \pi_{\mathsf{ref}})\log(|\Pi|T\delta^{-1})\log(T)}{T}}.$$

The following result shows that the SEC is always bounded by the coverability coefficient in Definition 3.1.

**Lemma C.1.** *Suppose that $\boldsymbol{\pi}_{\mathsf{samp}}$ sets $\widetilde{\pi}^{(t)} = \widetilde{\pi}$ for an arbitrary fixed policy $\widetilde{\pi}$ (e.g., $\widetilde{\pi} = \pi_{\mathsf{ref}}$). Then for any policy class $\Pi$ and $\beta > 0$, it holds that for all $T \in \mathbb{N}$,*

$$\mathsf{SEC}_{\mathsf{RLHF}}(\Pi, T, \beta; \boldsymbol{\pi}_{\mathsf{samp}}) \leq O(C_{\mathsf{cov}}(\Pi) \cdot \log(T)). \tag{18}$$

Theorem 3.1 follows immediately by combining Theorem 3.1′ with Lemma C.1.

### C.3 ADDITIONAL EXAMPLES FOR THEOREM 3.1′

In this section, we apply Theorem 3.1′ and bound the SEC for *log-linear* policy classes. For $f : \mathcal{S} \times \mathcal{A} \to \mathbb{R}$, define

$$\pi_f(a \mid s) = \pi_{\mathsf{ref}}(a \mid s)e^{\frac{f(s,a) - V_f(s)}{\beta}}, \quad \text{where} \quad V_f(s) = \beta \log \left( \sum_{a \in \mathcal{A}} \pi_{\mathsf{ref}}(a \mid s)e^{\frac{f(s,a)}{\beta}} \right).$$

We consider policy classes of the form

$$\Pi_\mathcal{F} := \{ \pi_f \mid f \in \mathcal{F} \}$$

for a given value function class $\mathcal{F} \subseteq (\mathcal{S} \times \mathcal{A} \to R_{\mathsf{max}})$. Note that for such a class, we can take $V_{\mathsf{max}} \leq R_{\mathsf{max}}$, and that $Q_\beta^\star \in \mathcal{F}$ implies that $\pi_\beta^\star \in \Pi_\mathcal{F}$.

The following lemma bounds the SEC for log-linear policy classes in terms of a preference-based analogue of the value function SEC in Xie et al. (2023).

**Lemma C.2** (SEC for log-linear policies)**.** *For any value function class $\mathcal{F} \subseteq (\mathcal{S} \times \mathcal{A} \to R_{\mathsf{max}})$, we have that $\mathsf{SEC}_{\mathsf{RLHF}}(\Pi, T, \beta; \boldsymbol{\pi}_{\mathsf{samp}}) \leq \mathsf{SEC}_{\mathsf{RLHF}}(\mathcal{F}, T; \boldsymbol{\pi}_{\mathsf{samp}})$, where*

$$\mathsf{SEC}_{\mathsf{RLHF}}(\mathcal{F}, T; \boldsymbol{\pi}_{\mathsf{samp}}) := \sup_{f^{(1)},\ldots,f^{(T)} \in \mathcal{F}}$$

$$\left\{ \sum_{t=1}^{T} \frac{\left( \mathbb{E}_{s_1 \sim \rho, \tau \sim \pi^{(t)} | s_1, \widetilde{\tau} \sim \widetilde{\pi}^{(t-1)} | s_1} \left[ \sum_{h=1}^{H} (f^{(t)}(s_h, a_h) - [\mathcal{T}_\beta f^{(t)}](s_h, a_h)) - (f^{(t)}(\widetilde{s}_h, \widetilde{a}_h) - [\mathcal{T}_\beta f^{(t)}](\widetilde{s}_h, \widetilde{a}_h)) \right] \right)^2}{R_{\mathsf{max}}^2 \vee (t-1) \cdot \mathbb{E}_{s_1 \sim \rho, (\tau, \widetilde{\tau}) \sim \boldsymbol{\mu}^{(t)} | s_1} \left[ \left( \sum_{h=1}^{H} (f^{(t)}(s_h, a_h) - [\mathcal{T}_\beta f^{(t)}](s_h, a_h)) - (f^{(t)}(\widetilde{s}_h, \widetilde{a}_h) - [\mathcal{T}_\beta f^{(t)}](\widetilde{s}_h, \widetilde{a}_h)) \right)^2 \right]} \right\},$$

*where $\pi^{(t)} := \pi_{f^{(t)}}$, $\widetilde{\pi}^{(t)} = \boldsymbol{\pi}_{\mathsf{samp}}(\pi^{(1)}, \ldots, \pi^{(t)})$, and $\boldsymbol{\mu}^{(t)} := \frac{1}{t-1} \sum_{i < t} \pi^{(i)} \otimes \widetilde{\pi}^{(i)}$ (with the convention that $\mu^{(1)}$ is arbitrary), and where $\mathcal{T}_\beta$ is the KL-regularized Bellman operator defined in Appendix C.4.*

**Proof of Lemma C.2.** This is an immediate corollary of Lemma C.4. □

We first apply this bound to give a polynomial bound on the SEC in tabular DCMDPs where $\mathcal{S}$ and $\mathcal{A}$ are finite.

**Example C.1** (Tabular MDP). Suppose that $\pi_{\mathsf{samp}}$ sets $\pi^{(t)} = \widetilde{\pi}$ for all $t$ for some fixed policy $\widetilde{\pi}$. When $\mathcal{F} = \{f : \mathcal{S} \times \mathcal{A} \to R_{\mathsf{max}}\}$ consists of all functions over tabular state and action spaces with $|\mathcal{S}|, |\mathcal{A}| < \infty$, we have $\mathsf{SEC}_{\mathsf{RLHF}}(\mathcal{F}, T; \pi_{\mathsf{samp}}) \leq \widetilde{O}(H|\mathcal{S}||\mathcal{A}|)$ and $\log|\Pi_{\mathcal{F}}| \lesssim \widetilde{O}(|\mathcal{S}||\mathcal{A}|)$. It follows that XPO (Algorithm 1) achieves

$$J_\beta(\pi_\beta^\star) - J_\beta(\widehat{\pi}) \lesssim \widetilde{O}\left(R_{\mathsf{max}} e^{2R_{\mathsf{max}}} \sqrt{\frac{H|\mathcal{S}|^2|\mathcal{A}|^2}{T}}\right).$$

◁

Example C.1 is a corollary of the following more general result.

**Example C.2** (Linear MDP). In a Linear MDP (Jin et al., 2020), we have

$$P(s' \mid s, a) = \langle \phi(s, a), \mu(s') \rangle, \tag{19}$$

and

$$r(s, a) = \langle \phi(s, a), \vartheta \rangle, \tag{20}$$

where $\phi(s, a) \in \mathbb{R}^d$ is a known feature map with $\|\phi(s, a)\| \leq 1$, $\mu(s') \in \mathbb{R}^d$ is an unknown feature map with $\|\sum_{s'} \mu(s')\| \leq \sqrt{d}$, and $\varphi \in \mathbb{R}^d$ is an unknown parameter with $\|\varphi\| \leq 1$. Here, the optimal KL-regularized value function $Q_\beta^\star$ (cf. Appendix C.4) is linear with respect to the feature map $\phi(s, a)$. In particular, if we take

$$\mathcal{F} := \big\{ f(s, a) = \langle \phi(s, a), \theta \rangle \mid \theta \in \mathbb{R}^d, \|\theta\| \leq B, |f(s, a)| \leq R \big\}$$

for $B = O(\sqrt{d})$ and $R = O(R_{\mathsf{max}})$, then $\pi_\beta^\star \in \Pi_{\mathcal{F}}$, satisfying Assumption 3.1. For this setting, when $\pi_{\mathsf{samp}}$ sets $\pi^{(t)} = \widetilde{\pi}$ for all $t$ for some fixed policy $\widetilde{\pi}$, we have $\mathsf{SEC}_{\mathsf{RLHF}}(\mathcal{F}, T; \pi_{\mathsf{samp}}) \leq \widetilde{O}(dH)$ and $\log|\Pi_{\mathcal{F}}| \lesssim \widetilde{O}(d)$. It follows that XPO (Algorithm 1) achieves

$$J_\beta(\pi_\beta^\star) - J_\beta(\widehat{\pi}) \lesssim \widetilde{O}\left(R_{\mathsf{max}} e^{2R_{\mathsf{max}}} \sqrt{\frac{Hd^2}{T}}\right).$$

◁

### C.4 KL-Regularized MDP Preliminaries and $Q^\star$-Approximation

In this section, we give some basic background on value functions and dynamic programming for the KL-regularized MDP (Ziebart et al., 2008; Ziebart, 2010), then use these properties to prove Lemmas C.3 and C.4, which show that the optimal KL-regularized policy implicitly performs models rewards and performs $Q^\star$-approximation.

**Dynamic programming and value functions for KL-regularized MDP.** First, for any function $f : \mathcal{S} \times \mathcal{A} \to \mathbb{R}$, define

$$V_f(s_h) := \beta \log \sum_{a_h \in \mathcal{A}} \pi_{\mathsf{ref}}(a_h \mid s_h) e^{f(s_h, a_h)/\beta} \quad \forall s \in \mathcal{S}_h.$$

It is straightforward to verify that

$$V_f(s_h) = \max_{\pi : \mathcal{S} \to \Delta(\mathcal{A})} \left( \mathbb{E}_{a_h \sim \pi(\cdot|s_h)} \left[ f(s_h, a_h) - \beta \log \frac{\pi(a_h \mid s_h)}{\pi_{\mathsf{ref}}(a_h \mid s_h)} \right] \right), \tag{21}$$

and that the policy that obtains the maximum above is

$$\pi_f(a_h \mid s_h) = \pi_{\mathsf{ref}}(a_h \mid s_h) e^{(f(s_h, a_h) - V_f(s_h))/\beta}. \tag{22}$$

From here, beginning with $Q_\beta^\star(s_H, a_H) := r(s_H, a_H)$, $\pi_\beta^\star(a_H \mid s_H) = \pi_{Q_\beta^\star}(a_H \mid s_H)$, and $V_\beta^\star(s_H) = V_{Q_\beta^\star}(s_H)$ for $s_H \in \mathcal{S}_H$, for each $s_h \in \mathcal{S}_h$, we can inductively define for each $h \in [H]$:

$$
\begin{aligned}
Q_\beta^\star(s_h, a_h) &:= r(s_h, a_h) + \mathbb{E}\big[V_\beta^\star(s_{h+1}) \mid s_h, a_h\big], \\
\pi_\beta^\star(a_h \mid s_h) &:= \pi_{Q_\beta^\star}(a_h \mid s_h), \\
V_\beta^\star(s_h) &:= V_{Q_\beta^\star}(s_h).
\end{aligned}
\tag{23}
$$

In light of Eq. (21), it is clear that $\pi_\beta^\star \in \mathrm{argmax}_{\pi:\mathcal{S}\to\Delta(\mathcal{A})} J_\beta(\pi)$. In addition, if we define the KL-regularized Bellman operator as

$$
[\mathcal{T}_\beta f](s_h, a_h) := r(s_h, a_h) + \mathbb{E}_{s_{h+1}\sim P(\cdot|s_h, a_h)}\left[V_f(s_{h+1})\right],
$$

we have that

$$
Q_\beta^\star(s_h, a_h) = \big[\mathcal{T}_\beta Q_\beta^\star\big](s_h, a_h).
$$

**Implicit $Q^\star$-approximation.** The next lemma, following Watson et al. (2023); Rafailov et al. (2024), shows that the optimal KL-regularized policy $\pi_\beta^\star$ can be viewed as implicitly modeling rewards.

**Lemma C.3** (Implicit $Q^\star$-Approximation). *For any DCMDP, it holds that for all admissible[8] trajectories $\tau = (s_1, a_1), \ldots, (s_H, a_H)$,*

$$
\beta \log \frac{\pi_\beta^\star(\tau)}{\pi_{\mathrm{ref}}(\tau)} = r(\tau) - V_\beta^\star(s_1),
\tag{24}
$$

*where $V_\beta^\star$ is the KL-regularized value function defined in Eq. (23).*

**Proof of Lemma C.3.** Let $\tau = (s_1, a_1), \ldots, (s_H, a_H)$, and recall that for any DCMDP, all state transitions except for $s_1 \sim \rho$ are deterministic. Then we have

$$
\begin{aligned}
0 &= \sum_{h=1}^H \big(Q_\beta^\star(s_h, a_h) - \big[\mathcal{T}_\beta Q_\beta^\star\big](s_h, a_h)\big) \\
&= \sum_{h=1}^H \big(Q_\beta^\star(s_h, a_h) - r(s_h, a_h) - V_\beta^\star(s_{h+1})\big) \\
&= \sum_{h=1}^H \left(V_\beta^\star(s_h) + \beta \log \frac{\pi_\beta^\star(a_h \mid s_h)}{\pi_{\mathrm{ref}}(a_h \mid s_h)} - r(s_h, a_h) - V_\beta^\star(s_{h+1})\right) \\
&= V_\beta^\star(s_1) + \sum_{h=1}^H \left(\beta \log \frac{\pi_\beta^\star(a_h \mid s_h)}{\pi_{\mathrm{ref}}(a_h \mid s_h)} - r(s_h, a_h)\right),
\end{aligned}
$$

where the second equality uses that $(\mathcal{T}_\beta f)(s_h, a_h) = r(s_h, a_h) + V_f(s_{h+1})$ for any admissible trajectory in a deterministic MDP, and the third equality uses the explicit form for $\pi_\beta^\star$ in terms of $V_\beta^\star$ and $Q_\beta^\star$ given in Eq. (22). Rearranging yields the result.

$\square$

We can also prove the following, more general version of Lemma C.4.

**Lemma C.4** (Implicit $Q^\star$-Approximation (general version)). *For any DCMDP, it holds that for any function $f : \mathcal{S} \times \mathcal{A} \to \mathbb{R}$ and all admissible trajectories $\tau = (s_1, a_1), \ldots, (s_H, a_H)$,*

$$
\beta \log \frac{\pi_f(\tau)}{\pi_{\mathrm{ref}}(\tau)} = r(\tau) - V_f(s_1) + \sum_{h=1}^H \left(f(s_h, a_h) - [\mathcal{T}_\beta f](s_h, a_h)\right).
\tag{25}
$$

---

[8]We use "admissible" to a refer to a trajectory generated by executing an arbitrary policy $\pi : \mathcal{S} \to \Delta(\mathcal{A})$ in the MDP.

**Proof of Lemma C.4.** Let $\tau = (s_1, a_1), \ldots, (s_H, a_H)$. Then we have

$$\sum_{h=1}^{H} \left( f(s_h, a_h) - [\mathcal{T}_\beta f](s_h, a_h) \right)$$

$$= \sum_{h=1}^{H} \left( f(s_h, a_h) - r(s_h, a_h) - V_f(s_{h+1}) \right)$$

$$= \sum_{h=1}^{H} \left( V_f(s_h) + \beta \log \frac{\pi_f(a_h \mid s_h)}{\pi_{\mathsf{ref}}(a_h \mid s_h)} - r(s_h, a_h) - V_f(s_{h+1}) \right)$$

$$= V_f(s_1) + \sum_{h=1}^{H} \left( \beta \log \frac{\pi_f(a_h \mid s_h)}{\pi_{\mathsf{ref}}(a_h \mid s_h)} - r(s_h, a_h) \right),$$

where the first equality uses the definition of $V_f$, the second equality uses that $(\mathcal{T}_\beta f)(s_h, a_h) = r(s_h, a_h) + V_f(s_{h+1})$ for any admissible trajectory in a deterministic MDP, and the third equality uses that $\pi_f(a \mid s) = \pi_{\mathsf{ref}}(a \mid s) e^{\frac{f(s,a) - V_f(s)}{\beta}}$. Rearranging yields the result. $\qquad\square$

### C.5 REGRET DECOMPOSITION

In this section we prove the central regret decomposition for XPO, restated below.

**Lemma 3.1** (Central regret decomposition). *For any pair of policies $\pi$ and $\nu$, it holds that*

$$J_\beta(\pi_\beta^\star) - J_\beta(\pi) = \mathbb{E}_{\tau \sim \nu} \left[ \beta \log \pi(\tau) \right] - \mathbb{E}_{\tau \sim \nu} \left[ \beta \log \pi_\beta^\star(\tau) \right] \tag{10}$$

$$+ \mathbb{E}_{\tau \sim \pi} \left[ \beta \log \frac{\pi(\tau)}{\pi_{\mathsf{ref}}(\tau)} - r(\tau) \right] - \mathbb{E}_{\tau \sim \nu} \left[ \beta \log \frac{\pi(\tau)}{\pi_{\mathsf{ref}}(\tau)} - r(\tau) \right]. \tag{11}$$

**Proof of Lemma 3.1.** It follows immediately from the definition of the KL-regularized reward that

$$J_\beta(\pi_\beta^\star) - J_\beta(\pi) = \mathbb{E}_\pi \left[ \beta \log \frac{\pi(\tau)}{\pi_{\mathsf{ref}}(\tau)} - r(\tau) \right] - \mathbb{E}_{\pi_\beta^\star} \left[ \beta \log \frac{\pi_\beta^\star(\tau)}{\pi_{\mathsf{ref}}(\tau)} - r(\tau) \right].$$

However, since $\beta \log \frac{\pi_\beta^\star(\tau)}{\pi_{\mathsf{ref}}(\tau)} - r(\tau) = V_\beta^\star(s_1)$ for all admissible trajectories by Lemma C.3, we have that

$$\mathbb{E}_{\pi_\beta^\star} \left[ \beta \log \frac{\pi_\beta^\star(\tau)}{\pi_{\mathsf{ref}}(\tau)} - r(\tau) \right] = \mathbb{E}_\nu \left[ \beta \log \frac{\pi_\beta^\star(\tau)}{\pi_{\mathsf{ref}}(\tau)} - r(\tau) \right]$$

for all policies $\nu$, as the initial state $s_1$ does not depend on the policy under consideration. The result now follows by rearranging

$$\mathbb{E}_\pi \left[ \beta \log \frac{\pi(\tau)}{\pi_{\mathsf{ref}}(\tau)} - r(\tau) \right] - \mathbb{E}_\nu \left[ \beta \log \frac{\pi_\beta^\star(\tau)}{\pi_{\mathsf{ref}}(\tau)} - r(\tau) \right]$$

$$= \mathbb{E}_\nu \left[ \beta \log \pi(\tau) \right] - \mathbb{E}_\nu \left[ \beta \log \pi_\beta^\star(\tau) \right] + \mathbb{E}_\pi \left[ \beta \log \frac{\pi(\tau)}{\pi_{\mathsf{ref}}(\tau)} - r(\tau) \right] - \mathbb{E}_\nu \left[ \beta \log \frac{\pi(\tau)}{\pi_{\mathsf{ref}}(\tau)} - r(\tau) \right].$$

$$\square$$

### C.6 CONCENTRATION LEMMAS

Recall that we define $\boldsymbol{\mu}^{(t)} = \frac{1}{t-1} \sum_{i<t} \pi^{(i)} \otimes \widetilde{\pi}^{(i)}$. For a given policy $\pi$, define

$$f_\pi(\tau, \widetilde{\tau}) = \beta \log \frac{\pi(\tau)}{\pi_{\mathsf{ref}}(\tau)} - \beta \log \frac{\pi(\widetilde{\tau})}{\pi_{\mathsf{ref}}(\widetilde{\tau})}.$$

The following lemma is our central concentration guarantee for Algorithm 1.

**Lemma C.5** (Concentration for XPO). *Suppose that Assumptions 3.1 and 3.2 hold. Then Algorithm 1 guarantees that with probability at least $1 - \delta$, for all steps $t \in [T]$,*

$$\alpha \cdot \mathbb{E}_{s_1 \sim \rho, \tau \sim \widetilde{\pi}^{(t-1)}} \left[ \log(\pi^{(t)}(\tau)) - \log(\pi_\beta^\star(\tau)) \right] + \kappa \cdot \mathbb{E}_{s_1 \sim \rho, (\tau, \widetilde{\tau}) \sim \boldsymbol{\mu}^{(t)}|s_1} \left[ \left( f_{\pi^{(t)}}(\tau, \widetilde{\tau}) - f_{\pi_\beta^\star}(\tau, \widetilde{\tau}) \right)^2 \right]$$

$$\leq \frac{2 \log(2|\Pi|T\delta^{-1})}{t-1} + \frac{\alpha}{\beta} V_{\max} \sqrt{\frac{2^4 \log(2|\Pi|T\delta^{-1})}{t-1}},$$

*for $\kappa := (8(R_{\max} + V_{\max})e^{2R_{\max}})^{-2}$.*

**Proof of Lemma C.5.** Let $t \in \{2, \dots, T+1\}$ be fixed.

$$\widehat{L}^{(t)}(\pi) = \sum_{i<t} -y^{(i)} \log \left[ \sigma \left( \beta \log \frac{\pi(\tau^{(i)})}{\pi_{\text{ref}}(\tau^{(i)})} - \beta \log \frac{\pi(\widetilde{\tau}^{(i)})}{\pi_{\text{ref}}(\widetilde{\tau}^{(i)})} \right) \right] \tag{26}$$

$$- (1 - y^{(i)}) \log \left[ \sigma \left( \beta \log \frac{\pi(\widetilde{\tau}^{(i)})}{\pi_{\text{ref}}(\widetilde{\tau}^{(i)})} - \beta \log \frac{\pi(\tau^{(i)})}{\pi_{\text{ref}}(\tau^{(i)})} \right) \right]$$

and $\widehat{B}^{(t)}(\pi) = \alpha \sum_{\tau \in \mathcal{D}_{\text{opt}}^{(t-1)}} \log \pi(\tau)$. Then we can equivalently write

$$\pi^{(t)} = \operatorname*{argmin}_{\pi \in \Pi} \left\{ \widehat{L}^{(t)}(\pi) + \widehat{B}^{(t)}(\pi) \right\}.$$

For a given policy $\pi$, recall that we define

$$f_\pi(\tau, \widetilde{\tau}) = \beta \log \frac{\pi(\tau)}{\pi_{\text{ref}}(\tau)} - \beta \log \frac{\pi(\widetilde{\tau})}{\pi_{\text{ref}}(\widetilde{\tau})},$$

and let

$$P_\pi(y \mid \tau, \widetilde{\tau}) = y \cdot \sigma(f_\pi(\tau, \widetilde{\tau})) + (1 - y) \cdot (1 - \sigma(f_\pi(\tau, \widetilde{\tau}))).$$

Then, in light of Lemma C.3, under the Bradley-Terry model (Eq. (1)), we have that for all $t$,

$$y^{(t)} \sim P_{\pi_\beta^\star}(\cdot \mid \tau^{(t)}, \widetilde{\tau}^{(t)}). \tag{27}$$

In addition, we can rewrite Eq. (26) as

$$\widehat{L}(\pi) = \sum_{i<t} -\log(P_\pi(y^{(t)} \mid \tau^{(t)}, \widetilde{\tau}^{(t)})).$$

Using this observation, we begin by proving an intermediate concentration result. For a pair of probability measures $\mathbb{P}$ and $\mathbb{Q}$, we define squared Hellinger distance via

$$D_{\mathsf{H}}^2(\mathbb{P}, \mathbb{Q}) = \int \left( \sqrt{d\mathbb{P}} - \sqrt{d\mathbb{Q}} \right)^2. \tag{28}$$

**Lemma C.6.** *For any fixed $t \geq 1$, with probability at least $1 - \delta$, all $\pi \in \Pi$ satisfy*

$$\sum_{i<t} \mathbb{E}_{s_1 \sim \rho, \tau \sim \pi^{(i)}|s_1, \widetilde{\tau} \sim \widetilde{\pi}^{(i)}|s_1} \left[ D_{\mathsf{H}}^2 \left( P_\pi(\cdot \mid \tau, \widetilde{\tau}), P_{\pi_\beta^\star}(\cdot \mid \tau, \widetilde{\tau}) \right) \right] \leq \widehat{L}^{(t)}(\pi) - \widehat{L}^{(t)}(\pi_\beta^\star) + 2 \log(|\Pi|\delta^{-1}).$$

Rearranging Lemma C.6, with probability at least $1 - \delta$, all $\pi \in \Pi$ satisfy

$$\widehat{B}^{(t)}(\pi) - \widehat{B}^{(t)}(\pi_\beta^\star) + \sum_{i<t} \mathbb{E}_{s_1 \sim \rho, \tau \sim \pi^{(i)}|s_1, \widetilde{\tau} \sim \widetilde{\pi}^{(i)}|s_1} \left[ D_{\mathsf{H}}^2 \left( P_\pi(\cdot \mid \tau, \widetilde{\tau}), P_{\pi_\beta^\star}(\cdot \mid \tau, \widetilde{\tau}) \right) \right]$$

$$\leq \widehat{L}^{(t)}(\pi) + \widehat{B}^{(t)}(\pi) - \widehat{L}^{(t)}(\pi_\beta^\star) - \widehat{B}^{(t)}(\pi_\beta^\star) + 2 \log(|\Pi|\delta^{-1}).$$

Hence, as long as $\pi_\beta^\star \in \Pi$ (Assumption 3.1), the definition of $\pi^{(t)}$ in Algorithm 2 implies that

$$\widehat{B}^{(t)}(\pi^{(t)}) - \widehat{B}^{(t)}(\pi_\beta^\star) + \sum_{i<t} \mathbb{E}_{s_1 \sim \rho, \tau \sim \pi^{(i)}|s_1, \widetilde{\tau} \sim \widetilde{\pi}^{(i)}|s_1} \left[ D_{\mathsf{H}}^2 \left( P_{\pi^{(t)}}(\cdot \mid \tau, \widetilde{\tau}), P_{\pi_\beta^\star}(\cdot \mid \tau, \widetilde{\tau}) \right) \right] \leq 2 \log(|\Pi|\delta^{-1}). \tag{29}$$

We next appeal to another basic concentration result.

**Lemma C.7.** *For any fixed $t \geq 1$, with probability at least $1 - \delta$, all $\pi \in \Pi$ satisfy*

$$\alpha \cdot (t-1) \cdot \mathbb{E}_{s_1 \sim \rho, \tau \sim \widetilde{\pi}^{(t-1)}|s_1} \left[ \log(\pi(\tau)) - \log(\pi_\beta^\star(\tau)) \right] \leq \widehat{B}^{(t)}(\pi) - \widehat{B}^{(t)}(\pi_\beta^\star) + \frac{\alpha}{\beta} V_{\mathsf{max}} \sqrt{2^4(t-1)\log(|\Pi|\delta^{-1})}.$$

Combining Lemma C.7 with Eq. (29), we conclude that with probability at least $1 - 2\delta$,

$$\alpha \cdot (t-1) \cdot \mathbb{E}_{s_1 \sim \rho, \tau \sim \widetilde{\pi}^{(t-1)}|s_1} \left[ \log(\pi^{(t)}(\tau)) - \log(\pi_\beta^\star(\tau)) \right]$$
$$+ \sum_{i<t} \mathbb{E}_{s_1 \sim \rho, \tau \sim \pi^{(i)}|s_1, \widetilde{\tau} \sim \widetilde{\pi}^{(i)}|s_1} \left[ D_{\mathsf{H}}^2 \left( P_{\pi^{(t)}}(\cdot \mid \tau, \widetilde{\tau}), P_{\pi_\beta^\star}(\cdot \mid \tau, \widetilde{\tau}) \right) \right]$$
$$\leq 2\log(|\Pi|\delta^{-1}) + \frac{\alpha}{\beta} V_{\mathsf{max}} \sqrt{2^6(t-1)\log(|\Pi|\delta^{-1})},$$

or equivalently,

$$\alpha \cdot \mathbb{E}_{s_1 \sim \rho, \tau \sim \widetilde{\pi}^{(t-1)}|s_1} \left[ \log(\pi^{(t)}(\tau)) - \log(\pi_\beta^\star(\tau)) \right] + \mathbb{E}_{s_1 \sim \rho, (\tau, \widetilde{\tau}) \sim \boldsymbol{\mu}^{(t)}|s_1} \left[ D_{\mathsf{H}}^2 \left( P_{\pi^{(t)}}(\cdot \mid \tau, \widetilde{\tau}), P_{\pi_\beta^\star}(\cdot \mid \tau, \widetilde{\tau}) \right) \right]$$
$$\leq \frac{2\log(|\Pi|\delta^{-1})}{t-1} + \frac{\alpha}{\beta} V_{\mathsf{max}} \sqrt{\frac{2^6 \log(|\Pi|\delta^{-1})}{t-1}}, \tag{30}$$

To conclude, we further simplify the expression via

$$\mathbb{E}_{s_1 \sim \rho, (\tau, \widetilde{\tau}) \sim \boldsymbol{\mu}^{(t)}|s_1} \left[ D_{\mathsf{H}}^2 \left( P_{\pi^{(t)}}(\cdot \mid \tau, \widetilde{\tau}), P_{\pi_\beta^\star}(\cdot \mid \tau, \widetilde{\tau}) \right) \right]$$
$$\geq \mathbb{E}_{s_1 \sim \rho, (\tau, \widetilde{\tau}) \sim \boldsymbol{\mu}^{(t)}|s_1} \left[ \left( \sqrt{\sigma(f_{\pi^{(t)}}(\tau, \widetilde{\tau}))} - \sqrt{\sigma(f_{\pi_\beta^\star}(\tau, \widetilde{\tau}))} \right)^2 \right]$$
$$\geq \frac{1}{8} \mathbb{E}_{s_1 \sim \rho, (\tau, \widetilde{\tau}) \sim \boldsymbol{\mu}^{(t)}|s_1} \left[ \left( \sigma(f_{\pi^{(t)}}(\tau, \widetilde{\tau})) - \sigma(f_{\pi_\beta^\star}(\tau, \widetilde{\tau})) \right)^2 \right],$$

where the last inequality uses that for $x, y \geq 0$, $(x-y)^2 \leq 4(x+y)(\sqrt{x}-\sqrt{y})^2$.

Finally, using Lemma C.3, we have $f_{\pi_\beta^\star} \in [-R_{\mathsf{max}}, R_{\mathsf{max}}]$ almost surely, while $f_{\pi^{(t)}} \in [-V_{\mathsf{max}}, V_{\mathsf{max}}]$ by Assumption 3.2. We appeal to the following lemma.

**Lemma C.8** (e.g., Rosset et al. (2024)). *If $x \in [-X, X]$ and $y \in [-Y, Y]$ for $X \geq 0$, $Y \geq 1$, then*

$$|x - y| \leq 8(X+Y)e^{2Y}|\sigma(x) - \sigma(y)|.$$

From this, we conclude that

$$\mathbb{E}_{s_1 \sim \rho, (\tau, \widetilde{\tau}) \sim \boldsymbol{\mu}^{(t)}|s_1} \left[ \left( \sigma(f_{\pi^{(t)}}(\tau, \widetilde{\tau})) - \sigma(f_{\pi_\beta^\star}(\tau, \widetilde{\tau})) \right)^2 \right]$$
$$\geq (8(R_{\mathsf{max}} + V_{\mathsf{max}})e^{2R_{\mathsf{max}}})^{-2} \cdot \mathbb{E}_{s_1 \sim \rho, (\tau, \widetilde{\tau}) \sim \boldsymbol{\mu}^{(t)}|s_1} \left[ \left( f_{\pi^{(t)}}(\tau, \widetilde{\tau}) - f_{\pi_\beta^\star}(\tau, \widetilde{\tau}) \right)^2 \right]$$

This proves the result after taking a union bound over all steps $t$.

$\square$

### C.6.1 Proofs for Supporting Lemmas

**Proof of Lemma C.6.** To begin, define

$$\ell^{(i)}(\pi) = -\log(P_\pi(y^{(t)} \mid \tau^{(t)}, \widetilde{\tau}^{(t)})).$$

For a fixed policy $\pi \in \Pi$, define $Z^{(i)}(\pi) = \frac{1}{2}(\ell^{(i)}(\pi) - \ell^{(i)}(\pi_\beta^\star))$. Define a filtration $\mathscr{F}^{(t)} = \sigma((\tau^{(1)}, \widetilde{\tau}^{(1)}), \dots, (\tau^{(t-1)}, \widetilde{\tau}^{(t-1)}))$. Applying Lemma B.2 with the sequence $(Z_i(\pi))$ and taking a union bound over $\pi \in \Pi$, have that with probability at least $1 - \delta$, all $\pi \in \Pi$ satisfy

$$-\sum_{i<t} \log \left( \mathbb{E}_{i-1} \left[ \exp \left( -\frac{1}{2} Z_i(\pi) \right) \right] \right) \leq \frac{1}{2} \left( \widehat{L}^{(t)}(\pi) - \widehat{L}^{(t)}(\pi_\beta^\star) \right) + \log(|\Pi|\delta^{-1}).$$

Next, using Eq. (27) and a somewhat standard argument from van de Geer (2000); Zhang (2006), we calculate that

$$\mathbb{E}_{i-1}\left[\exp\left(\frac{1}{2}Z_i(\pi)\right)\right]$$

$$= \mathbb{E}_{s_1\sim\rho,\tau\sim\pi^{(i)}|s_1,\widetilde{\tau}\sim\widetilde{\pi}^{(i)}|s_1,y\sim P_{\pi_\beta^\star}(\cdot|\tau,\widetilde{\tau})}\left[\exp\left(\frac{1}{2}\log(P_\pi(y\mid\tau,\widetilde{\tau})/P_{\pi_\beta^\star}(y\mid\tau,\widetilde{\tau}))\right)\right]$$

$$= \mathbb{E}_{s_1\sim\rho,\tau\sim\pi^{(i)}|s_1,\widetilde{\tau}\sim\widetilde{\pi}^{(i)}|s_1}\left[\sum_{y\in\{0,1\}}\sqrt{P_\pi(y\mid\tau,\widetilde{\tau})P_{\pi_\beta^\star}(y\mid\tau,\widetilde{\tau})}\right]$$

$$= \mathbb{E}_{s_1\sim\rho,\tau\sim\pi^{(i)}|s_1,\widetilde{\tau}\sim\widetilde{\pi}^{(i)}|s_1}\left[1-\frac{1}{2}D_{\mathsf{H}}^2\left(P_\pi(\cdot\mid\tau,\widetilde{\tau}),P_{\pi_\beta^\star}(\cdot\mid\tau,\widetilde{\tau})\right)\right].$$

Since $D_{\mathsf{H}}^2(\cdot,\cdot)\leq 2$ and $-\log(1-x)\geq x$ for $x\leq 1$, we conclude that

$$\sum_{i<t}\mathbb{E}_{s_1\sim\rho,\tau\sim\pi^{(i)}|s_1,\widetilde{\tau}\sim\widetilde{\pi}^{(i)}|s_1}\left[D_{\mathsf{H}}^2\left(P_\pi(\cdot\mid\tau,\widetilde{\tau}),P_{\pi_\beta^\star}(\cdot\mid\tau,\widetilde{\tau})\right)\right]\leq\widehat{L}^{(t)}(\pi)-\widehat{L}^{(t)}(\pi_\beta^\star)+2\log(|\Pi|\delta^{-1})$$

$\square$

**Proof of Lemma C.7.** Let $\tau^{(1)},\ldots,\tau^{(t-1)}$ denote the trajectories in $\mathcal{D}_{\mathsf{opt}}^{(t-1)}$. Let $\widehat{b}^{(i)}(\pi)=\alpha\log\pi(\tau^{(i)})$, and let

$$Z^{(i)}(\pi)=\widehat{b}^{(i)}(\pi)-\widehat{b}^{(i)}(\pi_\beta^\star).$$

We can equivalently re-write this as

$$Z^{(i)}(\pi)=\alpha\left(\log\left(\frac{\pi(\tau^{(i)})}{\pi_{\mathsf{ref}}(\tau^{(i)})}\right)-\log\left(\frac{\pi_\beta^\star(\tau^{(i)})}{\pi_{\mathsf{ref}}(\tau^{(i)})}\right)\right),$$

which implies that $|Z^{(i)}(\pi)|\leq 2\frac{\alpha}{\beta}V_{\mathsf{max}}$. From here, the result follows immediately by applying Lemma B.1 with the sequence $(Z_i(\pi))$ and taking a union bound over $\pi\in\Pi$. $\square$

**Proof of Lemma C.8.** We consider three cases. First, if $x\in[-2Y,2Y]$, then

$$|\sigma(x)-\sigma(y)|\geq\sigma'(z)|x-y|$$

for some $z\in[-2Y,2Y]$. In this regime, we have $\sigma'(z)\geq\sigma'(2Y)=e^{2Y}/(1+e^{2Y})^2\geq(4e^{2Y})^{-1}$. Next, if $x\geq 2Y>0$, we can directly bound

$$\sigma(x)-\sigma(y)\geq\sigma(2Y)-\sigma(Y)=\frac{e^{2Y}-e^Y}{(1+e^{2Y})(1+e^Y)}\geq\frac{1-e^{-Y}}{4e^Y}\geq\frac{1}{8e^Y},$$

where the last line holds whenever $Y\geq 1$. We conclude in this case that

$$\frac{|x-y|}{\sigma(x)-\sigma(y)}\leq\frac{X+Y}{\sigma(x)-\sigma(y)}\leq 8(X+Y)e^Y.$$

Finally, we consider the case where $x\leq -2Y\leq 0$. In this case, we can similarly lower bound

$$\sigma(y)-\sigma(x)\geq\sigma(-Y)-\sigma(-2Y)=\frac{e^{-Y}-e^{-2Y}}{(1+e^{-Y})(1+e^{-2Y})}\geq\frac{1-e^{-Y}}{4e^{2Y}}\geq\frac{1}{8e^{2Y}}$$

as long as $Y\geq 1$. From here, proceeding in the same fashion as the second case yields the result. $\square$

## C.7 Proof of Theorem 3.1′

**Proof of Theorem 3.1′.** Before diving into the proof, we re-state two central technical lemmas. The first lemma, generalizing Watson et al. (2023); Rafailov et al. (2024), shows that the optimal KL-regularized policy $\pi_\beta^\star$ can be viewed as implicitly modeling rewards.

**Lemma C.3** (Implicit $Q^\star$-Approximation). *For any DCMDP, it holds that for all admissible[9] trajectories $\tau = (s_1, a_1), \ldots, (s_H, a_H)$,*

$$\beta \log \frac{\pi_\beta^\star(\tau)}{\pi_{\mathsf{ref}}(\tau)} = r(\tau) - V_\beta^\star(s_1), \tag{24}$$

*where $V_\beta^\star$ is the KL-regularized value function defined in Eq. (23).*

This lemma allows us to view the DPO objective as a form of implicit $Q^\star$-approximation. Building on this lemma, we prove the following regret decomposition.

**Lemma 3.1** (Central regret decomposition). *For any pair of policies $\pi$ and $\nu$, it holds that*

$$J_\beta(\pi_\beta^\star) - J_\beta(\pi) = \mathbb{E}_{\tau \sim \nu} \left[ \beta \log \pi(\tau) \right] - \mathbb{E}_{\tau \sim \nu} \left[ \beta \log \pi_\beta^\star(\tau) \right] \tag{10}$$

$$+ \mathbb{E}_{\tau \sim \pi} \left[ \beta \log \frac{\pi(\tau)}{\pi_{\mathsf{ref}}(\tau)} - r(\tau) \right] - \mathbb{E}_{\tau \sim \nu} \left[ \beta \log \frac{\pi(\tau)}{\pi_{\mathsf{ref}}(\tau)} - r(\tau) \right]. \tag{11}$$

This result shows that the (regularized) regret of any policy $\pi$ can be decomposed into two terms. The term in Eq. (11) measures the extent to which $\pi$ (implicitly) models the reward; by Lemma C.3, this term is zero when $\pi = \pi_\beta^\star$. Meanwhile, the term in Eq. (10) measures the extent to which the policy $\pi$ over-estimates the internal reward; we will control this term using optimism. Importantly, the regret decomposition in Lemma 3.1 holds for an arbitrary roll-in policy $\nu$. This will facilitate minimizing the terms in the regret decomposition in a data-driven fashion. Before proceeding, we remark that Lemma C.3 and Lemma 3.1 together imply that

$$J_\beta(\pi_\beta^\star) - J_\beta(\pi) \leq 6V_{\mathsf{max}} \tag{31}$$

for all $\pi \in \Pi$.

We now begin the proof by writing

$$J_\beta(\pi_\beta^\star) - J_\beta(\widehat{\pi}) = \min_{t \in [T+1]} J_\beta(\pi_\beta^\star) - J_\beta(\pi^{(t)}) \leq \frac{1}{T} \sum_{t=1}^{T} J_\beta(\pi_\beta^\star) - J_\beta(\pi^{(t)}).$$

For each step $t$, we apply Lemma 3.1 with $\pi = \pi^{(t)}$ and $\nu = \widetilde{\pi}^{(t-1)}$, which gives

$$\frac{1}{T} \sum_{t=1}^{T} J_\beta(\pi_\beta^\star) - J_\beta(\pi^{(t)})$$

$$\leq \frac{1}{T} \sum_{t=1}^{T} \mathbb{E}_{\tau \sim \widetilde{\pi}^{(t-1)}} \left[ \beta \log \pi^{(t)}(\tau) - \beta \log \pi_\beta^\star(\tau) \right]$$

$$+ \frac{1}{T} \sum_{t=1}^{T} \mathbb{E}_{\tau \sim \pi^{(t)}} \left[ \beta \log \frac{\pi^{(t)}(\tau)}{\pi_{\mathsf{ref}}(\tau)} - r(\tau) \right] - \mathbb{E}_{\tau \sim \widetilde{\pi}^{(t-1)}} \left[ \beta \log \frac{\pi^{(t)}(\tau)}{\pi_{\mathsf{ref}}(\tau)} - r(\tau) \right].$$

$$= \frac{1}{T} \sum_{t=1}^{T} \mathbb{E}_{\tau \sim \widetilde{\pi}^{(t-1)}} \left[ \beta \log \pi^{(t)}(\tau) - \beta \log \pi_\beta^\star(\tau) \right]$$

$$+ \frac{1}{T} \sum_{t=1}^{T} \mathbb{E}_{s_1 \sim \rho, \tau \sim \pi^{(t)} | s_1, \widetilde{\tau} \sim \widetilde{\pi}^{(t-1)} | s_1} \left[ \beta \log \frac{\pi^{(t)}(\tau)}{\pi_{\mathsf{ref}}(\tau)} - r(\tau) - \beta \log \frac{\pi^{(t)}(\widetilde{\tau})}{\pi_{\mathsf{ref}}(\widetilde{\tau})} + r(\widetilde{\tau}) \right].$$

$$\leq \frac{6V_{\mathsf{max}}}{T} + \frac{1}{T} \sum_{t=2}^{T} \mathbb{E}_{\tau \sim \widetilde{\pi}^{(t-1)}} \left[ \beta \log \pi^{(t)}(\tau) - \beta \log \pi_\beta^\star(\tau) \right] \tag{32}$$

$$+ \frac{1}{T} \sum_{t=2}^{T} \mathbb{E}_{s_1 \sim \rho, \tau \sim \pi^{(t)} | s_1, \widetilde{\tau} \sim \widetilde{\pi}^{(t-1)} | s_1} \left[ \beta \log \frac{\pi^{(t)}(\tau)}{\pi_{\mathsf{ref}}(\tau)} - r(\tau) - \beta \log \frac{\pi^{(t)}(\widetilde{\tau})}{\pi_{\mathsf{ref}}(\widetilde{\tau})} + r(\widetilde{\tau}) \right],$$

where the last line follows by Eq. (31).

---

[9]We use "admissible" to a refer to a trajectory generated by executing an arbitrary policy $\pi : \mathcal{S} \to \Delta(\mathcal{A})$ in the MDP.

Next, recall that we define $\boldsymbol{\mu}^{(t)} = \frac{1}{t-1} \sum_{i<t} \pi^{(t)} \otimes \widetilde{\pi}^{(t)}$ Consider a fixed step $t \geq 2$, and define

$$\mathcal{I}^{(t)} := \frac{\left(\mathbb{E}_{s_1 \sim \rho, \tau \sim \pi^{(t)}|s_1, \widetilde{\tau} \sim \widetilde{\pi}^{(t-1)}|s_1}\left[\beta \log \frac{\pi^{(t)}(\tau)}{\pi_{\mathsf{ref}}(\tau)} - r(\tau) - \beta \log \frac{\pi^{(t)}(\widetilde{\tau})}{\pi_{\mathsf{ref}}(\widetilde{\tau})} + r(\widetilde{\tau})\right]\right)^2}{V_{\mathsf{max}}^2 \vee (t-1) \cdot \mathbb{E}_{s_1 \sim \rho, (\tau, \widetilde{\tau}) \sim \boldsymbol{\mu}^{(t)}|s_1}\left[\left(\beta \log \frac{\pi^{(t)}(\tau)}{\pi_{\mathsf{ref}}(\tau)} - r(\tau) - \beta \log \frac{\pi^{(t)}(\widetilde{\tau})}{\pi_{\mathsf{ref}}(\widetilde{\tau})} + r(\widetilde{\tau})\right)^2\right]}.$$

Then, using the AM-GM inequality, for any $\eta > 0$ we can bound

$$\mathbb{E}_{s_1 \sim \rho, \tau \sim \pi^{(t)}|s_1, \widetilde{\tau} \sim \widetilde{\pi}^{(t-1)}|s_1}\left[\beta \log \frac{\pi^{(t)}(\tau)}{\pi_{\mathsf{ref}}(\tau)} - r(\tau) - \beta \log \frac{\pi^{(t)}(\widetilde{\tau})}{\pi_{\mathsf{ref}}(\widetilde{\tau})} + r(\widetilde{\tau})\right]$$

$$\leq \frac{\mathcal{I}^{(t)}}{2\eta} + \frac{\eta}{2} \cdot \left(V_{\mathsf{max}}^2 \vee (t-1) \cdot \mathbb{E}_{s_1 \sim \rho, (\tau, \widetilde{\tau}) \sim \boldsymbol{\mu}^{(t)}|s_1}\left[\left(\beta \log \frac{\pi^{(t)}(\tau)}{\pi_{\mathsf{ref}}(\tau)} - r(\tau) - \beta \log \frac{\pi^{(t)}(\widetilde{\tau})}{\pi_{\mathsf{ref}}(\widetilde{\tau})} + r(\widetilde{\tau})\right)^2\right]\right)$$

$$\leq \frac{\mathcal{I}^{(t)}}{2\eta} + \frac{\eta}{2} \cdot \left(V_{\mathsf{max}}^2 + (t-1) \cdot \mathbb{E}_{s_1 \sim \rho, (\tau, \widetilde{\tau}) \sim \boldsymbol{\mu}^{(t)}|s_1}\left[\left(\beta \log \frac{\pi^{(t)}(\tau)}{\pi_{\mathsf{ref}}(\tau)} - r(\tau) - \beta \log \frac{\pi^{(t)}(\widetilde{\tau})}{\pi_{\mathsf{ref}}(\widetilde{\tau})} + r(\widetilde{\tau})\right)^2\right]\right).$$
(33)

Note that by definition, we have that $\sum_{t=1}^{T} \mathcal{I}^{(t)} \leq \mathsf{SEC}_{\mathsf{RLHF}}(\Pi, T, \beta; \boldsymbol{\pi}_{\mathsf{samp}})$. Hence, by plugging Eq. (33) into Eq. (32) and summing, we conclude that

$$\frac{1}{T} \sum_{t=1}^{T} J_\beta(\pi_\beta^\star) - J_\beta(\pi^{(t)})$$

$$\leq \frac{6 V_{\mathsf{max}}}{T} + \frac{\mathsf{SEC}_{\mathsf{RLHF}}(\Pi, T, \beta; \boldsymbol{\pi}_{\mathsf{samp}})}{2\eta T} + \frac{\eta}{2} V_{\mathsf{max}}^2 + \frac{1}{T} \sum_{t=2}^{T} \mathbb{E}_{\tau \sim \widetilde{\pi}^{(t-1)}}\left[\beta \log \pi^{(t)}(\tau) - \beta \log \pi_\beta^\star(\tau)\right]$$

$$+ \frac{\eta}{2T} \sum_{t=2}^{T} (t-1) \cdot \mathbb{E}_{s_1 \sim \rho, (\tau, \widetilde{\tau}) \sim \boldsymbol{\mu}^{(t)}|s_1}\left[\left(\beta \log \frac{\pi^{(t)}(\tau)}{\pi_{\mathsf{ref}}(\tau)} - r(\tau) - \beta \log \frac{\pi^{(t)}(\widetilde{\tau})}{\pi_{\mathsf{ref}}(\widetilde{\tau})} + r(\widetilde{\tau})\right)^2\right].$$
(34)

Fix $t$, and consider the term

$$\mathbb{E}_{\tau \sim \widetilde{\pi}^{(t-1)}}\left[\beta \log \pi^{(t)}(\tau) - \beta \log \pi_\beta^\star(\tau)\right] \tag{35}$$

$$+ \frac{\eta(t-1)}{2} \mathbb{E}_{s_1 \sim \rho, (\tau, \widetilde{\tau}) \sim \boldsymbol{\mu}^{(t)}|s_1}\left[\left(\beta \log \frac{\pi^{(t)}(\tau)}{\pi_{\mathsf{ref}}(\tau)} - r(\tau) - \beta \log \frac{\pi^{(t)}(\widetilde{\tau})}{\pi_{\mathsf{ref}}(\widetilde{\tau})} + r(\widetilde{\tau})\right)^2\right]$$

above. Let $f_\pi(\tau, \widetilde{\tau}) := \beta \log \frac{\pi(\tau)}{\pi_{\mathsf{ref}}(\tau)} - \beta \log \frac{\pi(\widetilde{\tau})}{\pi_{\mathsf{ref}}(\widetilde{\tau})}$. By Lemma C.3, we have that for any pair of admissible trajectories $(\tau, \widetilde{\tau})$ that share the initial state $s_1$, $f_{\pi_\beta^\star}(\tau, \widetilde{\tau}) = r(\tau) - r(\widetilde{\tau})$, so we can rewrite Eq. (35) as

$$\mathbb{E}_{\tau \sim \widetilde{\pi}^{(t-1)}}\left[\beta \log \pi^{(t)}(\tau) - \beta \log \pi_\beta^\star(\tau)\right] + \frac{\eta(t-1)}{2} \mathbb{E}_{s_1 \sim \rho, (\tau, \widetilde{\tau}) \sim \boldsymbol{\mu}^{(t)}|s_1}\left[\left(f_{\pi^{(t)}}(\tau, \widetilde{\tau}) - f_{\pi_\beta^\star}(\tau, \widetilde{\tau})\right)^2\right].$$
(36)

We now recall the central concentration lemma for XPO (Lemma C.5).

**Lemma C.5** (Concentration for XPO). *Suppose that Assumptions 3.1 and 3.2 hold. Then Algorithm 1 guarantees that with probability at least $1 - \delta$, for all steps $t \in [T]$,*

$$\alpha \cdot \mathbb{E}_{s_1 \sim \rho, \tau \sim \widetilde{\pi}^{(t-1)}}\left[\log(\pi^{(t)}(\tau)) - \log(\pi_\beta^\star(\tau))\right] + \kappa \cdot \mathbb{E}_{s_1 \sim \rho, (\tau, \widetilde{\tau}) \sim \boldsymbol{\mu}^{(t)}|s_1}\left[\left(f_{\pi^{(t)}}(\tau, \widetilde{\tau}) - f_{\pi_\beta^\star}(\tau, \widetilde{\tau})\right)^2\right]$$

$$\leq \frac{2 \log(2|\Pi|T\delta^{-1})}{t-1} + \frac{\alpha}{\beta} V_{\mathsf{max}} \sqrt{\frac{2^4 \log(2|\Pi|T\delta^{-1})}{t-1}},$$

*for $\kappa := (8(R_{\mathsf{max}} + V_{\mathsf{max}})e^{2R_{\mathsf{max}}})^{-2}$.*

It follows that if we set $\eta = \frac{\beta\kappa}{\alpha T} \leq \frac{\beta\kappa}{\alpha(t-1)}$, then with probability at least $1 - \delta$, for all $t \in [T]$,

$$\text{Eq. (36)} \lesssim \frac{\beta}{\alpha} \cdot \left( \frac{\log(|\Pi|T\delta^{-1})}{t-1} + \frac{\alpha}{\beta} V_{\max} \sqrt{\frac{\log(|\Pi|T\delta^{-1})}{t-1}} \right)$$

$$= \frac{\beta \log(|\Pi|T\delta^{-1})}{\alpha(t-1)} + V_{\max} \sqrt{\frac{\log(|\Pi|T\delta^{-1})}{t-1}}.$$

Plugging this bound back into Eq. (34), we have that

$$\frac{1}{T} \sum_{t=1}^{T} J_\beta(\pi_\beta^\star) - J_\beta(\pi^{(t)})$$

$$\lesssim \frac{V_{\max}}{T} + \frac{\mathsf{SEC}_{\mathsf{RLHF}}(\Pi, T, \beta; \boldsymbol{\pi}_{\mathsf{samp}})}{\eta T} + \eta V_{\max}^2 + \frac{1}{T} \sum_{t=2}^{T} \left( \frac{\beta \log(|\Pi|T\delta^{-1})}{\alpha(t-1)} + V_{\max} \sqrt{\frac{\log(|\Pi|T\delta^{-1})}{t-1}} \right)$$

$$\lesssim \frac{V_{\max}}{T} + \frac{\mathsf{SEC}_{\mathsf{RLHF}}(\Pi, T, \beta; \boldsymbol{\pi}_{\mathsf{samp}})}{\eta T} + \eta V_{\max}^2 + \frac{\beta \log(|\Pi|T\delta^{-1}) \log(T)}{\alpha T} + V_{\max} \sqrt{\frac{\log(|\Pi|T\delta^{-1})}{T}}$$

$$= \frac{V_{\max}}{T} + \frac{\alpha \cdot \mathsf{SEC}_{\mathsf{RLHF}}(\Pi, T, \beta; \boldsymbol{\pi}_{\mathsf{samp}})}{\beta\kappa} + \frac{\beta\kappa V_{\max}^2}{\alpha T} + \frac{\beta \log(|\Pi|T\delta^{-1}) \log(T)}{\alpha T} + V_{\max} \sqrt{\frac{\log(|\Pi|T\delta^{-1})}{T}}$$

$$\lesssim \frac{\alpha \cdot \mathsf{SEC}_{\mathsf{RLHF}}(\Pi, T, \beta; \boldsymbol{\pi}_{\mathsf{samp}})}{\beta\kappa} + \frac{\beta\kappa V_{\max}^2}{\alpha T} + \frac{\beta \log(|\Pi|T\delta^{-1}) \log(T)}{\alpha T} + V_{\max} \sqrt{\frac{\log(|\Pi|T\delta^{-1})}{T}}$$

$$\lesssim \frac{\alpha \cdot \mathsf{SEC}_{\mathsf{RLHF}}(\Pi, T, \beta; \boldsymbol{\pi}_{\mathsf{samp}})}{\beta\kappa} + \frac{\beta \log(|\Pi|T\delta^{-1}) \log(T)}{\alpha T} + V_{\max} \sqrt{\frac{\log(|\Pi|T\delta^{-1})}{T}},$$

where the last line uses that $\kappa \leq V_{\max}^{-2}$. It follows that by choosing

$$\alpha \propto \sqrt{\frac{\beta\kappa \cdot \beta \log(|\Pi|T\delta^{-1}) \log(T)}{T \cdot \mathsf{SEC}_{\mathsf{RLHF}}(\Pi, T, \beta; \boldsymbol{\pi}_{\mathsf{samp}})}},$$

we obtain

$$\frac{1}{T} \sum_{t=1}^{T} J_\beta(\pi_\beta^\star) - J_\beta(\pi^{(t)}) \tag{37}$$

$$\lesssim \sqrt{\frac{\kappa^{-1} \log(|\Pi|T\delta^{-1}) \log(T)) \cdot \mathsf{SEC}_{\mathsf{RLHF}}(\Pi, T, \beta; \boldsymbol{\pi}_{\mathsf{samp}})}{T}} + V_{\max} \sqrt{\frac{\log(|\Pi|T\delta^{-1})}{T}} \tag{38}$$

$$\leq O(V_{\max} + \kappa^{-1/2}) \cdot \sqrt{\frac{\mathsf{SEC}_{\mathsf{RLHF}}(\Pi, T, \beta; \boldsymbol{\pi}_{\mathsf{samp}}) \log(|\Pi|\delta^{-1}) \log(T)}{T}}. \tag{39}$$

Finally, we note that $(V_{\max} + \kappa^{-1/2}) = O((V_{\max} + R_{\max})e^{2R_{\max}})$.

$\square$

## C.8 PROOFS FOR SEC BOUNDS

**Proof of Lemma C.1.** This proof is based on Proposition 19 of Xie et al. (2023), with some additional modifications to handle the preference-based setting. Let $T \in \mathbb{N}$ and policies $\pi^{(1)}, \ldots, \pi^{(T)}$ be given, and recall that $\widetilde{\pi}^{(t)} = \boldsymbol{\pi}_{\mathsf{samp}}(\pi^{(1)}, \ldots, \pi^{(t)})$. Define

$$\delta^{(t)}(\tau, \widetilde{\tau}) = \beta \log \frac{\pi^{(t)}(\tau)}{\pi_{\mathsf{ref}}(\tau)} - r(\tau) - \beta \log \frac{\pi^{(t)}(\widetilde{\tau})}{\pi_{\mathsf{ref}}(\widetilde{\tau})} + r(\widetilde{\tau}),$$

and note that by Lemma C.3, we have $|\delta^{(t)}(\tau, \widetilde{\tau})| \leq 4V_{\max}$ whenever $\tau$ and $\widetilde{\tau}$ share the same initial state $s_1$. Let $\mathbb{E}_{\pi,\pi'}$ denote the expectation over trajectories induced by sampling $s_1 \sim \rho, \tau \sim \pi \mid s_1$, and $\widetilde{\tau} \sim \pi' \mid s_1$. Meanwhile, let $\mathbb{E}_{\boldsymbol{\mu}^{(t)}}$ denote the expectation over trajectories induced by sampling $s_1 \sim \rho$ and $(\tau, \widetilde{\tau}) \sim \boldsymbol{\mu}^{(t)} \mid s_1$. Then our goal is to bound

$$\mathsf{Val} := \sum_{t=1}^{T} \frac{\left( \mathbb{E}_{\pi^{(t)}, \widetilde{\pi}^{(t-1)}} [\delta^{(t)}(\tau, \widetilde{\tau})] \right)^2}{V_{\max}^2 \vee (t-1) \cdot \mathbb{E}_{\boldsymbol{\mu}^{(t)}} [(\delta^{(t)}(\tau, \widetilde{\tau}))^2]}.$$

Let

$$\nu = \underset{\nu \in \Delta((\mathcal{S} \times \mathcal{A})^H)}{\operatorname{argmin}} \sup_{\tau \in (\mathcal{S} \times \mathcal{A})^H} \sup_{\pi \in \Pi} \frac{d^\pi(\tau)}{\nu(\tau)}$$

be the distribution that achieves the value of the coverability coefficient in Definition 3.1. Let us abbreviate $C_{\mathsf{cov}} \equiv C_{\mathsf{cov}}(\Pi)$. For a trajectory $\tau$, let

$$\mathsf{t}(\tau) := \min\left\{ t \mid \sum_{i<t} d^{\pi^{(i)}}(\tau) \geq C_{\mathsf{cov}} \cdot \nu(\tau) \right\}.$$

Then we can bound

$$\mathsf{Val} \leq \underbrace{\sum_{t=1}^T \frac{\left(\mathbb{E}_{\pi^{(t)}, \widetilde{\pi}^{(t-1)}}[\delta^{(t)}(\tau, \widetilde{\tau})\mathbb{I}\{t < \mathsf{t}(\tau)\}]\right)^2}{V_{\mathsf{max}}^2 \vee (t-1) \cdot \mathbb{E}_{\boldsymbol{\mu}^{(t)}}[(\delta^{(t)}(\tau, \widetilde{\tau}))^2]}}_{=:(\mathrm{I})} + \underbrace{\sum_{t=1}^T \frac{\left(\mathbb{E}_{\pi^{(t)}, \widetilde{\pi}^{(t-1)}}[\delta^{(t)}(\tau, \widetilde{\tau})\mathbb{I}\{t \geq \mathsf{t}(\tau)\}]\right)^2}{V_{\mathsf{max}}^2 \vee (t-1) \cdot \mathbb{E}_{\boldsymbol{\mu}^{(t)}}[(\delta^{(t)}(\tau, \widetilde{\tau}))^2]}}_{=:(\mathrm{II})}.$$

We begin by bounding the first term by

$$(\mathrm{I}) \leq \frac{1}{V_{\mathsf{max}}^2} \sum_{t=1}^T \left(\mathbb{E}_{\pi^{(t)}, \widetilde{\pi}^{(t-1)}}[\delta^{(t)}(\tau, \widetilde{\tau})\mathbb{I}\{t < \mathsf{t}(\tau)\}]\right)^2 \leq 16 \sum_{t=1}^T \mathbb{E}_{\pi^{(t)}}[\mathbb{I}\{t < \mathsf{t}(\tau)\}].$$

Letting $\mathcal{T} := (\mathcal{S} \times \mathcal{A})^H$, we can further bound this by

$$\sum_{t=1}^T \mathbb{E}_{\pi^{(t)}}[\mathbb{I}\{t < \mathsf{t}(\tau)\}] = \sum_{\tau \in \mathcal{T}} \sum_{t=1}^T d^{\pi^{(t)}}(\tau)\mathbb{I}\{t < \mathsf{t}(\tau)\}$$

$$= \sum_{\tau \in \mathcal{T}} \left( \sum_{i=1}^{\mathsf{t}(\tau)-2} d^{\pi^{(i)}}(\tau) \right) + d^{\pi^{(\mathsf{t}(\tau)-1)}}(\tau)$$

$$\leq 2 C_{\mathsf{cov}} \sum_{\tau \in \mathcal{T}} \nu(\tau) = 2 C_{\mathsf{cov}},$$

so that $(\mathrm{I}) \leq 32 C_{\mathsf{cov}}$.

We now bound term (II). Define $d^{\pi, \pi'}(\tau', \widetilde{\tau}') = \mathbb{P}_{s_1 \sim \rho, \tau \sim \pi | s_1, \widetilde{\tau} \sim \pi' | s_1}(\tau = \tau', \widetilde{\tau} = \widetilde{\tau}')$ and $d^{\boldsymbol{\mu}^{(t)}}(\tau', \widetilde{\tau}') = \frac{1}{t-1} \sum_{i<t} d^{\pi^{(i)}, \widetilde{\pi}^{(i)}}(\tau', \widetilde{\tau}')$. For each $t$, we can write

$$\mathbb{E}_{\pi^{(t)}, \widetilde{\pi}^{(t-1)}}[\delta^{(t)}(\tau, \widetilde{\tau})\mathbb{I}\{t < \mathsf{t}(\tau)\}]$$

$$= \sum_{\tau, \widetilde{\tau} \in \mathcal{T}} d^{\pi^{(t)}, \widetilde{\pi}^{(t-1)}}(\tau, \widetilde{\tau})\delta^{(t)}(\tau, \widetilde{\tau})\mathbb{I}\{t \geq \mathsf{t}(\tau)\}$$

$$= \sum_{\tau, \widetilde{\tau} \in \mathcal{T}} d^{\pi^{(t)}, \widetilde{\pi}^{(t-1)}}(\tau, \widetilde{\tau})\delta^{(t)}(\tau, \widetilde{\tau}) \left( \frac{d^{\boldsymbol{\mu}^{(t)}}(\tau, \widetilde{\tau})}{d^{\boldsymbol{\mu}^{(t)}}(\tau, \widetilde{\tau})} \right)^{1/2} \mathbb{I}\{t \geq \mathsf{t}(\tau)\}$$

$$\leq \left( \sum_{\tau, \widetilde{\tau} \in \mathcal{T}} \frac{(d^{\pi^{(t)}, \widetilde{\pi}^{(t-1)}}(\tau, \widetilde{\tau}))^2 \mathbb{I}\{t \geq \mathsf{t}(\tau)\}}{(t-1) \cdot d^{\boldsymbol{\mu}^{(t)}}(\tau, \widetilde{\tau})} \right)^{1/2} \cdot \left( (t-1) \cdot \mathbb{E}_{\boldsymbol{\mu}^{(t)}}[(\delta^{(t)}(\tau, \widetilde{\tau}))^2] \right)^{1/2},$$

where the last inequality is by Cauchy-Schwarz. We conclude that

$$(\mathrm{II}) \leq \sum_{t=1}^T \sum_{\tau, \widetilde{\tau} \in \mathcal{T}} \frac{(d^{\pi^{(t)}, \widetilde{\pi}^{(t-1)}}(\tau, \widetilde{\tau}))^2 \mathbb{I}\{t \geq \mathsf{t}(\tau)\}}{(t-1) \cdot d^{\boldsymbol{\mu}^{(t)}}(\tau, \widetilde{\tau})}.$$

To proceed, we restrict our attention to the case where $\widetilde{\pi}^{(t)} = \widetilde{\pi}$ for all $t$ for some fixed $\widetilde{\pi}$. We observe that in this case, for all $t$,

$$\frac{d^{\pi^{(t)}, \widetilde{\pi}^{(t-1)}}(\tau, \widetilde{\tau})}{d^{\boldsymbol{\mu}^{(t)}}(\tau, \widetilde{\tau})} = \frac{d^{\pi^{(t)}, \widetilde{\pi}}(\tau, \widetilde{\tau})}{\frac{1}{t-1} \sum_{i<t} d^{\pi^{(i)}, \widetilde{\pi}}(\tau, \widetilde{\tau})} = \frac{d^{\pi^{(t)}}(\tau)}{\frac{1}{t-1} \sum_{i<t} d^{\pi^{(i)}}(\tau)},$$

since $\tau$ and $\widetilde{\tau}$ are conditionally independent given $s_1$, and since $d^{\pi,\pi'}(\tau, \widetilde{\tau}) = 0$ if $\tau, \widetilde{\tau}$ do not share the same $s_1$. It follows that

$$
\begin{aligned}
(\text{II}) &\leq \sum_{t=1}^{T} \sum_{\tau, \widetilde{\tau} \in \mathcal{T}} \frac{d^{\pi^{(t)}}(\tau) d^{\pi^{(t)}, \widetilde{\pi}}(\tau, \widetilde{\tau}) \mathbb{I}\{t \geq \mathsf{t}(\tau)\}}{\sum_{i<t} d^{\pi^{(i)}}(\tau)} \\
&= \sum_{\tau} \sum_{t=1}^{T} \frac{(d^{\pi^{(t)}}(\tau))^2 \mathbb{I}\{t \geq \mathsf{t}(\tau)\}}{\sum_{i<t} d^{\pi^{(i)}}(\tau)} \\
&\leq 2 \sum_{\tau} \sum_{t=1}^{T} \frac{(d^{\pi^{(t)}}(\tau))^2}{\sum_{i<t} d^{\pi^{(i)}}(\tau) + C_{\mathsf{cov}} \nu(\tau)} \\
&\leq 2 C_{\mathsf{cov}} \sum_{\tau} \nu(\tau) \sum_{t=1}^{T} \frac{d^{\pi^{(t)}}(\tau)}{\sum_{i<t} d^{\pi^{(i)}}(\tau) + C_{\mathsf{cov}} \nu(\tau)}.
\end{aligned}
$$

Finally, by Lemma 4 of Xie et al. (2023), we have that for all $\tau \in \mathcal{T}$, $\sum_{t=1}^{T} \frac{d^{\pi^{(t)}}(\tau)}{\sum_{i<t} d^{\pi^{(i)}}(\tau) + C_{\mathsf{cov}} \nu(\tau)} \leq O(\log(T))$, which yields $(\text{II}) \leq O(C_{\mathsf{cov}} \log(T))$. This proves the result.

$\square$

**Proof for Example C.2.** We claim for any pair of trajectories $\tau, \widetilde{\tau}$ and function $f \in \mathcal{F}$, we can write

$$
\sum_{h=1}^{H} (f(s_h, a_h) - [\mathcal{T}_\beta f](s_h, a_h)) - (f(\widetilde{s}_h, \widetilde{a}_h) - [\mathcal{T}_\beta f](\widetilde{s}_h, \widetilde{a}_h)) = \langle X(\tau, \widetilde{\tau}), W(f) \rangle \tag{40}
$$

for embeddings $X(\tau, \widetilde{\tau}), W(f) \in \mathbb{R}^d$. To see this, note that $f(s_h, a_h) = \langle \phi(s_h, a_h), \theta_f \rangle$ for some $\theta_f \in \mathbb{R}^d$ with $\|\theta_f\| \leq B$ by definition, while the linear MDP property implies that we can write $[\mathcal{T}_\beta f](s_h, a_h) = \langle \phi(s_h, a_h), w_f \rangle$ for some $w_f \in \mathbb{R}^d$ with $\|w_f\| \leq O(\sqrt{d})$. It follows that we can take

$$
X(\tau, \widetilde{\tau}) = \sum_{h=1}^{H} \phi(s_h, a_h) - \phi(\widetilde{s}_h, \widetilde{a}_h) \in \mathbb{R}^d
$$

and

$$
W(f) = \theta_f - w_f \in \mathbb{R}^d.
$$

With this definition, we observe that in the case where $\widetilde{\pi}^{(t)} = \widetilde{\pi}$ for all $t$, we can write the value of $\mathsf{SEC}_{\mathsf{RLHF}}$ for a sequence of policies $\pi^{(1)}, \ldots, \pi^{(T)}$ as

$$
\sum_{t=1}^{T} \frac{\left( \mathbb{E}_{s_1 \sim \rho, \tau \sim \pi^{(t)} | s_1, \widetilde{\tau} \sim \widetilde{\pi} | s_1} [\langle X(\tau, \widetilde{\tau}), W(f^{(t)}) \rangle] \right)^2}{V_{\mathsf{max}}^2 \vee \sum_{i<t} \mathbb{E}_{s_1 \sim \rho, \tau \sim \pi^{(i)} | s_1, \widetilde{\tau} \sim \widetilde{\pi} | s_1} \left[ \langle X(\tau, \widetilde{\tau}), W(f^{(t)}) \rangle^2 \right]}
$$

In particular, if we define $W^{(t)} := W(f^{(t)})$ and $X^{(t)} = \mathbb{E}_{s_1 \sim \rho, \tau \sim \pi^{(t)} | s_1, \widetilde{\tau} \sim \widetilde{\pi} | s_1} [X(\tau, \widetilde{\tau})]$, it follows from Jensen's inequality that we can bound the quantity above by

$$
\sum_{t=1}^{T} \frac{\langle X^{(t)}, W^{(t)} \rangle^2}{V_{\mathsf{max}}^2 \vee \sum_{i<t} \langle X^{(i)}, W^{(t)} \rangle^2}
$$

Using that $\|X(\tau, \widetilde{\tau})\|, \|W(f)\| \leq \mathsf{poly}(H, d)$, it now follows from the standard elliptic potential argument (e.g., Du et al. (2021); Jin et al. (2021)) that $\mathsf{SEC}_{\mathsf{RLHF}}(\mathcal{F}, T; \boldsymbol{\pi}_{\mathsf{samp}}) \leq \widetilde{O}(d)$. $\square$

# D  GUARANTEES FOR XPO WITH LARGE BATCH SIZE

This section presents a general version of XPO which draws a large batch of responses for each update, allowing for fewer updates over all

---

**Algorithm 4** Exploratory Preference Optimization (XPO) with general sampling policy and large batch size.

---

   **input:** Number of iterations $T$, batch size $K$, KL-regularization coefficient $\beta > 0$, optimism coefficient $\alpha > 0$, sampling strategy $\boldsymbol{\pi}_{\mathsf{samp}}$.
1: Initialize $\pi^{(1)}, \widetilde{\pi}^{(1)} \leftarrow \pi_{\mathsf{ref}}, \mathcal{D}_{\mathsf{pref}}^{(0)} \leftarrow \varnothing$.
2: **for** iteration $t = 1, 2, \ldots, T$ **do**
3:      **for** $k = 1, \ldots, K$ **do**
4:          **Generate pair** $(\tau^{(t,k)}, \widetilde{\tau}^{(t,k)})$**:** $s_1^{(t,k)} \sim \rho$, $\tau^{(t,k)} \sim \pi^{(t)} \mid s_1^{(t,k)}$, and $\widetilde{\tau}^{(t,k)} \sim \widetilde{\pi}^{(t)} \mid s_1^{(t,k)}$.
5:          Label $(\tau^{(t,k)}, \widetilde{\tau}^{(t,k)})$ as $(\tau_+^{(t,k)}, \tau_-^{(t,k)})$ with preference $y^{(t,k)} \sim \mathbb{P}(\tau^{(t,k)} \succ \widetilde{\tau}^{(t,k)})$.
6:      **Update preference data:** $\mathcal{D}_{\mathsf{pref}}^{(t)} \leftarrow \mathcal{D}_{\mathsf{pref}}^{(t-1)} \bigcup \{(\tau_+^{(t,1)}, \tau_-^{(t,1)}), \ldots, (\tau_+^{(t,K)}, \tau_-^{(t,K)})\}$.
7:      **Update optimism data:** Compute dataset $\mathcal{D}_{\mathsf{opt}}^{(t)}$ of $t \cdot K$ samples from $\widetilde{\pi}^{(t)}$.
                                       `// When` $\widetilde{\pi}^{(t)} = \pi_{\mathsf{ref}}$`, can re-use previous samples as in Algorithm 1.`
8:      **Direct preference optimization with global optimism:** Calculate $\pi^{(t+1)}$ via

$$\pi^{(t+1)} \leftarrow \operatorname*{argmin}_{\pi \in \Pi} \left\{ \alpha \sum_{\tau \in \mathcal{D}_{\mathsf{opt}}^{(t)}} \log \pi(\tau) - \sum_{(\tau_+, \tau_-) \in \mathcal{D}_{\mathsf{pref}}^{(t)}} \log \left[ \sigma \left( \beta \log \frac{\pi(\tau_+)}{\pi_{\mathsf{ref}}(\tau_+)} - \beta \log \frac{\pi(\tau_-)}{\pi_{\mathsf{ref}}(\tau_-)} \right) \right] \right\}.$$

9:      **Update sampling policy:** $\widetilde{\pi}^{(t+1)} \leftarrow \boldsymbol{\pi}_{\mathsf{samp}}(\pi^{(1)}, \ldots, \pi^{(t+1)})$.
10: **return:** $\widehat{\pi} = \operatorname{argmax}_{\pi \in \{\pi^{(1)}, \ldots, \pi^{(T+1)}\}} J_\beta(\pi^{(t)})$.          `// Can compute using validation data.`

---

### D.1   XPO WITH LARGE BATCH SIZE

Algorithm 4 presents a version of XPO which is identical to Algorithm 2, except that the algorithm draws a batch of $K$ responses for each update.

**Main sample complexity guarantee.**   Our general sample complexity guarantee is as follows.

**Theorem D.1** (Guarantee for XPO with large batch size)**.** *Suppose that Assumptions 3.1 and 3.2 hold. Consider Algorithm 4 with $\widetilde{\pi}^{(t)} = \widetilde{\pi}$ for all $t \in [T]$. For any $\beta > 0$ and $T, K \in \mathbb{N}$, if we set $\alpha = c \cdot \frac{\beta}{(V_{\mathsf{max}} + R_{\mathsf{max}})e^{2R_{\mathsf{max}}}} \cdot \sqrt{\frac{\log(|\Pi|T\delta^{-1})}{KT \cdot C_{\mathsf{cov}}(\Pi)}}$ for an absolute constant $c > 0$, then Algorithm 4 ensures that with probability at least $1 - \delta$,*

$$J_\beta(\pi_\beta^\star) - J_\beta(\widehat{\pi}) \lesssim \frac{V_{\mathsf{max}} C_{\mathsf{cov}}(\Pi)}{T} + (V_{\mathsf{max}} + R_{\mathsf{max}})e^{2R_{\mathsf{max}}} \cdot \sqrt{\frac{C_{\mathsf{cov}}(\Pi) \log(|\Pi|T\delta^{-1}) \log^2(T)}{KT}}.$$

*In particular, to learn an $\varepsilon$-optimal policy, it suffices to set $T = \widetilde{O}\left(\frac{V_{\mathsf{max}} C_{\mathsf{cov}}(\Pi)}{\varepsilon}\right)$ and $K = \widetilde{O}\left(\frac{(V_{\mathsf{max}} + R_{\mathsf{max}})e^{4R_{\mathsf{max}}} \log(|\Pi|\delta^{-1})}{\varepsilon}\right)$. That is, compared to Algorithm 1, we only require $O(1/\varepsilon)$ policy updates instead of $O(1/\varepsilon^2)$ policy updates.*

### D.2   PROOF OF THEOREM D.1

**Proof of Theorem D.1.**   This proof closely follows that of Theorem 3.1. We begin by re-stating the two central technical lemmas.

**Lemma C.3** (Implicit $Q^\star$-Approximation)**.** *For any DCMDP, it holds that for all admissible[10] trajectories $\tau = (s_1, a_1), \ldots, (s_H, a_H)$,*

$$\beta \log \frac{\pi_\beta^\star(\tau)}{\pi_{\mathsf{ref}}(\tau)} = r(\tau) - V_\beta^\star(s_1), \tag{24}$$

*where $V_\beta^\star$ is the KL-regularized value function defined in Eq. (23).*

**Lemma 3.1** (Central regret decomposition)**.** *For any pair of policies $\pi$ and $\nu$, it holds that*

$$J_\beta(\pi_\beta^\star) - J_\beta(\pi) = \mathbb{E}_{\tau \sim \nu}[\beta \log \pi(\tau)] - \mathbb{E}_{\tau \sim \nu}[\beta \log \pi_\beta^\star(\tau)] \tag{10}$$

---

[10]We use "admissible" to a refer to a trajectory generated by executing an arbitrary policy $\pi : \mathcal{S} \to \Delta(\mathcal{A})$ in the MDP.

$$+ \mathbb{E}_{\tau \sim \pi} \left[ \beta \log \frac{\pi(\tau)}{\pi_{\text{ref}}(\tau)} - r(\tau) \right] - \mathbb{E}_{\tau \sim \nu} \left[ \beta \log \frac{\pi(\tau)}{\pi_{\text{ref}}(\tau)} - r(\tau) \right]. \tag{11}$$

This result shows that the (regularized) regret of any policy $\pi$ can be decomposed into two terms. The term in Eq. (11) measures the extent to which $\pi$ (implicitly) models the reward; by Lemma C.3, this term is zero when $\pi = \pi_\beta^\star$. Meanwhile, the term in Eq. (10) measures the extent to which the policy $\pi$ over-estimates the internal reward; we will control this term using optimism. Importantly, the regret decomposition in Lemma 3.1 holds for an arbitrary roll-in policy $\nu$. This will facilitate minimizing the terms in the regret decomposition in a data-driven fashion. Before proceeding, we remark that Lemma C.3 and Lemma 3.1 together imply that

$$J_\beta(\pi_\beta^\star) - J_\beta(\pi) \le 6V_{\text{max}} \tag{41}$$

for all $\pi \in \Pi$.

We now begin the proof by writing

$$J_\beta(\pi_\beta^\star) - J_\beta(\widehat{\pi}) = \min_{t \in [T+1]} J_\beta(\pi_\beta^\star) - J_\beta(\pi^{(t)}) \le \frac{1}{T} \sum_{t=1}^{T} J_\beta(\pi_\beta^\star) - J_\beta(\pi^{(t)}).$$

For each step $t$, we apply Lemma 3.1 with $\pi = \pi^{(t)}$ and $\nu = \widetilde{\pi}^{(t-1)}$, which gives

$$\frac{1}{T} \sum_{t=1}^{T} J_\beta(\pi_\beta^\star) - J_\beta(\pi^{(t)})$$

$$\le \frac{1}{T} \sum_{t=1}^{T} \mathbb{E}_{\tau \sim \widetilde{\pi}^{(t-1)}} \left[ \beta \log \pi^{(t)}(\tau) - \beta \log \pi_\beta^\star(\tau) \right]$$

$$+ \frac{1}{T} \sum_{t=1}^{T} \mathbb{E}_{\tau \sim \pi^{(t)}} \left[ \beta \log \frac{\pi^{(t)}(\tau)}{\pi_{\text{ref}}(\tau)} - r(\tau) \right] - \mathbb{E}_{\tau \sim \widetilde{\pi}^{(t-1)}} \left[ \beta \log \frac{\pi^{(t)}(\tau)}{\pi_{\text{ref}}(\tau)} - r(\tau) \right].$$

$$= \frac{1}{T} \sum_{t=1}^{T} \mathbb{E}_{\tau \sim \widetilde{\pi}^{(t-1)}} \left[ \beta \log \pi^{(t)}(\tau) - \beta \log \pi_\beta^\star(\tau) \right]$$

$$+ \frac{1}{T} \sum_{t=1}^{T} \mathbb{E}_{s_1 \sim \rho, \tau \sim \pi^{(t)} | s_1, \widetilde{\tau} \sim \widetilde{\pi}^{(t-1)} | s_1} \left[ \beta \log \frac{\pi^{(t)}(\tau)}{\pi_{\text{ref}}(\tau)} - r(\tau) - \beta \log \frac{\pi^{(t)}(\widetilde{\tau})}{\pi_{\text{ref}}(\widetilde{\tau})} + r(\widetilde{\tau}) \right] \tag{42}$$

$$\le \frac{6V_{\text{max}}}{T} + \frac{1}{T} \sum_{t=2}^{T} \mathbb{E}_{\tau \sim \widetilde{\pi}^{(t-1)}} \left[ \beta \log \pi^{(t)}(\tau) - \beta \log \pi_\beta^\star(\tau) \right] \tag{43}$$

$$+ \frac{1}{T} \sum_{t=2}^{T} \mathbb{E}_{s_1 \sim \rho, \tau \sim \pi^{(t)} | s_1, \widetilde{\tau} \sim \widetilde{\pi}^{(t-1)} | s_1} \left[ \beta \log \frac{\pi^{(t)}(\tau)}{\pi_{\text{ref}}(\tau)} - r(\tau) - \beta \log \frac{\pi^{(t)}(\widetilde{\tau})}{\pi_{\text{ref}}(\widetilde{\tau})} + r(\widetilde{\tau}) \right],$$

where the last line follows by Eq. (41).

Let $\delta^{(t)}(\tau, \widetilde{\tau}) := \beta \log \frac{\pi^{(t)}(\tau)}{\pi_{\text{ref}}(\tau)} - r(\tau) - \beta \log \frac{\pi^{(t)}(\widetilde{\tau})}{\pi_{\text{ref}}(\widetilde{\tau})} + r(\widetilde{\tau})$, and recall that we define $\boldsymbol{\mu}^{(t)} = \frac{1}{t-1} \sum_{i<t} \pi^{(t)} \otimes \widetilde{\pi}^{(t)}$. Using Lemma D.2 and the AM-GM inequality, we have that for any $\eta > 0$,

$$\sum_{t=2}^{T} \mathbb{E}_{\pi^{(t)}, \widetilde{\pi}^{(t-1)}} \left[ \delta^{(t)}(\tau, \widetilde{\tau}) \right]$$

$$\le \frac{\eta}{2} \cdot \sum_{t=2}^{T} (t-1) \cdot \mathbb{E}_{s_1 \sim \rho, (\tau, \widetilde{\tau}) \sim \boldsymbol{\mu}^{(t)} | s_1} \left[ (\delta^{(t)}(\tau, \widetilde{\tau}))^2 \right] + \frac{4 C_{\text{cov}}(\Pi) \log(T)}{\eta} + 12 V_{\text{max}} C_{\text{cov}}(\Pi).$$

Plugging this result into Eq. (43) and summing, we conclude that

$$\frac{1}{T} \sum_{t=1}^{T} J_\beta(\pi_\beta^\star) - J_\beta(\pi^{(t)})$$

$$\leq \frac{4C_{\mathsf{cov}}(\Pi)\log(T)}{\eta T} + 18V_{\mathsf{max}}C_{\mathsf{cov}}(\Pi) + \frac{1}{T}\sum_{t=2}^{T}\mathbb{E}_{\tau\sim\widetilde{\pi}^{(t-1)}}\left[\beta\log\pi^{(t)}(\tau) - \beta\log\pi_{\beta}^{\star}(\tau)\right]$$

$$+ \frac{\eta}{2T}\sum_{t=2}^{T}(t-1)\cdot\mathbb{E}_{s_1\sim\rho,(\tau,\widetilde{\tau})\sim\boldsymbol{\mu}^{(t)}|s_1}\left[\left(\beta\log\frac{\pi^{(t)}(\tau)}{\pi_{\mathsf{ref}}(\tau)} - r(\tau) - \beta\log\frac{\pi^{(t)}(\widetilde{\tau})}{\pi_{\mathsf{ref}}(\widetilde{\tau})} + r(\widetilde{\tau})\right)^2\right].$$

$$(44)$$

Fix $t$, and consider the term

$$\mathbb{E}_{\tau\sim\widetilde{\pi}^{(t-1)}}\left[\beta\log\pi^{(t)}(\tau) - \beta\log\pi_{\beta}^{\star}(\tau)\right] \tag{45}$$

$$+ \frac{\eta(t-1)}{2}\mathbb{E}_{s_1\sim\rho,(\tau,\widetilde{\tau})\sim\boldsymbol{\mu}^{(t)}|s_1}\left[\left(\beta\log\frac{\pi^{(t)}(\tau)}{\pi_{\mathsf{ref}}(\tau)} - r(\tau) - \beta\log\frac{\pi^{(t)}(\widetilde{\tau})}{\pi_{\mathsf{ref}}(\widetilde{\tau})} + r(\widetilde{\tau})\right)^2\right]$$

above. Let $f_{\pi}(\tau,\widetilde{\tau}) := \beta\log\frac{\pi(\tau)}{\pi_{\mathsf{ref}}(\tau)} - \beta\log\frac{\pi(\widetilde{\tau})}{\pi_{\mathsf{ref}}(\widetilde{\tau})}$. By Lemma C.3, we have that for any pair of admissible trajectories $(\tau,\widetilde{\tau})$ that share the initial state $s_1$, $f_{\pi_{\beta}^{\star}}(\tau,\widetilde{\tau}) = r(\tau) - r(\widetilde{\tau})$, so we can rewrite Eq. (45) as

$$\mathbb{E}_{\tau\sim\widetilde{\pi}^{(t-1)}}\left[\beta\log\pi^{(t)}(\tau) - \beta\log\pi_{\beta}^{\star}(\tau)\right] + \frac{\eta(t-1)}{2}\mathbb{E}_{s_1\sim\rho,(\tau,\widetilde{\tau})\sim\boldsymbol{\mu}^{(t)}|s_1}\left[\left(f_{\pi^{(t)}}(\tau,\widetilde{\tau}) - f_{\pi_{\beta}^{\star}}(\tau,\widetilde{\tau})\right)^2\right].$$

$$(46)$$

We now state a concentration lemma for XPO; this result is a straightforward generalization of Lemma C.5, and we omit the proof.

**Lemma D.1** (Concentration for XPO). *Suppose that Assumptions 3.1 and 3.2 hold. Then Algorithm 4 guarantees that with probability at least $1 - \delta$, for all steps $t \in [T]$,*

$$\alpha\cdot\mathbb{E}_{s_1\sim\rho,\tau\sim\widetilde{\pi}^{(t-1)}}\left[\log(\pi^{(t)}(\tau)) - \log(\pi_{\beta}^{\star}(\tau))\right] + \kappa\cdot\mathbb{E}_{s_1\sim\rho,(\tau,\widetilde{\tau})\sim\boldsymbol{\mu}^{(t)}|s_1}\left[\left(f_{\pi^{(t)}}(\tau,\widetilde{\tau}) - f_{\pi_{\beta}^{\star}}(\tau,\widetilde{\tau})\right)^2\right]$$

$$\leq \frac{2\log(2|\Pi|T\delta^{-1})}{K(t-1)} + \frac{\alpha}{\beta}V_{\mathsf{max}}\sqrt{\frac{2^4\log(2|\Pi|T\delta^{-1})}{K(t-1)}},$$

*for $\kappa := (8(R_{\mathsf{max}} + V_{\mathsf{max}})e^{2R_{\mathsf{max}}})^{-2}$.*

It follows that if we set $\eta = \frac{\beta\kappa}{\alpha T} \leq \frac{\beta\kappa}{\alpha(t-1)}$, then with probability at least $1 - \delta$, for all $t \in [T]$,

$$\text{Eq. (46)} \lesssim \frac{\beta}{\alpha}\cdot\left(\frac{\log(|\Pi|T\delta^{-1})}{K(t-1)} + \frac{\alpha}{\beta}V_{\mathsf{max}}\sqrt{\frac{\log(|\Pi|T\delta^{-1})}{K(t-1)}}\right)$$

$$= \frac{\beta\log(|\Pi|T\delta^{-1})}{\alpha K(t-1)} + V_{\mathsf{max}}\sqrt{\frac{\log(|\Pi|T\delta^{-1})}{K(t-1)}}.$$

Plugging this bound back into Eq. (44), we have that

$$\frac{1}{T}\sum_{t=1}^{T}J_{\beta}(\pi_{\beta}^{\star}) - J_{\beta}(\pi^{(t)})$$

$$\lesssim \frac{V_{\mathsf{max}}C_{\mathsf{cov}}(\Pi)}{T} + \frac{C_{\mathsf{cov}}(\Pi)\log(T)}{\eta T} + \frac{1}{T}\sum_{t=2}^{T}\left(\frac{\beta\log(|\Pi|T\delta^{-1})}{\alpha K(t-1)} + V_{\mathsf{max}}\sqrt{\frac{\log(|\Pi|T\delta^{-1})}{K(t-1)}}\right)$$

$$\lesssim \frac{V_{\mathsf{max}}C_{\mathsf{cov}}(\Pi)}{T} + \frac{C_{\mathsf{cov}}(\Pi)\log(T)}{\eta T} + \frac{\beta\log(|\Pi|T\delta^{-1})\log(T)}{\alpha KT} + 33V_{\mathsf{max}}\sqrt{\frac{\log(|\Pi|T\delta^{-1})}{KT}}$$

$$= \frac{V_{\mathsf{max}}C_{\mathsf{cov}}(\Pi)}{T} + \frac{\alpha\cdot C_{\mathsf{cov}}(\Pi)\log(T)}{\beta\kappa} + \frac{\beta\log(|\Pi|T\delta^{-1})\log(T)}{\alpha KT} + V_{\mathsf{max}}\sqrt{\frac{\log(|\Pi|T\delta^{-1})}{KT}}$$

It follows that by choosing

$$\alpha \propto \sqrt{\frac{\beta\kappa\cdot\beta\log(|\Pi|T\delta^{-1})}{KT\cdot C_{\mathsf{cov}}(\Pi)}},$$

we obtain

$$\frac{1}{T} \sum_{t=1}^{T} J_\beta(\pi_\beta^\star) - J_\beta(\pi^{(t)}) \tag{47}$$

$$\lesssim \frac{V_{\max} C_{\mathsf{cov}}(\Pi)}{T} + \sqrt{\frac{\kappa^{-1} \log(|\Pi|T\delta^{-1}) \log^2(T)) \cdot C_{\mathsf{cov}}(\Pi)}{KT}} + V_{\max} \sqrt{\frac{\log(|\Pi|T\delta^{-1})}{KT}} \tag{48}$$

$$\lesssim \frac{V_{\max} C_{\mathsf{cov}}(\Pi)}{T} + (V_{\max} + \kappa^{-1/2}) \cdot \sqrt{\frac{C_{\mathsf{cov}}(\Pi) \log(|\Pi|\delta^{-1}) \log^2(T)}{KT}}. \tag{49}$$

Finally, we note that $(V_{\max} + \kappa^{-1/2}) = O((V_{\max} + R_{\max})e^{2R_{\max}})$.

$\square$

### D.3 SUPPORTING LEMMAS

**Lemma D.2.** *Suppose that $\widetilde{\pi}^{(t)} = \widetilde{\pi}$ for all $t$. Then for any sequence of functions $\delta^{(1)}, \ldots, \delta^{(T)}$ with $|\delta^{(t)}| \leq B$,*

$$\sum_{t=1}^{T} \mathbb{E}_{\pi^{(t)}, \widetilde{\pi}^{(t-1)}}[\delta^{(t)}(\tau, \widetilde{\tau})] \leq \sqrt{8 C_{\mathsf{cov}}(\Pi) \log(T) \cdot \sum_{t=1}^{T} \sum_{i<t} \mathbb{E}_{\pi^{(i)}, \widetilde{\pi}^{(i)}}[(\delta^{(t)}(\tau, \widetilde{\tau}))^2]} + 2B C_{\mathsf{cov}}(\Pi).$$

**Proof of Lemma D.2.** Define $\boldsymbol{\mu}^{(t)} := \frac{1}{t-1} \sum_{i<t} \pi^{(i)} \otimes \widetilde{\pi}^{(i)}$. Let

$$\nu = \underset{\nu \in \Delta((\mathcal{S}\times\mathcal{A})^H)}{\arg\min} \sup_{\tau \in (\mathcal{S}\times\mathcal{A})^H} \sup_{\pi \in \Pi} \frac{d^\pi(\tau)}{\nu(\tau)}$$

be the distribution that achieves the value of the coverability coefficient in Definition 3.1. Let us abbreviate $C_{\mathsf{cov}} \equiv C_{\mathsf{cov}}(\Pi)$. For a trajectory $\tau$, let

$$\mathsf{t}(\tau) := \min\left\{ t \mid \sum_{i<t} d^{\pi^{(i)}}(\tau) \geq C_{\mathsf{cov}} \cdot \nu(\tau) \right\}.$$

Then we can bound

$$\sum_{t=1}^{T} \mathbb{E}_{\pi^{(t)}, \widetilde{\pi}^{(t-1)}}[\delta^{(t)}(\tau, \widetilde{\tau})]$$

$$\leq \underbrace{\sum_{t=1}^{T} \mathbb{E}_{\pi^{(t)}, \widetilde{\pi}^{(t-1)}}[\delta^{(t)}(\tau, \widetilde{\tau}) \mathbb{I}\{t < \mathsf{t}(\tau)\}]}_{=:(\mathrm{I})}$$

$$+ \underbrace{\sqrt{\sum_{t=1}^{T} \frac{\left(\mathbb{E}_{\pi^{(t)}, \widetilde{\pi}^{(t-1)}}[\delta^{(t)}(\tau, \widetilde{\tau}) \mathbb{I}\{t \geq \mathsf{t}(\tau)\}]\right)^2}{(t-1) \cdot \mathbb{E}_{\boldsymbol{\mu}^{(t)}}[(\delta^{(t)}(\tau, \widetilde{\tau}))^2]}}_{=:(\mathrm{II})} \cdot \sum_{t=1}^{T} \sum_{i<t} \mathbb{E}_{\pi^{(i)}, \widetilde{\pi}^{(i)}}[(\delta^{(t)}(\tau, \widetilde{\tau}))^2].$$

We begin by bounding the first term by

$$(\mathrm{I}) \leq \sum_{t=1}^{T} \mathbb{E}_{\pi^{(t)}, \widetilde{\pi}^{(t-1)}}[\delta^{(t)}(\tau, \widetilde{\tau}) \mathbb{I}\{t < \mathsf{t}(\tau)\}] \leq B \sum_{t=1}^{T} \mathbb{E}_{\pi^{(t)}}[\mathbb{I}\{t < \mathsf{t}(\tau)\}].$$

Letting $\mathcal{T} := (\mathcal{S} \times \mathcal{A})^H$, we can further bound this by

$$\sum_{t=1}^{T} \mathbb{E}_{\pi^{(t)}}[\mathbb{I}\{t < \mathsf{t}(\tau)\}] = \sum_{\tau \in \mathcal{T}} \sum_{t=1}^{T} d^{\pi^{(t)}}(\tau) \mathbb{I}\{t < \mathsf{t}(\tau)\}$$

$$= \sum_{\tau \in \mathcal{T}} \left( \sum_{i=1}^{\mathsf{t}(\tau)-2} d^{\pi^{(i)}}(\tau) \right) + d^{\pi^{(\mathsf{t}(\tau)-1)}}(\tau)$$

$$\leq 2C_{\mathsf{cov}} \sum_{\tau \in \mathcal{T}} \nu(\tau) = 2C_{\mathsf{cov}},$$

so that (I) $\leq 2BC_{\mathsf{cov}}$.

We now bound term (II). Define $d^{\pi,\pi'}(\tau', \widetilde{\tau}') = \mathbb{P}_{s_1 \sim \rho, \tau \sim \pi | s_1, \widetilde{\tau} \sim \pi' | s_1}(\tau = \tau', \widetilde{\tau} = \widetilde{\tau}')$ and $d^{\boldsymbol{\mu}^{(t)}}(\tau', \widetilde{\tau}') = \frac{1}{t-1} \sum_{i<t} d^{\pi^{(i)}, \widetilde{\pi}^{(i)}}(\tau', \widetilde{\tau}')$. For each $t$, we can write

$$\mathbb{E}_{\pi^{(t)}, \widetilde{\pi}^{(t-1)}} \big[ \delta^{(t)}(\tau, \widetilde{\tau}) \mathbb{I}\{t < \mathsf{t}(\tau)\} \big]$$

$$= \sum_{\tau, \widetilde{\tau} \in \mathcal{T}} d^{\pi^{(t)}, \widetilde{\pi}^{(t-1)}}(\tau, \widetilde{\tau}) \delta^{(t)}(\tau, \widetilde{\tau}) \mathbb{I}\{t \geq \mathsf{t}(\tau)\}$$

$$= \sum_{\tau, \widetilde{\tau} \in \mathcal{T}} d^{\pi^{(t)}, \widetilde{\pi}^{(t-1)}}(\tau, \widetilde{\tau}) \delta^{(t)}(\tau, \widetilde{\tau}) \left( \frac{d^{\boldsymbol{\mu}^{(t)}}(\tau, \widetilde{\tau})}{d^{\boldsymbol{\mu}^{(t)}}(\tau, \widetilde{\tau})} \right)^{1/2} \mathbb{I}\{t \geq \mathsf{t}(\tau)\}$$

$$\leq \left( \sum_{\tau, \widetilde{\tau} \in \mathcal{T}} \frac{(d^{\pi^{(t)}, \widetilde{\pi}^{(t-1)}}(\tau, \widetilde{\tau}))^2 \mathbb{I}\{t \geq \mathsf{t}(\tau)\}}{(t-1) \cdot d^{\boldsymbol{\mu}^{(t)}}(\tau, \widetilde{\tau})} \right)^{1/2} \cdot \left( (t-1) \cdot \mathbb{E}_{\boldsymbol{\mu}^{(t)}} \big[ (\delta^{(t)}(\tau, \widetilde{\tau}))^2 \big] \right)^{1/2},$$

where the last inequality is by Cauchy-Schwarz. We conclude that

$$\text{(II)} \leq \sum_{t=1}^{T} \sum_{\tau, \widetilde{\tau} \in \mathcal{T}} \frac{(d^{\pi^{(t)}, \widetilde{\pi}^{(t-1)}}(\tau, \widetilde{\tau}))^2 \mathbb{I}\{t \geq \mathsf{t}(\tau)\}}{(t-1) \cdot d^{\boldsymbol{\mu}^{(t)}}(\tau, \widetilde{\tau})}.$$

To proceed, we use the assumption that $\widetilde{\pi}^{(t)} = \widetilde{\pi}$ for all $t$ for some fixed $\widetilde{\pi}$. We observe that in this case, for all $t$,

$$\frac{d^{\pi^{(t)}, \widetilde{\pi}^{(t-1)}}(\tau, \widetilde{\tau})}{d^{\boldsymbol{\mu}^{(t)}}(\tau, \widetilde{\tau})} = \frac{d^{\pi^{(t)}, \widetilde{\pi}}(\tau, \widetilde{\tau})}{\frac{1}{t-1} \sum_{i<t} d^{\pi^{(i)}, \widetilde{\pi}}(\tau, \widetilde{\tau})} = \frac{d^{\pi^{(t)}}(\tau)}{\frac{1}{t-1} \sum_{i<t} d^{\pi^{(i)}}(\tau)},$$

since $\tau$ and $\widetilde{\tau}$ are conditionally independent given $s_1$, and since $d^{\pi, \pi'}(\tau, \widetilde{\tau}) = 0$ if $\tau, \widetilde{\tau}$ do not share the same $s_1$. It follows that

$$\text{(II)} \leq \sum_{t=1}^{T} \sum_{\tau, \widetilde{\tau} \in \mathcal{T}} \frac{d^{\pi^{(t)}}(\tau) d^{\pi^{(t)}, \widetilde{\pi}}(\tau, \widetilde{\tau}) \mathbb{I}\{t \geq \mathsf{t}(\tau)\}}{\sum_{i<t} d^{\pi^{(i)}}(\tau)}$$

$$= \sum_{\tau} \sum_{t=1}^{T} \frac{(d^{\pi^{(t)}}(\tau))^2 \mathbb{I}\{t \geq \mathsf{t}(\tau)\}}{\sum_{i<t} d^{\pi^{(i)}}(\tau)}$$

$$\leq 2 \sum_{\tau} \sum_{t=1}^{T} \frac{(d^{\pi^{(t)}}(\tau))^2}{\sum_{i<t} d^{\pi^{(i)}}(\tau) + C_{\mathsf{cov}} \nu(\tau)}$$

$$\leq 2C_{\mathsf{cov}} \sum_{\tau} \nu(\tau) \sum_{t=1}^{T} \frac{d^{\pi^{(t)}}(\tau)}{\sum_{i<t} d^{\pi^{(i)}}(\tau) + C_{\mathsf{cov}} \nu(\tau)}.$$

Finally, by Lemma 4 of Xie et al. (2023), we have that for all $\tau \in \mathcal{T}$, $\sum_{t=1}^{T} \frac{d^{\pi^{(t)}}(\tau)}{\sum_{i<t} d^{\pi^{(i)}}(\tau) + C_{\mathsf{cov}} \nu(\tau)} \leq 4 \log(T)$, which yields (II) $\leq 8C_{\mathsf{cov}} \log(T)$. This proves the result.

$\square$

# E  ADDITIONAL PROOFS

This section contains proofs for supporting results found throughout Section 2 and Section 3.

### E.1 PROOFS FROM SECTION 2

**Proof of Proposition 2.1.** Consider the bandit setting where $H = 1$, $\mathcal{S} = \varnothing$, and $\mathcal{A} = \{\mathfrak{a}, \mathfrak{b}\}$. Let $\beta > 0$ be given. We consider the reward function $r$ given by $r(\mathfrak{a}) = 1$ and $r(\mathfrak{b}) = \frac{1}{2}$. We choose the reference model to set $\pi_{\text{ref}}(\mathfrak{a}) = \varepsilon$ and $\pi_{\text{ref}}(\mathfrak{b}) = 1 - \varepsilon$ for a parameter $\varepsilon := \exp(-\frac{c}{\beta})$, where $c > 0$ is an absolute constant whose value will be chosen at the end of the proof. We choose $\Pi = \{\pi_{\text{ref}}, \pi_\beta^\star\}$, which we note satisfies Assumption 3.1 and Assumption 3.2 with $V_{\text{max}} = O(1)$.

Specialized to the bandit setting, Online DPO takes the following simplified form:

1. Sample pair of actions $a^{(t)}, \widetilde{a}^{(t)} \sim \pi^{(t)}$.

2. Label the actions as $(a_+^{(t)}, a_-^{(t)})$ according the Bradley-Terry model:

$$\mathbb{P}(a^{(t)} \succ \widetilde{a}^{(t)}) = \frac{\exp(r(a^{(t)}))}{\exp(r(a^{(t)})) + \exp(r(\widetilde{a}^{(t)}))},$$

and update $\mathcal{D}_{\text{pref}}^{(t+1)} \leftarrow \mathcal{D}_{\text{pref}}^{(t)} \cup \{(a_+^{(t)}, a_-^{(t)})\}$.

3. Compute $\pi^{(t+1)}$ via

$$\pi^{(t+1)} = \underset{\pi \in \Pi}{\operatorname{argmin}} \sum_{(a_+, a_-) \in \mathcal{D}_{\text{pref}}^{(t+1)}} -\log\left[\sigma\left(\beta \log \frac{\pi(a_+)}{\pi_{\text{ref}}(a_+)} - \beta \log \frac{\pi(a_-)}{\pi_{\text{ref}}(a_-)}\right)\right]. \qquad (50)$$

Our construction uses the fact that depending on the preference dataset $\mathcal{D}_{\text{pref}}^{(t)}$, the minimizer in Eq. (50) may not be uniquely defined. Let $\mathcal{E}^{(t)}$ denote the event that at iteration $t$, $a^{(t)} = \widetilde{a}^{(t)} = \mathfrak{b}$. We appeal to a technical lemma.

**Lemma E.1.** *Suppose we initialize with $\pi^{(1)} = \pi_{\text{ref}}$. As long as $c \leq \frac{1}{8}$, $\varepsilon \leq 1/2$, the following properties hold:*

- $\mathbb{P}(\mathcal{E}^{(t)} \mid \mathcal{E}^{(1)}, \dots \mathcal{E}^{(t-1)}) \geq 1 - 2\varepsilon$.

- *Whenever $\mathcal{E}^{(1)}, \dots, \mathcal{E}^{(t)}$ hold, we can choose the policy $\pi^{(t+1)}$ to satisfy $\pi^{(t+1)} = \pi_{\text{ref}}$, which has*

$$\max_\pi J_\beta(\pi) - J_\beta(\pi^{(t+1)}) = \max_\pi J_\beta(\pi) - J_\beta(\pi_{\text{ref}}) \geq \frac{1}{8}$$

By Lemma E.1 and the union bound, we have that

$$\mathbb{P}(\mathcal{E}^{(1)}, \dots, \mathcal{E}^{(T)}) \geq (1 - 2\varepsilon)^T \geq \frac{1}{4},$$

as long as $\varepsilon \leq 1/4$ and $T \leq \frac{1}{2\varepsilon}$. It follows that whenever this occurs, $\max_\pi J_\beta(\pi) - J_\beta(\pi^{(t)}) \geq \frac{1}{8}$ for all $t \in [T + 1]$.

Note that since online DPO selects $\pi^{(t)} = \pi_{\text{ref}}$ for all $t$ in our counterexample above, this also immediately implies a lower bound for offline DPO (interpreting $\pi^{(T+1)}$ as the policy returned by offline DPO).

$\square$

**Proof of Lemma E.1.** We prove this claim inductively. Let $t \in [T]$ be fixed, and suppose the claim holds for $1, \dots, t - 1$. If we assume $\mathcal{E}^{(1)}, \dots, \mathcal{E}^{(t-1)}$ hold, then we have $\pi^{(t)} = \pi_{\text{ref}}$ inductively. In this case,

$$\mathbb{P}(a^{(t)} = \widetilde{a}^{(t)} = \mathfrak{b}) = (\pi_{\text{ref}}(\mathfrak{b}))^2 = (1 - \varepsilon)^2 \geq 1 - 2\varepsilon,$$

so that $\mathbb{P}(\mathcal{E}^{(t)} \mid \mathcal{E}^{(1)}, \dots \mathcal{E}^{(t-1)}) \geq 1 - 2\varepsilon$ as desired.

Now, for the second part of the claim, suppose that $\mathcal{E}^{(1)}, \dots, \mathcal{E}^{(t+1)}$ hold. Then for all $t' \in [t + 1]$, $a_+^{(t')} = a_-^{(t')} = \mathfrak{b}$, which implies that

$$\sum_{(a_+, a_-) \in \mathcal{D}_{\text{pref}}^{(t+1)}} -\log\left[\sigma\left(\beta \log \frac{\pi(a_+)}{\pi_{\text{ref}}(a_+)} - \beta \log \frac{\pi(a_-)}{\pi_{\text{ref}}(a_-)}\right)\right] = -\log(\sigma(0)) \cdot t$$

for all $\pi \in \Pi$ such that $\pi \ll \pi_{\mathsf{ref}}$. It follows that $\pi^{(t+1)} = \pi_{\mathsf{ref}}$ is a valid minimizer for Eq. (50).

Finally, we compute that as long as $\varepsilon \leq 1/2$ and $c \leq \frac{1}{8}$

$$\max_\pi J_\beta(\pi) - J_\beta(\pi_{\mathsf{ref}}) \geq \max_\pi J(\pi) - J(\pi_{\mathsf{ref}}) - \beta \log(\varepsilon^{-1})$$

$$= (1 - (1 - \varepsilon) \cdot \tfrac{1}{2} - \varepsilon \cdot 1) - \beta \log(\varepsilon^{-1}) \geq \frac{1}{4} - c \geq \frac{1}{8}.$$

$\square$

The following hardness result generalizes Proposition E.1 with a large action space construction, which illustrates the necessity of deliberate exploration with an arbitrary reference policy.

**Proposition E.1** (Necessity of deliberate exploration, large action space). *Fix $\beta \in (0, \frac{1}{16 \log(2)})$. Given an arbitrary policy $\pi_{\mathsf{ref}}$, there exists a bandit instance with $H = 1$, $\mathcal{S} = \varnothing$, and $|\mathcal{A}| = K \in [4, \exp(1/8\beta)]$, but $C_{\mathsf{cov}}(\Pi) = O(1)$, such that for all $T \leq \frac{K}{2}$, with constant probability, all of the policies $\pi^{(1)}, \ldots, \pi^{(T+1)}$ produced by Online DPO satisfy*

$$\max_\pi J_\beta(\pi) - J_\beta(\pi^{(t)}) \geq \frac{1}{8} \quad \forall t \in [T+1].$$

**Proof of Proposition E.1.** The proof closely resembles the proof of Proposition 2.1, but with a large action space construction. For completeness and readability, we include the full proof below.

Consider the bandit instance where $H = 1$, $\mathcal{S} = \varnothing$, and $\mathcal{A} = \{\mathfrak{a}_1, \mathfrak{a}_2, \ldots, \mathfrak{a}_K\}$. Let $\beta > 0$ be given. We consider the reward function $r$ given by $r(\mathfrak{a}_1) = 1$ and $r(\mathfrak{a}_2) = r(\mathfrak{a}_3) = \cdots = r(\mathfrak{a}_K) = 0$. Without loss of generality, we suppose $\arg\min_{a \in \mathcal{A}} \pi_{\mathsf{ref}}(a) = \mathfrak{a}_1$ and $\pi_{\mathsf{ref}}(\mathfrak{a}_1) \leq 1/K$ for the given $\pi_{\mathsf{ref}}$ (since we could construct the bandit instance given $\pi_{\mathsf{ref}}$). We choose $\Pi = \{\pi_{\mathsf{ref}}, \pi_\beta^\star\}$, which we note satisfies Assumption 3.1 and Assumption 3.2 with $V_{\mathsf{max}} = O(1)$, as well as $C_{\mathsf{cov}}(\Pi) = O(1)$. This means that the constructed instance has polynomial sample complexity for XPO as shown in Theorem 3.1.

Specialized to the bandit setting, Online DPO takes the following simplified form:

1. Sample pair of actions $a^{(t)}, \widetilde{a}^{(t)} \sim \pi^{(t)}$.

2. Label the actions as $(a_+^{(t)}, a_-^{(t)})$ according the Bradley-Terry model:

$$\mathbb{P}(a^{(t)} \succ \widetilde{a}^{(t)}) = \frac{\exp(r(a^{(t)}))}{\exp(r(a^{(t)})) + \exp(r(\widetilde{a}^{(t)}))},$$

and update $\mathcal{D}_{\mathsf{pref}}^{(t+1)} \leftarrow \mathcal{D}_{\mathsf{pref}}^{(t)} \cup \{(a_+^{(t)}, a_-^{(t)})\}$.

3. Compute $\pi^{(t+1)}$ via

$$\pi^{(t+1)} = \underset{\pi \in \Pi}{\arg\min} \sum_{(a_+, a_-) \in \mathcal{D}_{\mathsf{pref}}^{(t+1)}} -\log \left[ \sigma \left( \beta \log \frac{\pi(a_+)}{\pi_{\mathsf{ref}}(a_+)} - \beta \log \frac{\pi(a_-)}{\pi_{\mathsf{ref}}(a_-)} \right) \right]. \quad (51)$$

Our construction uses the fact that depending on the preference dataset $\mathcal{D}_{\mathsf{pref}}^{(t)}$, the minimizer in Eq. (51) may not be uniquely defined.

Let $\mathcal{E}^{(t)}$ denote the event that at iteration $t$, $a^{(t)} \neq \mathfrak{a}_1$ and $\widetilde{a}^{(t)} \neq \mathfrak{a}_1$. We appeal to a technical lemma.

**Lemma E.2.** *Suppose we initialize with $\pi^{(1)} = \pi_{\mathsf{ref}}$, the following properties hold:*

- $\mathbb{P}(\mathcal{E}^{(t)} \mid \mathcal{E}^{(1)}, \ldots \mathcal{E}^{(t-1)}) \geq 1 - 2/K$.

- *Whenever $\mathcal{E}^{(1)}, \ldots, \mathcal{E}^{(t)}$ hold, we can choose the policy $\pi^{(t+1)}$ to satisfy $\pi^{(t+1)} = \pi_{\mathsf{ref}}$, which has*

$$\max_\pi J_\beta(\pi) - J_\beta(\pi^{(t+1)}) = \max_\pi J_\beta(\pi) - J_\beta(\pi_{\mathsf{ref}}) \geq \frac{1}{4}$$

By Lemma E.2 and the union bound, we have that

$$\mathbb{P}(\mathcal{E}^{(1)},\ldots,\mathcal{E}^{(T)}) \geq (1-2/K)^T \geq \frac{1}{4},$$

as long as $K \geq 4$ and $T \leq \frac{K}{2}$. It follows that whenever this occurs, $\max_\pi J_\beta(\pi) - J_\beta(\pi^{(t)}) \geq \frac{1}{8}$ for all $t \in [T+1]$.

Note that since online DPO selects $\pi^{(t)} = \pi_{\mathsf{ref}}$ for all $t$ in our counterexample above, this also immediately implies a lower bound for offline DPO (interpreting $\pi^{(T+1)}$ as the policy returned by offline DPO).

$\square$

**Proof of Lemma E.1.** We prove this claim inductively. Let $t \in [T]$ be fixed, and suppose the claim holds for $1,\ldots,t-1$. If we assume $\mathcal{E}^{(1)},\ldots,\mathcal{E}^{(t-1)}$ hold, then we have $\pi^{(t)} = \pi_{\mathsf{ref}}$ inductively. In this case,

$$\mathbb{P}(a^{(t)} \neq \mathfrak{a}_1, \widetilde{a}^{(t)} \neq \mathfrak{a}_1) = (1-\pi_{\mathsf{ref}}(\mathfrak{a}_1))^2 = \left(1-\frac{1}{K}\right)^2 \geq 1-\frac{2}{K},$$

so that $\mathbb{P}(\mathcal{E}^{(t)} \mid \mathcal{E}^{(1)},\ldots\mathcal{E}^{(t-1)}) \geq 1-2/K$ as desired.

Now, for the second part of the claim, suppose that $\mathcal{E}^{(1)},\ldots,\mathcal{E}^{(t+1)}$ hold. Then for all $t' \in [t+1]$, $\frac{\pi(a_+^{(t')})}{\pi_{\mathsf{ref}}(a_+^{(t')})} = \frac{\pi(a_-^{(t')})}{\pi_{\mathsf{ref}}(a_-^{(t')})}$ for all $\pi \in \Pi$, because $\beta \log \frac{\pi_\beta^\star(a_+^{(t')})}{\pi_{\mathsf{ref}}(a_+^{(t')})} - \beta \log \frac{\pi_\beta^\star(a_-^{(t')})}{\pi_{\mathsf{ref}}(a_-^{(t')})} = r(a_+^{(t')}) - r(a_-^{(t')}) = 0$, which implies that

$$\sum_{(a_+,a_-)\in\mathcal{D}_{\mathsf{pref}}^{(t+1)}} -\log\left[\sigma\left(\beta\log\frac{\pi(a_+)}{\pi_{\mathsf{ref}}(a_+)} - \beta\log\frac{\pi(a_-)}{\pi_{\mathsf{ref}}(a_-)}\right)\right] = -\log(\sigma(0))\cdot t$$

for all $\pi \in \Pi$ such that $\pi \ll \pi_{\mathsf{ref}}$. It follows that $\pi^{(t+1)} = \pi_{\mathsf{ref}}$ is a valid minimizer for Eq. (51).

Finally, we compute that as long as $\varepsilon \leq 1/2$

$$\max_\pi J_\beta(\pi) - J_\beta(\pi_{\mathsf{ref}}) \geq \max_\pi J(\pi) - J(\pi_{\mathsf{ref}}) - \beta\log(K)$$

$$= \frac{\exp(1/\beta)}{\exp(1/\beta) + K - 1} - \frac{1}{K} - \beta\log(K)$$

$$\geq \frac{\exp(1/\beta)}{\exp(1/\beta) + \exp(1/8\beta)} - \frac{1}{4} - \frac{1}{8} \geq \frac{1}{8}.$$

$\square$

