# OpenReview forum: "Exploratory Preference Optimization: Harnessing Implicit Q*-Approximation for Sample-Efficient RLHF"
_ICLR.cc/2025/Conference — ICLR 2025 Poster_

### Official Review · Reviewer_SkL8 · 2024-10-26

**Soundness:** 3
**Presentation:** 3
**Contribution:** 3
**Rating:** 6
**Confidence:** 4

**Summary:**

This paper introduces an algorithm called XPO, designed to enable efficient online exploration using preference feedback and general function approximation within KL-regularized MDPs with deterministic transitions. With just a single-line modification to online DPO, XPO is provably sample-efficient and reliably converges to a near-optimal language model policy under standard exploration conditions. Importantly, XPO can achieve optimal policy convergence regardless of the initial model’s coverage.

**Strengths:**

Overall, the core idea of this paper—introducing a novel approach to promote exploration—is intriguing, even though it is demonstrated in a limited setting, specifically deterministic MDPs. Achieving sufficient exploration with only a single-line modification to the DPO algorithm could have a non-trivial practical impact.

Additionally, the theoretical result revealing the limitations of Online DPO (Proposition 2.1) is particularly compelling and supports the need for deliberate exploration.

**Weaknesses:**

1. Although the proposed algorithm is a simple extension of DPO and guarantees a strong sample efficiency, it remains unclear why the algorithm is considered *highly practical* (Line 312). While I understand that this is a theoretical paper, the lack of empirical experiments makes it difficult to be convinced of the algorithm's practicality. Without experiments, one of the authors' main claims—ease of implementation—is not fully persuasive. I am not suggesting that experiments are required; it is simply a limitation of the paper.

2. The optimism parameter $\alpha$ depends on Coverability $C_{cov}(\Pi)$, which is hard to know in practice.

**Questions:**

1. Line 408: You mentioned that the optimistic exploration objective is differentiable and directly amenable to implementation with language models. Could you explain this in more detail?
2. Are there any advantages to using a token-level MDP (where $s_h$ is simply the initial state and sequence of actions) compared to the contextual (dueling) bandit setting, where each action corresponds to an entire text rather than a single token? In Line 762, you mentioned that the token-level MDP is strictly more general than the contextual bandit formulation. However, since your setting is quite restricted, this claim is not clear enough to me.
3. In Lemma C.1, it appears that the SEC depends on the value of $\beta$. Does the upper bound of the SEC also depend on $\beta$? If so, it would be helpful to explicitly show this dependence on $\beta$.

---

> ### Author Response · Authors · 2024-11-20
> **Authors' response (1/2)**
>
> Thank you for the positive review! Please see responses to individual questions below.
>
> ---
> > Although the proposed algorithm is a simple extension of DPO and guarantees a strong sample efficiency, it remains unclear why the algorithm is considered highly practical (Line 312). While I understand that this is a theoretical paper, the lack of empirical experiments makes it difficult to be convinced of the algorithm's practicality. Without experiments, one of the authors' main claims—ease of implementation—is not fully persuasive. I am not suggesting that experiments are required; it is simply a limitation of the paper.
>
> We would like to briefly emphasize that this is a theoretical paper submitted to the learning theory category. The main result in the paper is Theorem 3.1, which is the first theoretical sample complexity result which offers convergence to a near-optimal policy in any Deterministic Contextual MDP (which subsumes popular application settings like LLM policy) under natural exploration conditions (and does so with a provably efficient algorithm and general function approximation). We believe this result on its own represents a significant advance in the theory of reinforcement learning, and is sufficient to merit acceptance.
>
> ---
> > The optimism parameter $\alpha$ depends on Coverability $C_\textsf{cov}(\Pi)$, which is hard to know in practice.
>
> In practice, $\alpha$ should be treated as a hyperparameter and tuned tuned. This limitation is shared by virtually all optimistic methods in contextual bandits and reinforcement learning, such as LinUCB [1].
>
> ---
> > Line 408: You mentioned that the optimistic exploration objective is differentiable and directly amenable to implementation with language models. Could you explain this in more detail?
>
> For the coverability/SEC setting considered in our paper, the only prior algorithms for optimistic exploration are OLIVE [1] and GOLF [2], which involve multiple non-differentiable argmax operators over action spaces (for construction of confidence sets, and in the implementation of optimism over the confidence set). Our objective, which we derive from a KL-regularized dynamic programming perspective, removes these explicit argmax operators in a principled fashion, leading to a fully differentiable objective.
>
> In addition, unlike the aforementioned algorithms [1] and [2], which separately estimate value functions and policies, our algorithm is purely (language model) policy-based, similar to DPO itself. As a result, its implementation simply involves adding a single line to the online DPO loss function.
>
> ---
> > Are there any advantages to using a token-level MDP (where $s_h$ is simply the initial state and sequence of actions) compared to the contextual (dueling) bandit setting, where each action corresponds to an entire text rather than a single token? In Line 762, you mentioned that the token-level MDP is strictly more general than the contextual bandit formulation. However, since your setting is quite restricted, this claim is not clear enough to me.
>
> It is important to note that our setting is more general than a token-level MDP, and captures any deterministic contextual MDP, even if the transition dynamics are unknown a-priori. This is a natural formulation which captures many interesting multi-turn settings that go beyond vanilla single-turn question answering. Promising application cases include self-correction [4] and tool-integrated agents [5], which cannot be simply modeled as contextual bandits. This is because, for example, the output from external tools is part of deterministic state transition rather than part of action. And in this case, our algorithm will not take gradients on the output from external tools due to our definition in line 126, whereas a naive application of the contextual bandit formulation will still take gradients on the output from external tools. This deficiency in the contextual bandits model has been empirically verified by [5].
>
> Let us also mention that while the special case of the token-level MDP is equivalent to a contextual dueling bandit problem statistically, algorithms that work in the contextual bandit formulation may not necessarily be computationally efficient when we specialize them to the token-level MDP (e.g., if they require explicit optimization over the contextual bandit "action space," which corresponds to all sequences of tokens in the token-level MDP). So we believe that explicitly showing that our algorithm is efficient in the token-level formulation is an important sanity check.

---

> ### Author Response · Authors · 2024-11-20
> **Authors' response (2/2)**
>
> > In Lemma C.1, it appears that the SEC depends on the value of $\beta$. Does the upper bound of the SEC also depend on $\beta$? If so, it would be helpful to explicitly show this dependence on $\beta$.
>
> For most natural settings, the upper bounds on the SEC do not depend on $\beta$. For example, if $\pi$ is the set of softmax policies induced by a reward function class, then the dependence on \beta in the SEC definition vanishes, and it coincides with standard value/reward-based definitions from prior work. See Example C.1 and C.2 for natural special cases where this happens.
>
> Additionally, $\beta$ is part of the problem setting/definition (Eq.(2) on page 3) in KL-regularized reinforcement learning, as it is not unique to our specific algorithm. For example, other algorithms like (online) DPO would also require dependence on $\beta$ if we wanted to use the SEC (or other similar complexity measures) to analyze them in this setting.
>
> ---
> Reference:
> [1] A Contextual-Bandit Approach to Personalized News Article Recommendation. Li et al. 2010
> [2] Contextual decision processes with low bellman rank are pac-learnable. Jiang et al. 2018
> [3] Bellman Eluder Dimension: New Rich Classes of RL Problems, and Sample-Efficient Algorithms. Jin et al. 2021
> [4] Training Language Models to Self-Correct via Reinforcement Learning. Kumar et al. 2024
> [5] Building math agents with multi-turn iterative preference learning. Wei et al. 2024

---

> ### Comment · Reviewer_SkL8 · 2024-11-22
>
> Thank you for the detailed explanation. As the next step, I really look forward to seeing successful empirical results based on the proposed algorithm. I will keep my positive response and have increased my confidence.

---

### Official Review · Reviewer_Yamv · 2024-10-26

**Soundness:** 3
**Presentation:** 4
**Contribution:** 3
**Rating:** 6
**Confidence:** 4

**Summary:**

The paper gives a theoretical analysis of online RLHF. In particular, they proposed a XPO algorithm that enjoys sample complexity guarantee.

**Strengths:**

1. The literature review is extensive and helpful for the readers. Due to the fact that RLHF is currently a fast-growing domain, the paper gives a nice comparison between its contribution and concurrent works helps the reader understand how to posit their work.
2. Their theoretical analysis is meaningful. In particular, I like the general algorithm 2 provided in the Appendix C.1.

**Weaknesses:**

1. Due to that it is a theoretical paper, I do wonder the practical implication of XPO--- I looked up the online DPO work the authors referenced, and their empirical results made me believe in this work (or online approach in general) more.

**Questions:**

1. The regret bound is scaling down w.r.t. the number of iterations. And for every model update, it just requires one new pair. In practice it sounds very costly. In the simplicity session (line 311), the authors also suggest cutting down the number of iterations and sampling more data during each iteration. I wonder how do they anticipate this way to guarantee a good policy if their analysis is compeletely dependent on a good number of iterations.

---

> ### Author Response · Authors · 2024-11-20
> **Authors' response**
>
> Thank you for the positive review! Please see responses to individual questions below.
>
> ---
> > Due to that it is a theoretical paper, I do wonder the practical implication of XPO---I looked up the online DPO work the authors referenced, and their empirical results made me believe in this work (or online approach in general) more.
>
> We do plan to evaluate XPO empirically, but we view this as follow-up work to this theoretical investigation. In particular, we expect the principle of deliberate exploration, as used in XPO, to be most beneficial in settings where an explicit verifier is available (e.g., math or coding). Performing a full-scale evaluation for such settings is quite non-trivial, which is why we leave this for followup work.
>
> We would also like to briefly emphasize that this is a theoretical paper submitted to the learning theory category. The main result in the paper is Theorem 3.1, which is the first theoretical sample complexity result which offers convergence to a near-optimal policy in any Deterministic Contextual MDP (which subsumes popular application settings like LLM policy) under natural exploration conditions (and does so with a provably efficient algorithm and general function approximation). We believe this result on its own represents a significant advance in the theory of reinforcement learning, and is sufficient to merit acceptance.
>
> ---
> > The regret bound is scaling down w.r.t. the number of iterations. And for every model update, it just requires one new pair. In practice it sounds very costly. In the simplicity session (line 311), the authors also suggest cutting down the number of iterations and sampling more data during each iteration. I wonder how do they anticipate this way to guarantee a good policy if their analysis is compeletely dependent on a good number of iterations.
>
> This is a very interesting question:
> 1. For the current analysis, to learn an $\varepsilon$-optimal policy, we use $1/\varepsilon^2$ iterations and draw one pair per iteration. We can show that the analysis will still go through if we instead use $1/\varepsilon$ iterations and draw a large batch of $1/\varepsilon$ pairs per iteration. We would be happy to include this extension in the final version of the paper.
> 2. Whether we can use fewer than $1/\varepsilon$ iterations is a very interesting question. This is likely possible under stronger assumptions (e.g., linear function approximation), but it is not clear if this is possible in general. We emphasize that RL with low-switching/batches/deployment efficiency is an active area of research [1-5], which is not yet well understood. As this is complementary to the point of our paper, we leave a deeper understanding for future work.
>
> ---
> Reference:
> [1] Online learning with switching costs and other adaptive adversaries. Cesa-Bianchi et al. 2013
> [2] Provably efficient q-learning with low switching cost. Bai et al. 2019
> [3] Batched multi-armed bandits problem. Gao et al. 2019
> [4] Deployment-efficient reinforcement learning via model-based offline optimization. Matsushima et al. 2020
> [5] Towards deployment-efficient reinforcement learning: Lower bound and optimality. Huang et al. 2022

---

> > ### Comment · Reviewer_Yamv · 2024-11-21
> >
> > Thank your for your clarification.
> > Yes, please add the further analysis to the revision. I am very interested in reading it.
> > I have increaed my confidence score.

---

> > > ### Author Response · Authors · 2024-11-25
> > >
> > > Thank you for the interest! We have updated the revision to include this result. See Appendix D and Theorem D.1. As described in our initial response, this result uses $1/\varepsilon$ policy updates/iterations, and draws a batch of $1/\varepsilon$ preference pairs at each iteration.

---

> > > > ### Author Response · Authors · 2024-11-30
> > > > **Following up re: using fewer iterations**
> > > >
> > > > Dear reviewer Yamv,
> > > >
> > > > We wanted to briefly follow up to see if you had a chance to look at the new result we have added regarding using fewer iterations (Theorem D.1 in Appendix D). Please let us know if this result addresses any of your concerns.
> > > >
> > > > Thank you,
> > > > Authors

---

### Official Review · Reviewer_hYT9 · 2024-10-30

**Soundness:** 3
**Presentation:** 4
**Contribution:** 4
**Rating:** 8
**Confidence:** 4

**Summary:**

This paper proposes a simple yet provably efficient RLHF algorithm called XPO with general function approximation. The authors focus on a KL-regularized Deterministic Contextual MDP (DCMDP) and decompose the regret into policy expression, thereby formulating it within a direct preference optimization framework. By employing a coverability measure and implicit optimism, XPO provides a provably guaranteed sampling policy of preference pairs, which is a notable theoretical contribution in the RLHF field.

**Strengths:**

- Despite the limited references due to the field's recent development, the authors have effectively organized concurrent studies and provided ample comparisons with their own contributions.

- The authors clearly address the paper's limitation in focusing on DCMDP, which is valuable information for future researchers working on related papers.

- Particularly in Section 3, the interpretation of theoretical results is well-explained in an intuitive manner, with concepts from conventional regret analysis effectively connected within the DPO perspective.

**Weaknesses:**

- Although the paper lacks experimental content, this is a minor drawback given the theoretical nature and significance of its contributions. The emphasis on simplicity in the main text raises questions on whether the authors have experience conducting practical experiments with XPO or plan to do so.

- It appears that some implicit assumptions are used in the main text; depending on the authors' response, I am willing to increase the score further. Please refer to the "Questions" section for more details.

**Questions:**

-	What are the authors' views on the practicality of human "online" feedback in the online learning process?

-	In Line 188 and Line 4 of the pseudocode, $y_t \sim \mathbb{P}(\tau^{(t)} \succ \tilde{\tau}^{(t)})$ suggests that feedback follows the preference model. If labeling is done via human online feedback, it seems quite a strong assumption that this property is satisfied in every iteration. Was this assumption necessary for theoretical development?

-	While I am familiar with regret analysis using eluder dimension, I lack a deep understanding of coverability. Specifically, as stated in Remark 3.3, coverability at the trajectory level appears to increase exponentially with $H$, suggesting a much larger upper bound compared to the eluder dimension. Are there specific advantages of coverability over eluder dimension?

-	Theorem 3.1 requires $|\Pi|$ to be defined, suggesting an implicit assumption that the policy class is finite. Could this be addressed by a covering number?

---

> ### Author Response · Authors · 2024-11-20
> **Authors' response (1/2)**
>
> Thank you for the positive review! Please see responses to individual questions below.
>
> ---
> > Although the paper lacks experimental content, this is a minor drawback given the theoretical nature and significance of its contributions. The emphasis on simplicity in the main text raises questions on whether the authors have experience conducting practical experiments with XPO or plan to do so.
>
> We do plan to evaluate XPO empirically, but we view this as follow-up work to this theoretical investigation. In particular, we expect the principle of deliberate exploration, as used in XPO, to be most beneficial in settings where an explicit verifier is available (e.g., math or coding). However, performing a full-scale evaluation for such settings is quite non-trivial, which is why we leave this for followup work.
>
> ---
> > What are the authors' views on the practicality of human "online" feedback in the online learning process?
>
> We see several practical scenarios where online feedback could be effectively implemented:
> 1. AI Feedback / Learned Reward Models: Another alternative is to use large language models as automated feedback/reward providers, which is commonly used in industry [3], and an active area of academic research with a growing community (see, e.g., [4]).
> 2. Automated Verification Settings: There are domains with explicit verification mechanisms, for example, math (Lean Prover) and coding (Python interpreter).
> 3. Batched Mode / Low-switching Cost Scenario: The XPO algorithm can be applied in a batch mode in which the algorithm makes a small number of updates, but draws a large batch of responses per update. This would be more practical for the case of human feedback. This variant of the algorithm can achieve theoretical guarantees, but this is beyond the scope of this paper (RL with few switches/large batches is a complementary topic with its own dedicated line of research [e.g., 1,2]); please see the response to reviewer Yamv for more detail.
>
> Due to the high engineering overhead, we view this as beyond the scope of the present theoretical investigation.
>
> ---
> > In Line 188 and Line 4 of the pseudocode, $y_t \sim \mathbb P(\tau^{(t)} \succ\tilde\tau^{(t)})$ suggests that feedback follows the preference model. If labeling is done via human online feedback, it seems quite a strong assumption that this property is satisfied in every iteration. Was this assumption necessary for theoretical development?
>
> We agree that assuming binary preference $y_t \sim \mathbb P(\tau^{(t)} \succ\tilde\tau^{(t)})$ is an extra assumption, as we mentioned in line 145. Using the Bradley-Terry model to model human preference, as we did in this paper, is a standard assumption for modeling human preference in both RLHF/PBRL literature and industry (e.g., InstructGPT). Going beyond Bradley-Terry, such as general preferences, or handling misspecification/outliers both seem like interesting directions for future work. Additionally, such assumption is easily satisfied via the learned reward models (our algorithm can also work with explicit reward signals under minor change), but translation to good performance with respect to actual human preferences or benchmarks then depends on the quality of the learned reward model (this is another separate research topic see, e.g., [4]).
>
> ---
> > While I am familiar with regret analysis using eluder dimension, I lack a deep understanding of coverability. Specifically, as stated in Remark 3.3, coverability at the trajectory level appears to increase exponentially with H, suggesting a much larger upper bound compared to the eluder dimension. Are there specific advantages of coverability over eluder dimension?
>
> We present Theorem 3.1 in terms of coverability to keep the result as simple and compact as possible. More general guarantees, based on a complexity measure called the Sequential Extrapolation Coefficient (SEC) [5] which generalizes both Coverability and the Eluder Dimension, are provided in the appendix as Theorem 3.1'.
>
> It's important to note that SEC is not always exponential in $H$. For example, in linear MDPs, the SEC can be bounded by $O(dH)$, as shown in Example C.2.
>
> Furthermore, existing work [5, Section 5] has demonstrated that traditional complexity measures, including eluder dimension and its extensions like Bellman eluder dimension, are insufficient to capture coverability, as shown through hardness instances. SEC bridges this gap.

---

> ### Author Response · Authors · 2024-11-20
> **Authors' response (2/2)**
>
> > Theorem 3.1 requires $|\Pi|$ to be defined, suggesting an implicit assumption that the policy class is finite. Could this be addressed by a covering number?
>
> Yes, we present the result for finite classes only to keep presentation as compact as possible, but an extension to infinite classes is quite straightforward, and only requires change our concentration analysis for maximum likelihood; we briefly remark on this in Footnote 4, and would be happy to include this extension in the final version of the paper.
>
> ---
> Reference:
> [1] Online learning with switching costs and other adaptive adversaries. Cesa-Bianchi et al. 2013
> [2] Provably efficient q-learning with low switching cost. Bai et al. 2019
> [3] Constitutional AI: Harmlessness from AI Feedback. Bai et al. 2022
> [4] https://huggingface.co/spaces/allenai/reward-bench
> [5] The Role of Coverage in Online Reinforcement Learning. Xie et al. 2023

---

> ### Comment · Reviewer_hYT9 · 2024-11-22
>
> The author's response addressed the concerns I raised. I read the paper with great interest, as it contributes to deepening the theoretical understanding of the RLHF domain. Therefore, I have raised the score.

---

### Official Review · Reviewer_G41J · 2024-11-03

**Soundness:** 2
**Presentation:** 3
**Contribution:** 2
**Rating:** 6
**Confidence:** 3

**Summary:**

This paper considers the sample complexity of online reinforcement learning with human feedback (RLHF). For contextual MDPs with deterministic transitions, the authors show that an optimism-based variant of Direct Preference Optimization (DPO) is provably efficient with general function approximation.

**Strengths:**

1. The proposed method is simple to implement;
2. The paper shows that online DPO can fail if $\pi_\text{ref}$ does not have good coverage (Proposition 2.1);
3. A connection is built between DPO and Bellman error minimization for DCMPDs (Lemma C.3)

**Weaknesses:**

1. My major concern with the paper is on the assumption of deterministic transitions, which ignores the challenge of learning transitions in RL;
2. The paper does not provide numerical results to show the effectiveness of the proposed method.

**Questions:**

1. The hardness result in Proposition 2.1 can be bypassed by setting $\pi_\text{ref}$ to be uniform. I think an alternative way is to show that for *any* $\pi_\text{ref}$, there exists a hard DCMDP instance such that the lower bound either 1) scales with the action space (e.g., for uniform policy), which can be prohibitively large for token-level MDPs, or 2) scales with, e.g., $\exp(1/\beta)$.
2. Can we just output the uniform mixture of $\pi^{(1)},\cdots,\pi^{(T+1)}$ in line 7 of Algorithm 1? In this case, we do not need validation data.

---

> ### Author Response · Authors · 2024-11-20
> **Authors' response**
>
> Thank you for the review! Please see responses to individual questions below.
>
> ---
> > My major concern with the paper is on the assumption of deterministic transitions, which ignores the challenge of learning transitions in RL
>
> While deterministic contextual MDPs are indeed more restrictive than general RL, we believe our work constitutes an important contribution for a few reasons:
> 1. The deterministic transition assumption stems from fundamental limitations of the DPO objective itself, as noted in recent works [e.g., 1,2], rather than being a limitation specific to our approach.
> 2. Deterministic transitions do not remove the challenge of learning transitions in RL—the deterministic transition function is still unknown to the learner, and to obtain guarantees like our Theorem 3.1 which are independent of the number of states, we need to be able to generalize across the (stochastic) contexts.
> 3. Our formulation of the problem is strictly more general than the vast number of other previous RLHF (theory) works, which only treat the problem as a contextual bandit [e.g., 3-8] with H=1.
> Thus, we believe our focus on deterministic contextual MDPs allows us to address a very important subset of practical applications (including LLM post-training), while nonetheless advancing the field of provably efficient and practical RL exploration algorithms through theoretical innovations.
>
> Lastly, we mention in passing that various RL settings with function approximation are known to be computationally and statistically hard under even deterministic transitions [e.g., 9,10]. Hence, we believe that showing the assumptions in our paper lead to tractable algorithms is an important finding.
>
> ---
> > The paper does not provide numerical results to show the effectiveness of the proposed method.
>
> We emphasize that our submission is a theoretical paper submitted to the learning theory category. The main result in the paper is Theorem 3.1, which is the first theoretical sample complexity result which offers convergence to a near-optimal policy in any Deterministic Contextual MDP (which subsumes popular application settings like LLM policy) under natural exploration conditions (and does so with a provably efficient algorithm and general function approximation). This result—not the XPO algorithm itself—is our main contribution, and we believe it represents a significant advance in the theory of reinforcement learning.
>
> ---
> > The hardness result in Proposition 2.1 can be bypassed by setting πref to be uniform. I think an alternative way is to show that for any $\pi_\text{ref}$, there exists a hard DCMDP instance such that the lower bound either 1) scales with the action space (e.g., for uniform policy), which can be prohibitively large for token-level MDPs, or 2) scales with, e.g., $\exp(1/\beta)$.
>
> We agree that the hardness in Proposition 2.1 can be bypassed by choosing $\pi_\text{ref}$ to be uniform, but this is precisely the point of our result: online DPO can fail when $\pi_\text{ref}$ is non-uniform, but this failure can be addressed through deliberate exploration.
>
> However, we find the reviewer's proposal interesting, and it turns out that we can indeed prove a lower bound that holds for arbitrary $\pi_\text{ref}$ using a small modification to the proof of Proposition 2.1. As the reviewer suggested, this result uses a large action space, and shows that for any $\pi_\text{ref}$, there exists a DCMDP for which the suboptimality of Online DPO scales with $ \min \big( |Y|, \exp(1/\beta) \big) $ ($|Y|$ denotes the number of actions). The new result is included at the end of Appendix D.
>
> ---
> > Can we just output the uniform mixture of $\pi^{(1)}, \dotsc, \pi^{(T)}$ in line 7 of Algorithm 1? In this case, we do not need validation data.
>
> Yes, this would work as well. We are happy to remark on this in the final version of the paper.
>
> ---
> Reference:
> [1] From r to Q∗: Your Language Model is Secretly a Q-Function. Rafailov et al. 2024
> [2] Building Math Agents with Multi-Turn Iterative Preference Learning. Xiong et al. 2023
> [3] Direct Preference Optimization: Your Language Model is Secretly a Reward Model. Rafailov et al. 2023
> [4] Statistical Rejection Sampling Improves Preference Optimization. Liu et al. 2023
> [5] A General Theoretical Paradigm to Understand Learning from Human Preferences. Azar et al. 2023
> [6] Nash Learning from Human Feedback. Munos et al. 2023
> [7] Iterative Preference Learning from Human Feedback: Bridging Theory and Practice for RLHF under KL-Constraint. Xiong et al. 2023
> [8] KTO: Model Alignment as Prospect Theoretic Optimization. Ethayarajh et al. 2024
> [9] On Oracle-Efficient PAC RL with Rich Observations. Dann et al. 2018
> [10] Computational-Statistical Gaps in Reinforcement Learning. Kane et al. 2022

---

> > ### Comment · Reviewer_G41J · 2024-11-20
> > **Thank you for the response**
> >
> > The authors' response has addressed most of my concerns. I have raised the score accordingly.

---

### Meta-Review · Area_Chair_b6w6 · 2024-12-20

**Metareview:**

Summary:
This work establishes a theoretical foundation for RLHF (Reinforcement Learning with Human Feedback) with general function approximation. By incorporating the optimistic exploration concept widely used in the RL community, the authors propose a general framework that unifies the treatment of various reward models and policies.

Strengths:
- The theoretical results are rigorous and well-founded.
- The presentation is clear and easy to follow.

Weaknesses:
As pointed out by some reviewers, while the authors propose new RLHF algorithms, they do not include corresponding experiments to demonstrate their empirical effectiveness. Although the lack of experiments does not weigh heavily in my decision to accept, I encourage the authors to include some initial empirical evaluations to support the practical utility of their methods.

Decision: Accept.

**Additional Comments On Reviewer Discussion:**

There were no significant changes to the paper during the discussion phase.

---

### Decision · Program_Chairs · 2025-01-22

Accept (Poster)